# Active Learning with Safety Constraints

**Romain Camilleri, Andrew Wagenmaker, Jamie Morgenstern, Lalit Jain, Kevin Jamieson**
University of Washington, Seattle, WA
{camilr,ajwagen,jamiemmt,jamieson}@cs.washington.edu,lalitj@uw.edu

## Abstract

Active learning methods have shown great promise in reducing the number of samples necessary for learning. As automated learning systems are adopted into real-time, real-world decision-making pipelines, it is increasingly important that such algorithms are designed with *safety* in mind. In this work we investigate the complexity of learning the best *safe* decision in interactive environments. We reduce this problem to a constrained linear bandits problem, where our goal is to find the best arm satisfying certain (unknown) safety constraints. We propose an adaptive experimental design-based algorithm, which we show efficiently trades off between the difficulty of showing an arm is unsafe vs suboptimal. To our knowledge, our results are the first on best-arm identification in linear bandits with safety constraints. In practice, we demonstrate that this approach performs well on synthetic and real world datasets.

## 1 Introduction

In many problems in online decision-making, the goal of the learner is to take measurements in such a way as to learn a near-optimal policy. Oftentimes, though the space of policies may be large, the set of feasible, or safe policies could be much smaller, effectively constraining the search space of the learner. Furthermore, these constraints may themselves depend on unknown problem parameters.

For example, consider the problem of bidding sequentially in a series of auctions where the bidder bids a price $w_t$, the value of winning an item $t$ is denoted $v_t$, and the utility of winning that item and paying price $p_t$ is $v_t - p_t$. The goal of the bidder is to choose an optimal strategy amongst bidding strategies $s \in S, s : \mathbb{R} \to \mathbb{R}$. When a bidder is deciding how to choose these strategies, they often face constraints: they may have a budget $B$ they must abide to; they may wish to have those auctions they win be well-distributed across time (e.g. in the case of advertising campaigns); they may want to ensure the set of items they win satisfy some other property (e.g. for advertisements, they might want to ensure they are not over-targeting any demographic group).

As another example, inventory management systems may face similar issues of deciding amongst strategies, where there is some objective function (such as revenue) and a variety of constraints at play in this choice (e.g. capacity of a set of warehouses, employee scheduling constraints, or limits on the duration of delivery lag). They also operate in markets with changing demand and other uncertainties, leading to uncertainty about which strategies are feasible or safe (satisfy constraints) and uncertainty about the revenue they generate.

Both of these scenarios motivate understanding the sample complexity of selecting an action or strategy which approximately maximizes an objective while also satisfying some constraints, where samples are needed to both learn the objective value of actions and whether or not they satisfy said constraints. In this work, we study the *active* sample complexity of this task—if the learner can choose which examples to observe and have labeled, how many fewer samples might they need compared to the number needed in a passive setting? We pose this as a best-arm identification problem in the setting of linear bandits with safety constraints, where the goal is to estimate the best arm, subject to it meeting certain (initially unknown) safety constraints. We propose an experiment design-based algorithm which efficiently learns the best safe decision, and show the efficacy of this

36th Conference on Neural Information Processing Systems (NeurIPS 2022).

approach in practice through several experimental examples. To the best of our knowledge, ours is the first approach to handle best-arm identification in linear bandits with safety constraints.

## 1.1 Linear Bandits with Safety Constraints

Let $\delta \in (0, 1)$ be a confidence parameter, $\mathcal{X}, \mathcal{Z} \subseteq \mathbb{R}^d$ be finite known sets of vectors, and assume there exists $\theta_* \in \mathbb{R}^d$, $\mu_* \in \mathbb{R}^{m \times d}$ unknown to the learner. For simplicity, we assume that $\|\theta_*\|_2 \leq 1$, and $\|\mu_{*,i}\|_2 \leq 1, i \in [m]$ and $\|x\|_2 \leq 1, \|z\|_2 \leq 1, \forall x \in \mathcal{X}, z \in \mathcal{Z}$. The learner plays according to the following protocol: at each time step $t$ the learner chooses some action $x_t \in \mathcal{X}$, observes $(r_t, \{s_{t,i}\}_{i=1}^m)$ where $r_t = \theta_*^\top x_t + w_t^\theta$ and $s_{t,i} = \mu_{*,i}^\top x_t + w_{t,i}^\mu$ for all $i \in [m]$, where $w_t^\theta, w_{t,i}^\mu$ are i.i.d. mean zero 1-subGaussian noise. The choice of action $x_t$ is measurable with respect to the history $\mathcal{F}_t = \{(x_j, r_j, \{s_{j,i}\}_{i=1}^m)\}_{j=1}^{t-1}$. The learner stops at a stopping time $\tau_\delta$ which is measurable with respect to the filtration generated by $\mathcal{F}_{t \leq \tau}$, and returns $\widehat{z}_\tau \in \mathcal{Z}$. In general, when referring to any expectation $\mathbb{E}$ or probability $\mathbb{P}$, the underlying measure will be with respect to the actions, observed rewards, and internal randomness of the algorithm.

We are interested in the *safe transductive best-arm identification problem* (**STBAI**), where the goal of the learner is to identify

$$z_* := \arg\max_{z \in \mathcal{Z}} z^\top \theta_* \quad \text{s.t.} \quad z^\top \mu_{*,i} \leq \gamma, \forall i \in [m]$$

for some (known) threshold $\gamma$. In words, our goal is to identify the best *safe* arm in $\mathcal{Z}$, $z_*$, where we say an arm $z$ is safe if it satisfies every linear constraint: $z^\top \mu_{*,i} \leq \gamma, \forall i \in [m]$. We are interested in obtaining learners that take the fewest number of samples possible to accomplish this. In practice, we will consider a slightly easier objective. Fix some tolerance $\epsilon > 0$ and let

$$\mathcal{Z}_\epsilon := \{z \in \mathcal{Z} \ : \ z^\top \theta_* \geq z_*^\top \theta_* - \epsilon, \ z^\top \mu_{*,i} \leq \gamma + \epsilon, \forall i \in [m]\}.$$

Then our goal is to obtain an $(\epsilon, \delta)$-PAC learner defined as follows:

**Definition 1** (($\epsilon, \delta$)-PAC Learner). *A learner is $(\epsilon, \delta)$-PAC if for any instance it returns $\widehat{z}_\tau$ such that* $\mathbb{P}[\widehat{z}_\tau \in \mathcal{Z}_\epsilon] \geq 1 - \delta$.

We define the *optimality gap* for any $z \in \mathcal{Z}$ as $\Delta(z) := \theta_*^\top(z_* - z)$, and the *safety gap* for constraint $i$ as $\Delta_{\text{safe}}^i(z) := \gamma - \mu_{*,i}^\top z$. Note that either $\Delta(z)$ or $\Delta_{\text{safe}}^i(z)$ can be negative. If $\Delta(z) < 0$, it follows that $z$ has larger *value*—$z^\top \theta_*$—than the best safe arm $z_*$, which implies it must be unsafe. If $\Delta_{\text{safe}}^i(z) < 0$ for some $i$, then arm $z$ is unsafe. We also define the $\epsilon$-*safe optimality gap* as:

$$\Delta^\epsilon(z) = \max_{z' \in \mathcal{Z}}(z' - z)^\top \theta_* \quad \text{s.t.} \quad \min_{i \in [m]} \Delta_{\text{safe}}^i(z) \geq \epsilon. \tag{1}$$

$\Delta^\epsilon(z)$ is then the gap in value between arm $z$ and the best arm with minimum safety gap at least $\epsilon$.

**Mathematical Notation.** Let $\|x\|_A^2 = x^\top A x$ and $\mathfrak{p}(x) := \max\{x, 0\}$. $\widetilde{\mathcal{O}}(\cdot)$ hides factors that are logarithmic in the arguments. $\lesssim$ denotes inequality up to constants. We denote the simplex as $\triangle_{\mathcal{X}} := \{\lambda \in \mathbb{R}_{\geq 0}^{|\mathcal{X}|} : \sum_{x \in \mathcal{X}} \lambda_x = 1\}$.

## 2 Safe Best-Arm Identification in Linear Bandits

### 2.1 Algorithm Definition

The main challenge in algorithm design for the safe best-arm identification problem is ensuring that we are efficiently balancing our exploration between refining our estimates of both the safety gaps, as well as the optimality gaps. Our approach is given in Algorithm 1, BESIDE.

BESIDE relies on a round-based adaptive experimental design approach. In each round BESIDE consists of three phases. In the first phase, it solves an experimental design over $\lambda_\ell \in \triangle_{\mathcal{X}}$, with the goal of refining our estimates of the safety gaps. It then takes $\tau_\ell$ samples from $\lambda_\ell$. In the second phase these samples are used to estimate the safety constraints, $\widehat{\mu}^{i,\ell}$, and the safety gaps of each arm, $\widehat{\Delta}_{\text{safe}}^{i,\ell}(z)$. Finally, in Phase 3, an additional experimental design is solved which now aims to refine our estimates of the optimality gaps, and the estimates of the optimality gaps $\widehat{\Delta}^\ell(z)$ for each $z \in \mathcal{Z}$ are then computed. We encapsulate Phase 3 in a subroutine, RAGE$^\epsilon$, which we outline in the following. We now carefully describe each phase—we begin with Phase 2 to explain how our estimator works.

---

**Algorithm 1 Be**st **S**afe **Arm Ide**ntification (BESIDE)

---
1: **input:** tolerance $\epsilon$, confidence $\delta$
2: $\iota_\epsilon \leftarrow \lceil \log(\frac{20}{\epsilon})\rceil$, $\widehat{\Delta}_{\text{safe}}^{i,0}(z) \leftarrow 0$, $\widehat{\Delta}^0(z) \leftarrow 0$ for all $z \in \mathcal{Z}$
3: **for** $\ell = 1, 2, \ldots, \iota_\epsilon$ **do**
4: $\quad \epsilon_\ell \leftarrow 20 \cdot 2^{-\ell}$
$\quad$ // Phase 1: Solve design to reduce uncertainty in safety constraints
5: $\quad$ Define
$$c_\ell(z) = \min_j |\widehat{\Delta}_{\text{safe}}^{j,\ell-1}(z)| + \max_j \mathfrak{p}(-\widehat{\Delta}_{\text{safe}}^{j,\ell-1}(z)) + \mathfrak{p}(\widehat{\Delta}^{\ell-1}(z))$$
6: $\quad$ Let $\tau_\ell$ be the minimal value of $\tau \in \mathbb{R}_+$ which is greater than $4\log\frac{4m|\mathcal{Z}|\ell^2}{\delta}$ such that the objective to the following is no greater than $\epsilon_\ell/100$, and $\lambda_\ell$ the corresponding optimal distribution
$$\inf_{\lambda \in \triangle_{\mathcal{X}}} \max_{z \in \mathcal{Z}} -\frac{1}{100}\left(c_\ell(z) + \epsilon_\ell\right) + \sqrt{\tau^{-1} \cdot \|z\|_{A(\lambda)^{-1}}^2 \cdot \log(\tfrac{4m|\mathcal{Z}|\ell^2}{\delta})}$$
7: $\quad$ Sample $x_t \sim \lambda_\ell$, collect $\tau_\ell$ observations $\{(x_t, r_t, s_{t,1}, \ldots, s_{t,m})\}_{t=1}^{\tau_\ell}$
$\quad$ // Phase 2: Estimate safety constraints
8: $\quad \{\widehat{\mu}^{i,\ell}\}_{i=1}^m \leftarrow \mathsf{RIPS}(\{(x_t, s_{t,i})\}_{t=1}^{\tau_\ell}, \mathcal{Z}, \frac{\delta}{2m\ell^2})$
9: $\quad \widehat{\Delta}_{\text{safe}}^{i,\ell}(z) \leftarrow \gamma - z^\top \widehat{\mu}^{i,\ell} + \|z\|_{A(\lambda_\ell)^{-1}}\sqrt{\tau_\ell^{-1}\log(\tfrac{4m|\mathcal{Z}|\ell^2}{\delta})}$
$\quad$ // Phase 3: Refine estimates of optimality gaps
10: $\quad \{\widehat{\Delta}^\ell(z)\}_{z \in \mathcal{Z}} \leftarrow \mathsf{RAGE}^\epsilon\Big(\mathcal{Z}, \mathcal{Y}_\ell, \epsilon_\ell, \frac{\delta}{4\ell^2}, \{\widehat{\Delta}_{\text{safe}}(z) \leftarrow \max_j \mathfrak{p}(-\widehat{\Delta}_{\text{safe}}^{j,\ell}(z))\}_{z \in \mathcal{Z}}\Big)$
$\quad$ // Perform final round of exploration to ensure we find $\epsilon$-good arm
11: $\mathcal{Y}_{\text{end}} \leftarrow \{z \in \mathcal{Z} : c_\ell(z) \lesssim \widehat{\Delta}_{\text{safe}}^{i,\ell}(z) + \epsilon\}$
12: $\{\widehat{\Delta}^{\text{end}}(z)\}_{z \in \mathcal{Y}_{\text{end}}} \leftarrow \mathsf{RAGE}^\epsilon(\mathcal{Y}_{\text{end}}, \mathcal{Y}_{\text{end}}, \epsilon, \delta, \{\widehat{\Delta}_{\text{safe}}(z) \leftarrow \max_j \mathfrak{p}(-\widehat{\Delta}_{\text{safe}}^{j,\ell}(z))\}_{z \in \mathcal{Z}})$
13: **return** $\widehat{z} = \arg\min_{z \in \mathcal{Y}_{\text{end}}} \widehat{\Delta}^{\text{end}}(z)$

---

**Phase 2:** In Phase 2 the algorithm would like to use the $\tau_\ell$ samples drawn from the design $\lambda_\ell$ to estimate the constraints for each $z \in \mathcal{Z}$: $z^\top \mu_{*,i}$ for each $i \in [m]$. Past works using adaptive experimental design in the linear bandits literature have utilized the least-squares estimator along with complicated rounding schemes [13] which may require an additional $\text{poly}(d)$ samples each round (this $\text{poly}(d)$ factor could be prohibitively large—for example, in active classification problems, $d$ is the total number of data points). We instead utilize the $\mathsf{RIPS}$ estimator of [6] which gives us a guarantee of the form: with probability greater than $1 - \delta$, for all $z \in \mathcal{Z}$,

$$|z^\top(\widehat{\mu}^{i,\ell} - \mu_{*,i})| \lesssim \|z\|_{A(\lambda_\ell)^{-1}} \cdot \sqrt{\tau_\ell^{-1}\log(\tfrac{4m|\mathcal{Z}|\ell^2}{\delta})}. \tag{2}$$

We describe the $\mathsf{RIPS}$ estimator in more detail in Appendix B.

**Phase 1:** By our definition of the experimental design on Line 6, our safety gap estimation error bound in (2) satisfies, for each $z \in \mathcal{Z}$:

$$|z^\top(\widehat{\mu}^{i,\ell} - \mu_{*,i})| \lesssim \|z\|_{A(\lambda_\ell)^{-1}} \cdot \sqrt{\tau_\ell^{-1}\log(\tfrac{4m|\mathcal{Z}|\ell^2}{\delta})} \lesssim c_\ell(z) + \epsilon_\ell. \tag{3}$$

Note that our design chooses an allocation that minimizes the variance in our estimate of each safety constraint (up to some tolerance), which scales as $\|z\|_{A(\lambda)^{-1}}^2$. This can be thought of as a form of $\mathcal{X}\mathcal{Y}$-*design*—a design of the form $\inf_{\lambda \in \triangle_{\mathcal{X}}} \max_{y \in \mathcal{Y}} \|y\|_{A(\lambda)^{-1}}^2$—where here $\mathcal{Y} \leftarrow \mathcal{Z}$ is chosen to reduce our uncertainty in estimating the safety value for each $z \in \mathcal{Z}$. We refer to such a design objective henceforth as $\mathcal{X}\mathcal{Y}_{\text{safe}}$. Assume that at round $\ell - 1$, we can guarantee

$$c_\ell(z) = \min_j |\widehat{\Delta}_{\text{safe}}^{j,\ell-1}(z)| + \max_j \mathfrak{p}(-\widehat{\Delta}_{\text{safe}}^{j,\ell-1}(z)) + \mathfrak{p}(\widehat{\Delta}^{\ell-1}(z)) + \epsilon_\ell$$

$$\lesssim \min_j |\Delta_{\text{safe}}^j(z)| + \max_j \mathfrak{p}(-\Delta_{\text{safe}}^j(z)) + \mathfrak{p}(\Delta^{\epsilon_{\ell-1}}(z)) + \epsilon_\ell. \tag{4}$$

Then combining the above inequalities, we see that the experiment design on Line 6 aims to minimize the uncertainty in our estimate of $z^\top \mu_{*,i}$ up to a tolerance that scales as the maximum of the four

terms in (4). It follows that if any of these terms is large, we will only allocate a small number of samples to refining our estimate of arm $z$. Each one of these terms can be intuitively motivated by thinking through what is needed to prove that an arm $z \neq z_*$.

- $z$ **has small safety gap** $\min_j |\Delta_{\text{safe}}^j(z)|$: if this term is large, it implies that minimum safety gap for $z$ is large. To show an arm is safe or unsafe, it suffices to learn each safety gap up to a tolerance a constant factor from its value—regularizing by this term ensures we do just that.

- $z$ **fails some safety constraint** $\max_j \mathfrak{p}(-\Delta_{\text{safe}}^j(z))$: if this term is large, it implies that arm $z$ is very unsafe for some constraint. In this case, we can easily determine $z$ is unsafe, and therefore do not need to reduce our uncertainty in the safety gap any more.

- $z$ **is sub-optimal** $\mathfrak{p}(\Delta^{\epsilon_{\ell-1}}(z))$: if this term is large, it implies that $z$ is very suboptimal compared to some safe arm with safety gap at least $\epsilon_{\ell-1}$. In this case, we do not need to estimate $z$'s safety gap, as we will have already eliminated it.

It remains to ensure that (4) holds. As we show in Appendix D through a careful inductive argument, combining (3) with our guarantee on the estimates of the optimality gaps obtained in Phase 3, $\widehat{\Delta}^\ell(z)$, is sufficient to guarantee (4) holds. In particular, if any gap is greater than $\epsilon_\ell$ it is estimated up to a constant factor, and otherwise it is estimated up to $\mathcal{O}(\epsilon_\ell)$. This ensures that our gaps are estimated at the correct rate while guaranteeing we do not collect too many samples in each round.

**Phase 3:** In this phase we estimate the suboptimality gaps using RAGE$^\epsilon$. RAGE$^\epsilon$ is inspired by the RAGE algorithm of [13] for best-arm identification. In the interest of space, we defer the full definition of RAGE$^\epsilon$ to Appendix C but provide some intuition here. After Phase 2, by (3) the set of arms $\mathcal{Y}_\ell := \{z \in \mathcal{Z} : c_s(z) \lesssim \widehat{\Delta}^{i,s}(z), \forall i \in [m]\}$ for $s \leq \ell$ are precisely the ones that we can certify are safe (note that we do not need to ever explicitly construct such a set—we can instead maintain an implicit definition through the constraints). RAGE$^\epsilon$ uses an adaptive experimental design procedure to sample in such a way as to optimally estimate the gaps $(z - \widehat{y})^\top \theta_*, \forall z \in \mathcal{Z}$ and some $\widehat{y} \in \mathcal{Y}_\ell$ up to some (sufficient) tolerance. In particular, it also solves an $\mathcal{X}\mathcal{Y}$-design, but now on the set $\mathcal{Y} \leftarrow \{z - \widehat{y} : z \in \mathcal{Z}\}$. Thus, rather than minimizing $\|z\|_{A(\lambda)^{-1}}^2$, we minimize $\|z - \widehat{y}\|_{A(\lambda)^{-1}}^2$. This design reduces uncertainty on the *differences* between arms, which allows us to refine our estimates of their optimality gaps. Henceforth we refer to such a design as $\mathcal{X}\mathcal{Y}_{\text{diff}}$. We describe the importance of the choice of design in more detail in Section 2.4. Ultimately, if an arm $z$ has value within a factor of $\epsilon_\ell$ of the best safe arm in $\mathcal{Y}_\ell$, and if we have not yet shown arm $z$ is unsafe, then we will estimate its optimality gap up to a constant factor of $\epsilon_\ell$. If we were maintaining arm sets explicitly (similar to the original RAGE algorithm of [13]) we would eliminate arms at this point.

**Remark 1** (Computational Complexity). *The main computational challenge in* BESIDE *and* RAGE$^\epsilon$ *is the calculation of the experimental designs (i.e. Line 6 and the corresponding design in* RAGE$^\epsilon$). *In general, the presence of the square root implies that the resulting optimization problem may not be convex in* $\lambda$. *To handle this issue we note that* $2\sqrt{xy} = \min_{\alpha > 0} \alpha x + \frac{y}{\alpha}$—*thus we can replace the existing design with* $\inf_{\lambda \in \triangle_\mathcal{X}} \max_{z \in \mathcal{Z}} \min_{\alpha > 0} -\frac{1}{100}(c_\ell(z) + \epsilon_\ell) + \alpha \|z\|_{A(\lambda)^{-1}}^2 + \log(\frac{4m|\mathcal{Z}|\ell^2}{\delta})/(\alpha\tau)$. *By appropriately discretizing the space we search over for* $\tau$ *and* $\alpha$ *we can then apply the Frank-Wolfe algorithm to minimize over* $\lambda$. *While computationally efficient in theory, this procedure is quite complicated and impractical for large problems. In the experiments section we provide a practical heuristic that is motivated by the above algorithm and is computationally efficient for larger problems.*

### 2.2 Main Result

BESIDE achieves the following complexity.

**Theorem 1.** BESIDE *is* $(\epsilon, \delta)$-*PAC. In other words, with probability at least* $1 - \delta$, BESIDE *returns an arm* $\widehat{z} \in \mathcal{Z}$ *such that*

$$\widehat{z}^\top \theta_* \geq z_*^\top \theta_* - \epsilon, \quad \min_{i \in [m]} \Delta_{\text{safe}}^i(\widehat{z}) \geq -\epsilon$$

*and terminates after collecting at most*

$$C \cdot \sup_{\widetilde{\epsilon} \geq \epsilon} \inf_{\lambda \in \triangle_\mathcal{X}} \max_{z \in \mathcal{Z}} \frac{\|z\|_{A(\lambda)^{-1}}^2 \cdot \log(\frac{m|\mathcal{Z}|}{\delta})}{\left(\min_j |\Delta_{\text{safe}}^j(z)| + \max_j \mathfrak{p}(-\Delta_{\text{safe}}^j(z)) + \mathfrak{p}(\Delta^{\widetilde{\epsilon}}(z)) + \widehat{\epsilon}\right)^2} \qquad \text{(safety)}$$

$$+ C \cdot \sup_{\widehat{\epsilon} \geq \epsilon} \inf_{\lambda \in \triangle_{\mathcal{X}}} \max_{z \in \mathcal{Z}} \frac{\|z - z_*\|^2_{A(\lambda)^{-1}} \cdot \log(\frac{|\mathcal{Z}|}{\delta})}{\left( \max_j \mathfrak{p}(-\Delta^j_{\text{safe}}(z)) + \mathfrak{p}(\Delta^{\widetilde{\epsilon}}(z)) + \widehat{\epsilon} \right)^2} + C_0 \qquad \text{(optimality)}$$

samples for some $C = \text{poly} \log(\frac{1}{\epsilon})$ and $C_0 = \text{poly} \log(\frac{1}{\epsilon}, |\mathcal{Z}|) \cdot \log \frac{1}{\delta}$.

The complexity bound given in Theorem 1 may, at first glance, appear rather opaque, yet it in fact yields a very intuitive interpretation. The first term in the complexity, the safety term, is the complexity needed to show each arm is safe or unsafe, *if they have not otherwise been eliminated*. As described in the previous section, if $\mathfrak{p}(\Delta^{\widetilde{\epsilon}}(z))$ is large, this implies we have found an arm better than $z$, so learning its safety value is irrelevant.

The second term in the complexity, the optimality term, corresponds to the difficulty of showing an arm is worse *than the best arm we can guarantee is safe*. Note that we can only guarantee an arm is suboptimal if we can find a safe arm with higher value. Recall the definition of $\Delta^{\widetilde{\epsilon}}(z)$ given in (1). Intuitively, $\Delta^{\widetilde{\epsilon}}(z)$ denotes the gap in value between arm $z$ and the best arm with safety gap at least $\widetilde{\epsilon}$. As we make $\widetilde{\epsilon}$ smaller, we can show additional arms are safe, which increases $\Delta^{\widetilde{\epsilon}}(z)$. While this makes it easier to show $z$ is suboptimal, it comes at a cost—the extra samples necessary to decrease our safety tolerance, given by the first term in the complexity. BESIDE trades off between optimizing for each of these terms—gradually decreasing its tolerance on both the safety and optimality terms to more easily eliminate suboptimal arms, while not allocating too many samples to guarantee safety.

To help illustrate this complexity, we consider a simple example with orthogonal arms, i.e. a multi-armed bandit example.

**Example 1** (BESIDE on Multi-Armed Bandits)**.** *In the multi-armed bandit setting, we have $\mathcal{X} = \mathcal{Z} = \{e_1, \ldots, e_d\}$. Let $m = 1$, $d = 3$, and consider the settings of $\theta_*$ and $\mu_*$ given in Figure 1. Here we see that arm $e_1$ is safe and has value much higher than any other arm, so $z_* = e_1$, and can be shown to be safe relatively easily; arm $e_2$ has near-optimal value but is very unsafe; and arm $e_3$ is unsafe with very small safety gap, but has the smallest value.*

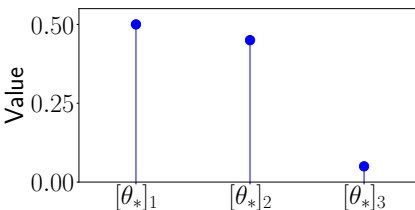 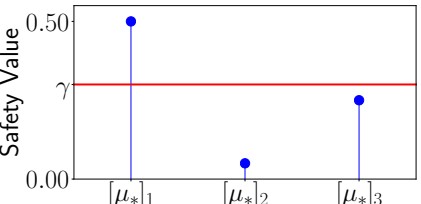

Figure 1: Multi-Armed Bandit Instance

**Showing $e_2$ is Suboptimal.** *As $e_2$ has near-optimal value, $\Delta(e_2)$ is very small and it is very difficult to show $e_2$ is suboptimal. However, $-\Delta_{\text{safe}}(e_2) = \mathcal{O}(1)$, so it is very easy to show $e_2$ is unsafe. It follows that $\mathfrak{p}(-\Delta_{\text{safe}}(e_2)) = \mathcal{O}(1)$ so both denominators in our complexity will always be $\mathcal{O}(1)$ for $z = e_2$—BESIDE does not attempt to show $e_2$ is suboptimal, but instead shows it is unsafe, and therefore does not pay for the small optimality gap of $\Delta(e_2)$ in the complexity.*

**Showing $e_3$ is Suboptimal.** *Recall the definition of $\Delta^{\epsilon}(z) = \max_{z': \Delta_{\text{safe}}(z') \geq \epsilon} \theta_*^\top (z' - z)$. In this case, for $\epsilon = \mathcal{O}(1)$, we will have $\Delta_{\text{safe}}(e_1) \geq \epsilon$, which implies that $\Delta^{\epsilon}(e_3) = \theta_*^\top (e_1 - e_2) = \Delta(e_3) = \mathcal{O}(1)$. To show $e_3$ is suboptimal, we could either show it is unsafe (which is very difficult) or suboptimal (which is very easy). Observing the sample complexity of Theorem 1 we see that the denominator of both terms will always be $\mathcal{O}(1)$ for $z = e_3$ since $\Delta^{\epsilon}(e_2) = \mathcal{O}(1)$—BESIDE never pays for the small safety gap of $e_3$, it instead takes advantage of the fact that $e_3$ can easily be shown to be suboptimal, and uses this to eliminate it.*

*In both of these cases we see that BESIDE does the "right" thing, always using the easier of the two criteria—either showing an arm is unsafe or suboptimal—to show that $z \neq z_*$. Combining the above observations, for $\epsilon \approx \min\{\Delta(e_3), -\Delta_{\text{safe}}(e_2), \Delta_{\text{safe}}(e_1)\}$, it follows that on this example the total*

*sample complexity of* BESIDE *given by Theorem 1 scales as:*

$$\widetilde{\mathcal{O}}\left(\left(\frac{1}{\Delta_{\text{safe}}(e_1)^2} + \frac{1}{\Delta_{\text{safe}}(e_2)^2} + \frac{1}{\Delta(e_3)^2}\right) \cdot \log\frac{1}{\delta}\right)$$

*where the $1/\Delta_{\text{safe}}(e_1)^2$ arises because we must also show $e_1$ is safe.*

### 2.3 Optimality of BESIDE

**Optimality in Best-Arm Identification.** Consider applying BESIDE to a problem instance where $m = 1$, $\mu_{*,1} = 0$, and $\gamma = 1$. In this case, every arm is safe, and the safety constraints are essentially vacuous—every arm can easily be shown safe. We can therefore think of this as simply an instance of the best-arm identification problem. In this setting, we obtain the following corollary.

**Corollary 1.** *Consider running* BESIDE *on a problem instance where $m = 1$, $\mu_{*,1} = 0$, and $\gamma = 1$, and set $\epsilon = \frac{1}{2}\max_{z \neq z_*} \theta_*^\top(z_* - z)$. Then with probability at least $1 - \delta$,* BESIDE *returns $z_*$ and has sample complexity bounded by:*

$$\widetilde{\mathcal{O}}\left(\inf_{\lambda \in \triangle_{\mathcal{X}}} \max_{z \in \mathcal{Z}} \frac{\|z - z_*\|^2_{A(\lambda)^{-1}}}{\Delta(z)^2} \cdot \log\frac{|\mathcal{Z}|}{\delta} + \inf_{\lambda \in \triangle_{\mathcal{X}}} \max_{z \in \mathcal{Z}} \|z\|^2_{A(\lambda)^{-1}} \cdot \log\frac{|\mathcal{Z}|}{\delta}\right).$$

Up to lower-order terms, this exactly matches the lower bound on best-arm identification given in [13]. Thus, in settings where the safety constraint is vacuous, BESIDE hits the optimal rate.

**Worst-Case Performance of BESIDE.** We next consider the worst-case performance of BESIDE in settings when $\mathcal{X} = \mathcal{Z}$. We have the following result.

**Corollary 2.** *Assume that $\mathcal{X} = \mathcal{Z}$. Then for any $\theta_*$ and $(\mu_{*,i})_{i=1}^m$, the sample complexity of* BESIDE *necessary to return an $\epsilon$-good and $\epsilon$-safe arm is bounded as $\widetilde{\mathcal{O}}(\frac{d}{\epsilon^2} \cdot (\log(m|\mathcal{X}|) + \log\frac{1}{\delta}))$.*

Theorem 2 of [38] shows a worst-case lower bound of $\Omega(d^2/\epsilon^2)$ on the sample complexity of identifying an $\epsilon$-optimal arm in the standard linear bandit setting. Safe best-arm identification problems in which the safety constraint is vacuous are at least as hard as the standard best-arm identification problem, since at minimum we need to find the best arm out of every safe arm. Thus, $\Omega(d^2/\epsilon^2)$ is also a worst-case lower bound for the safe best-arm identification problem. The hard instance of [38] has $|\mathcal{X}| = \mathcal{O}(2^d)$, so it follows that on this instance, BESIDE achieves a complexity of $\widetilde{\mathcal{O}}(\frac{d}{\epsilon^2} \cdot (d + \log\frac{1}{\delta}))$, and therefore BESIDE has optimal dimensionality dependence. In addition, this also implies that safe best-arm identification, in the worst-case, is no harder than the standard best-arm identification problem—it is no harder to find the best *safe* arm, regardless of the number of safety constraints, than to find the best arm, ignoring safety constraints.

### 2.4 The Role of Experiment Design

We can think of the safe best-arm identification problem, in some sense, as an interpolation of the standard best-arm identification problem, as well as the level-set estimation problem, where the goal is to identify $z \in \mathcal{Z}$ satisfying $z^\top \mu_* \leq \gamma$ [29]. In the former problem, [13] shows that the instance-optimal rate can be attained by running a round-based algorithm and at every round solving an instance of the $\mathcal{X}\mathcal{Y}_{\text{diff}}$ experiment design, as defined in Section 2.1. In the latter problem, [29] also show that a round-based algorithm can hit the instance-optimal rate, but instead solving the $\mathcal{X}\mathcal{Y}_{\text{safe}}$ problem at each round. It is natural to ask whether either of these strategies could be applied to the safe best-arm identification problem directly, or if it is necessary to alternate between them. The following results show that, on their own, each of these designs is unable to hit the optimal rate.

**Proposition 2.** *Fix some small enough $\epsilon > 0$. Then there exist instances of the safe best-arm identification problem, $\mathcal{I}_i = (\theta_*^i, \mu_*^i, \mathcal{X}^i, \mathcal{Z}^i)$, $i = 1, 2$, with $d = |\mathcal{X}^i| = |\mathcal{Z}^i| = 2$, $m = 1$, such that:*

- *On $\mathcal{I}^1$, any $(\epsilon, \delta)$-PAC algorithm which plays only allocations minimizing $\mathcal{X}\mathcal{Y}_{\text{diff}}$ must have $\mathbb{E}[\tau_\delta] \geq \Omega\left(\frac{1}{\epsilon^3} \cdot \log\frac{1}{\delta}\right)$, while* BESIDE *identifies an $\epsilon$-optimal arm after $\widetilde{\mathcal{O}}(\frac{1}{\epsilon^2} \cdot \log 1/\delta)$ samples.*

- *On $\mathcal{I}^2$, any $(\epsilon, \delta)$-PAC algorithm which plays only allocations minimizing $\mathcal{X}\mathcal{Y}_{\text{safe}}$ must have $\mathbb{E}[\tau_\delta] \geq \Omega\left(\frac{1}{\epsilon^{3/2}} \cdot \log\frac{1}{\delta}\right)$, while* BESIDE *identifies an $\epsilon$-optimal arm after $\widetilde{\mathcal{O}}(\frac{1}{\epsilon} \cdot \log 1/\delta)$ samples.*

Proposition 2 implies that, to solve the safe best-arm identification problem optimally, more care must be taken in exploring than either standard experiment design induces—we must trade off between $\mathcal{XY}_{\text{diff}}$ and $\mathcal{XY}_{\text{safe}}$ as BESIDE does. We remark briefly on the instance $\mathcal{I}^1$. On this instance we have $\mathcal{X} = \{e_1, e_2\}$ and $\mathcal{Z} = \{z_1, z_2\}$ with $z_1 = [1/4, 1/2]$ and $z_2 = [3/4, 1/2 + \alpha]$. We set $\theta_*^1 = [1, 0]$, $\mu_*^1 = [0, 1]$, and $\gamma = 1/2 + \alpha/2$. Here $z_2$ is unsafe while $z_1$ is safe, so it follows that $z_* = z_1$. As $z_2^\top \theta_*^1 > z_1^\top \theta_*^1$, to show $z_2 \neq z_*$, we must show it is unsafe. However, if we solve the design $\mathcal{XY}_{\text{diff}}$, we see that it places nearly all of the mass on the first coordinate. While this would be optimal if both $z_1$ and $z_2$ were safe and we simply wished to determine which has a higher value, to show $z_2$ is unsafe, the optimal strategy places (roughly) the same mass on each coordinate, since each coordinate could contribute to the safety value. This is precisely the allocation BESIDE will play, so it is able to show that $z_2$ is unsafe much more efficiently than a naive $\mathcal{XY}_{\text{diff}}$ approach.

## 3 Experiments for Safe Best Arm Identification in Linear Bandits

We next present experimental results on BESIDE to demonstrate the advantage of experimental design—especially combining $\mathcal{XY}_{\text{diff}}$ and $\mathcal{XY}_{\text{safe}}$ designs. As there are no existing algorithms that consider safe best-arm identification, as a benchmark we consider the naive adaptive approach BASELINE that first solves the problem of finding the safe arms up to a desired tolerance, and then solves the problem of finding the best (safe) arm among the arms that were found to be safe. We first describe instances on which we test BESIDE. Our experimental details and precise implementation of BESIDE using elimination are described in Section F.

**Multi-Armed Bandit.** We consider a best-arm identification problem in which every arm is safe, but the arm with highest value is very difficult to identify as safe, while the second-best arm can easily be shown safe. We vary the total number of arms and run BESIDE and BASELINE with $\epsilon = 0.5$ and $\delta = 0.1$. From Figure 2, we observe that the sample complexity of BESIDE is smaller (up to about two times for 100 arms) than the sample complexity of its baseline.

**Linear Response Model.** *Random Instance*: We also consider the more general setup where $\mathcal{X}, \mathcal{Z} \subset \mathbb{R}^d$, $\theta \in \mathbb{R}^d$ and $\mu \in \mathbb{R}^d$ are randomly generated from independent Gaussian random variables with mean 0 and variance 1. We set $|\mathcal{X}| = 50$ and vary the size of $|\mathcal{Z}|$. In Figure 3, we see again that BESIDE significantly outperforms the baseline.

*Hard Instance*: We last consider the instance of Proposition 2 and benchmark against the strategy playing only allocations minimizing $\mathcal{XY}_{\text{diff}}$. In Figure 4, we see again that BESIDE significantly outperforms this baseline, corroborating the theoretical result of Proposition 2.

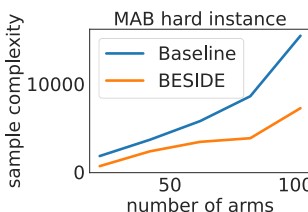

Figure 2: Total arm pulls to termination vs. number of arms

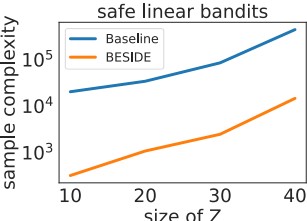

Figure 3: Total arm pulls to termination vs. $|\mathcal{Z}|$

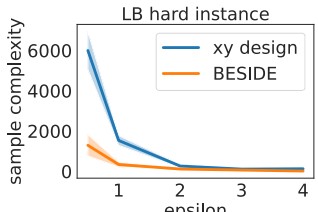

Figure 4: Total arm pulls to termination vs. $\epsilon$

### 3.1 Practical Algorithms for Active Classification Under Constraints

Next, we provide an application of the above ideas to pool-based active classification with constraints—namely, adaptive sampling to learn the highest accuracy classifier with a constraint on the false discovery rate (FDR). We first explain how this problem maps to the linear bandit setting. Precisely, let $\mathcal{X}$ be the example space and $\mathcal{Y} = \{0, 1\}$ the label space. Fix a hypothesis class $\mathcal{H}$ such that each $h \in \mathcal{H}$ is a classifier $h : \mathcal{X} \to \mathcal{Y}$. We represent each $h$ with an associated indicator vector $z_h \in \{0, 1\}^{|\mathcal{X}|}$ where $z_h(x) = 1 \iff h(x) = 1$. Similarly, let $\eta \in [0, 1]^{|\mathcal{X}|}$ represent the label distribution, i.e. $\eta(x) = \mathbb{P}(Y = 1 | X = x)$. Then the risk of a classifier $R(h) := \mathbb{E}_{x \sim \text{Unif}(\mathcal{X}), Y \sim \text{Ber}(\eta(x))}[\mathbb{1}[h(x) \neq Y]] = z_h^\top(2\eta - 1)$ and the FDR is defined as $\text{FDR}(h) := (\mathbf{1} - \eta)^\top z / \mathbf{1}^\top z$. In the case when $\eta \in \{0, 1\}^{|\mathcal{X}|}$, $\text{FDR}(h)$ is the proportion of examples that $h$ incorrectly labels as 1 out of all

examples $h$ labels as 1. Our goal is to solve the following constrained best arm identification problem:

$$\widehat{h} = \min_{h \in \mathcal{H}} R(h) \quad \text{s.t.} \quad \text{FDR}(h) \leq q \iff \min_{h \in \mathcal{H}} z_h^\top \eta \quad \text{s.t.} \quad ((\mathbf{1} - \eta)^\top - q\mathbf{1}^\top)^\top z \leq 0. \quad (5)$$

The main challenge in running BESIDE on this problem directly is a potentially high computational cost from computing a design over an extremely large hypothesis class $\mathcal{H}$ (e.g. neural networks of a bounded width). In this section we provide an alternative approach motivated by BESIDE. Algorithm 2 follows a similar design as BESIDE and relies on an oracle, CERM, that can solve (5), i.e. given a dataset it returns the highest accuracy classifier under an FDR constraint. Such oracles are available in, for example in [1, 10]. In each round of Algorithm 2 we perform *randomized exploration* by perturbing the labels on our existing dataset with mean zero Gaussian noise, and then training $k$ classifiers $\widehat{h}_i, i \in [k]$, on the resulting datasets. Implicitly, we are making the assumption that the loss function in the training of ERM can handle continuous labels, such as the MLE of logistic regression. As described in [25], randomized exploration emulates sampling from a posterior distribution on our possible set of classifiers. We then use the labels generated from these classifiers to compute safe classifiers $h_i, i \in [k]$. Finally, mimicking the strategy of BESIDE, we compute $\mathcal{X}\mathcal{Y}_{\text{safe}}$ and $\mathcal{X}\mathcal{Y}_{\text{diff}}$ designs on these $k$ safe classifiers and repeat (note that the designs computed on Line 5 are equivalent to $\mathcal{X}\mathcal{Y}_{\text{safe}}$ and $\mathcal{X}\mathcal{Y}_{\text{diff}}$ in the classification setting).

---

**Algorithm 2** `Active constrained classification with randomized exploration`

---

**Require:** Batch size $n$, initial (labeled) data $x_1^{(0)}, \ldots, x_n^{(0)}$, number of rounds $L$, number of classifiers per round $k$, perturbation variance $\sigma$

1: **for** $\ell = 1, \ldots, L$ **do**
2:     **for** $i = 1, \ldots, k$ **do**
3:         $\widehat{h}_i = \text{ERM}(\{(x_t^{(\ell)}, y_t^{(\ell)} + \epsilon_t^{(i)})\}_{t=1}^n)$, where $\{\epsilon_t^{(i)}\}_{1 \leq t \leq n} \overset{i.i.d.}{\sim} \mathcal{N}(0, \sigma^2)$
4:         $h_i = \text{CERM}(\{(x, \widehat{h}_i(x))\}_{x \in \mathcal{X}})$
5:     Compute designs: $\lambda_{\text{safe}} = \arg\min_{\lambda \in \triangle_{\mathcal{X}}} \max_{1 \leq i \leq k} \sum_{x \in \mathcal{X}} \frac{\mathbb{1}\{h_i(x) \neq 0\}}{\lambda_x}$, $\lambda_{\text{diff}} = \arg\min_{\lambda \in \triangle_{\mathcal{X}}} \max_{1 \leq i \neq j \leq k} \sum_{x \in \mathcal{X}} \frac{\mathbb{1}\{h_i(x) \neq h_j(x)\}}{\lambda_x}$
6:     Sample $x_1^{(\ell)}, \ldots, x_n^{(\ell)}$ from a uniform mixture of $\lambda_{\text{safe}}, \lambda_{\text{diff}}$
7:     Observe corresponding labels $y_1^{(\ell)}, \ldots, y_n^{(\ell)}$
    **return** $\widetilde{h} = \text{CERM}(\{(x_t^{(\ell)}, y_t^{(\ell)})\}_{1 \leq t \leq n, 0 \leq \ell \leq L})$

---

To validate Algorithm 2, we experiment against a passive baseline that selects points uniformly at randoms from the pool of examples $\mathcal{X}$, retrains the model using the same Constrained Empirical Risk Minimization oracle (CERM) as Algorithm 2 on its current samples, and report the accuracy and FDR. We evaluate on two real world datasets and on one synthetic dataset next and provide an additional details on the experiments in Section F.

**Adult dataset.** We evaluate on the adult income data set [27] (48,842 examples) where the goal is to predict whether someone's income is above \$50k per year. We set the constraint to be FDR $< 0.15$ and report in Figure 5 the accuracy and the FDR obtained when varying the number of labels given to each method (batch size is set to 25 and initial number of queried labels is 50). We observe that for any desired accuracy Algorithm 2 allows us to provide a classifier with lower FDR. Also, for any chosen number of total labels—such as 500, 750, 2000 as reported in Figure 5—the Algorithm 2 gives a classifier with higher accuracy and lower FDR. In general we found that the active method needed half the number of samples as the passive sampling to achieve a given FDR. This demonstrates the effectiveness of Algorithm 2 to learn simultaneously the objective (risk) and the constraint (FDR), in a similar favorable way as characterized by our theoretical findings.

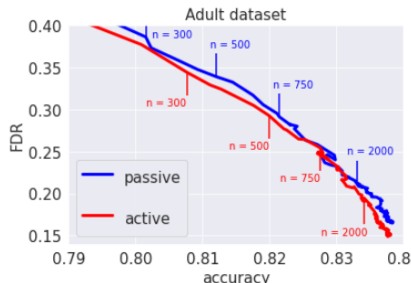

Figure 5: FDR vs accuracy for active (Algorithm 2) and passive sampling, ticks report number of samples. FDR and accuracy are averaged over 5 trials

**German Credit dataset.** We consider the German Credit Dataset originally from the Staflog Project Databases [24]. The goal is to predict whether someone's credit is 'bad' or 'good'. We report in Figure 6 the recall (TPR) and the precision $(1-\text{FDR})$ obtained when varying the number of labels given to each method. We observe that for any desired precision Algorithm 2 allows us to provide a classifier with higher recall. Also, for any chosen number of total labels—such as $170, 270, 330, 450, 600$ as reported in Figure 6—the Algorithm 2 gives a classifier with higher precision and higher recall. As for the Adult dataset we found that the active method needed half the number of samples as the passive sampling to achieve a given precision.

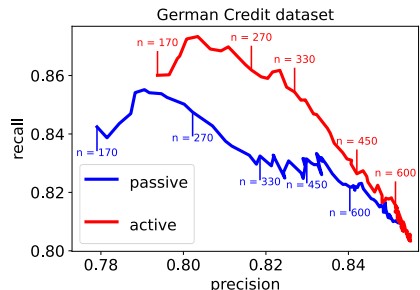

Figure 6: TPR vs FDR for active active (Algorithm 2) and passive sampling, ticks report number of samples. Precision is $1-$ FDR, recall is TPR. Precision and recall are averaged over 25 trials

**Half circle dataset.** We consider a two-dimensional half circle dataset, visualized on Figure 10. We report in Figures 11 and 12 the precision and (respectively) the recall obtained when varying the number of labels given to each method. The confidence intervals are obtained over 25 repetitions. We observe that Algorithm 2 allows us to provide a classifier satisfying a given recall or precision in far fewer queries. This is in line with the results of [16] on One Dimensional Thresholds, where the sample complexity of the active strategy is $O(log(n))$ while the sample complexity of the passive strategy is at least of order $n$.

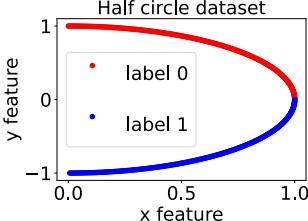

Figure 7: Half circle dataset.

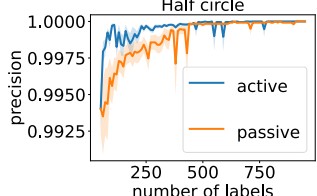

Figure 8: Precision

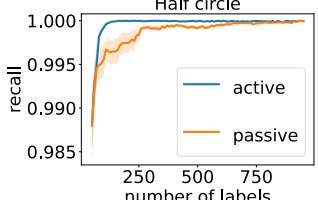

Figure 9: Recall

## 4 Related works

**Constrained Bandits.** A growing body of work seeks to address the question of safe learning in interactive environments. In particular, the majority of such works have considered the problem of regret minimization in linear bandits with linear safety constraints. Here, the goal is to maximize online reward, $x_t^\top \theta_*$, by choosing actions $x_t \in \mathcal{X} \subseteq \mathbb{R}^d$, while ensuring a safety constraint of the form $x_t^\top \mu_* \leq \gamma$ is met at all times (either in expectation or with high probability). A variety of algorithms have been proposed, including UCB-style [23, 2, 32], and Thompson Sampling [30, 31]. While these works show that $\sqrt{T}$ regret is attainable, they only provide worst-case bounds (while we obtain instance-dependent bounds) and do not study the pure-exploration best-arm identification problem. To our knowledge, the only work to offer instance-dependent guarantees is [9], yet they focus exclusively on the regret setting, and offer a relatively coarse notion of instance-dependence — analogous to $\mathcal{O}(d \cdot \text{poly} \log T / \Delta_{\min})$ bounds in the unconstrained linear bandits setting — in contrast to the more fine-grained notion of instance-dependence we provide.

To our knowledge, only several existing works consider the question of best-arm identification with safety constraints [36, 37, 39, 28]. The most related to ours is [28] which focuses on the easier problem of safe best arm identification with known rewards and unknown constraints. Since the reward is known, the main challenge in the setting of [28] is to learn constraints via G-optimal designs. The key and novel challenge of our framework is to carefully balance between G and $\mathcal{X}\mathcal{Y}$ designs: naively spending enough budget to either learn the reward model (via a $\mathcal{X}\mathcal{Y}$ design) or to learn the safety constraints (via a G design) will fail catastrophically (see Example 2.1 and Proposition 1). [36, 37] consider a general constrained optimization setting where the goal of the learner is to minimize some function $f(x)$ over a domain $x \in \mathcal{D}$, while only having access to noisy samples of $f(x)$, $f(x_t) + w_t$, and guaranteeing that a safety constraint $g(x_t) \geq h$ is met for every query point $x_t$.

While they do provide a sample complexity upper bound, they give no lower bound, and, as shown in [39], their approach can be very suboptimal. [39] considers the setting of best-arm identification in multi-armed bandits. In their setting, at every step $t$ they query a value $a_t \in \mathcal{A}$ for a particular coordinate $i_t$, and their goal is to identify the coordinate $i^*$ such that $a_{i^*}^* \theta_{i^*} \geq \max_i a_i^* \theta_i$, where $a_i^*$ is the largest value respecting the safety constraint: $a_i^* = \arg\max_{a \in \mathcal{A}} a\theta_i$ s.t. $a\mu_i \leq \gamma$. Similar to [36, 37], they require that the safety constraint $a_t \mu_{i_t} \leq \gamma$ must be met while learning. Though they do show matching upper and lower bounds, and in addition consider a slightly more general setting that allows for nonlinear (but monotonic) response functions, they treat every coordinate as independent, and do not allow for information-sharing between coordinates—the key generalization the linear bandit setting targets. We remark as well that in our setting, unlike these works, we allow the learner to query unsafe points during exploration, and only require that they output a safe decision at termination.

**Best-Arm Identification in Linear Bandits.** The best-arm identification problem in multi-armed bandits (without safety constraints) is a classical and well-studied problem [3, 33, 12, 5], and near-optimal algorithms exist [18, 22]. More recently, there has been a growing interest in understanding the sample complexity of best-arm identification in linear bandits [35, 19, 40, 13, 20, 11]. We highlight in particular the work of [13] which proposes an experiment-design based algorithm, RAGE, that our approach takes inspiration from. While much progress has been made in understanding best-arm identification in linear bandits, to our knowledge, no existing works consider the setting of best-arm identification in linear bandits with safety constraints, the setting of this work.

**Active Classification under FDR constraints** We finally mention one other related body of work— the problem of actively sampling to find a classifier with high accuracy or recall under precision constraints. Motivated by the experimental design approach of our main algorithm, BESIDE, we provide a heuristic algorithm for this problem with good empirical performance in Section 3.1. There is an extensive body of work on active learning (see the survey [14]) but only recently have works made the connection between best-arm identification for linear bandits and classification [21, 16, 7]. Precision constraints has been less studied in the adaptive context, we only know of [16, 4].

## 5 Conclusion

In this work we have shown that it is possible to efficiently find the best *safe* arm in linear bandits with a carefully designed adaptive experiment design-based approach. Our results open up several interesting directions for future work.

**Instance Optimality.** While BESIDE is worst-case optimal, in Appendix A we show an instance-dependent lower bound which BESIDE does not, in general, seem to hit. We conjecture that this lower bound may be loose—addressing this discrepancy and showing matching instance-dependent upper and lower bounds is an exciting direction for future work.

**Safety During Exploration.** Though there are many interesting applications where we may not require safety during exploration (i.e. only querying safe arms), in other cases we may need to ensure safety is met during exploration. Extending our work to this setting is an interesting open problem.

**Potential Impacts.** As with any algorithm making stochastic assumptions, if assumptions are not met we can not guarantee the performance. In this case, one limitation is that if the underlying environment is changing (i.e. the constraints vary over time) the algorithm could have unexpected behavior with unintended consequences. Such a situation could lead to harmful results in examples such as the online advertising bidding example from the introduction. To mitigate this limitation of our setting, practitioners are encouraged to monitor many metrics, both short and long-term.

### Acknowledgements

The work of AW was supported by an NSF GFRP Fellowship DGE-1762114. The work of JM was supported by an NSF Career award, and NSF AI institute (IFML) and the Simons collaborative grant on the foundations of fairness. The work of KJ was funded in part by the AFRL and NSF TRIPODS 2023166.

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
