# Contents

# A Lower Bounds

## A.1 Oracle Lower Bound

**Theorem 3** (Oracle Lower Bound). *Let $\tau$ denote the stopping time for any $(0,\delta)$-PAC algorithm for pure exploration in safe linear bandits. Then*

$$\frac{\mathbb{E}_{\theta_*,\mu_*}[\tau]}{\log\frac{1}{2.4\delta}} \geq \min_{\lambda\in\triangle_{\mathcal{X}}}\max\left\{\max_{z\in\mathcal{Z}\setminus z_*}\min\left\{\frac{\|z\|^2_{A(\lambda)^{-1}}}{\mathfrak{p}(-\Delta_{\text{safe}}(z))^2},\frac{\|z-z_*\|^2_{A(\lambda)^{-1}}}{\mathfrak{p}(\Delta(z))^2}\right\},\frac{\|z_*\|^2_{A(\lambda)^{-1}}}{(z_*^\top\mu_*-\alpha)^2}\right\}.$$

**Comparing Complexity with Theorem 1.** In the single-constraint setting, the complexity of BESIDE reduces to

$$C\cdot\sup_{\widetilde{\epsilon}\geq\epsilon}\inf_{\lambda\in\triangle_{\mathcal{X}}}\max_{z\in\mathcal{Z}}\frac{\|z\|^2_{A(\lambda)^{-1}}\cdot\log(\frac{m|\mathcal{Z}|}{\delta})}{\left(|\Delta_{\text{safe}}(z)|+\mathfrak{p}(\Delta^{\widetilde{\epsilon}}(z))+\widetilde{\epsilon}\right)^2}$$

$$+\,C\cdot\sup_{\widetilde{\epsilon}\geq\epsilon}\inf_{\lambda\in\triangle_{\mathcal{X}}}\max_{z\in\mathcal{Z}}\frac{\|z-z_*\|^2_{A(\lambda)^{-1}}\cdot\log(\frac{|\mathcal{Z}|}{\delta})}{\left(\mathfrak{p}(-\Delta_{\text{safe}}(z))+\mathfrak{p}(\Delta^{\widetilde{\epsilon}}(z))+\widetilde{\epsilon}\right)^2}+C_0$$

Consider the case when $\Delta^{\widetilde{\epsilon}}(z)$ is "smooth" in $\widetilde{\epsilon}$, in the sense that $\Delta^{\widetilde{\epsilon}}(z)\geq\Delta(z)-\widetilde{\epsilon}$. This condition corresponds to the case, for example, where $z_*$ has a large safety gap (in which case we simply have $\Delta^{\widetilde{\epsilon}}(z)=\Delta(z)$ for moderate values of $\widetilde{\epsilon}$), or where $z_*$ might have a small safety gap, but where there are arms placed at even intervals so that, as we let the safety gap get smaller, we are always able to find better arms. Under this assumption, the complexity can be upper bounded as

$$C\cdot\inf_{\lambda\in\triangle_{\mathcal{X}}}\max_{z\in\mathcal{Z}}\frac{\|z\|^2_{A(\lambda)^{-1}}\cdot\log(\frac{m|\mathcal{Z}|}{\delta})}{\left(|\Delta_{\text{safe}}(z)|+\mathfrak{p}(\Delta(z))\right)^2}+C\cdot\inf_{\lambda\in\triangle_{\mathcal{X}}}\max_{z\in\mathcal{Z}\setminus z_*}\frac{\|z-z_*\|^2_{A(\lambda)^{-1}}\cdot\log(\frac{|\mathcal{Z}|}{\delta})}{\left(\mathfrak{p}(-\Delta_{\text{safe}}(z))+\mathfrak{p}(\Delta(z))\right)^2}+C_0$$

$$\leq C\cdot\inf_{\lambda\in\triangle_{\mathcal{X}}}\max_{z\in\mathcal{Z}}\frac{\|z\|^2_{A(\lambda)^{-1}}\cdot\log(\frac{m|\mathcal{Z}|}{\delta})}{\left(|\Delta_{\text{safe}}(z)|+\mathfrak{p}(\Delta(z))\right)^2}+C\cdot\inf_{\lambda\in\triangle_{\mathcal{X}}}\max_{z\in\mathcal{Z}\setminus z_*}\frac{\|z-z_*\|^2_{A(\lambda)^{-1}}\cdot\log(\frac{|\mathcal{Z}|}{\delta})}{\left(\mathfrak{p}(-\Delta_{\text{safe}}(z))+\mathfrak{p}(\Delta(z))\right)^2}+C_0$$

which can be upper bounded as

$$C\log(\tfrac{m|\mathcal{Z}|}{\delta})\cdot\left(\inf_{\lambda\in\triangle_{\mathcal{X}}}\max_{z\in\mathcal{Z}}\frac{\|z\|^2_{A(\lambda)^{-1}}}{\max\{\Delta_{\text{safe}}(z)^2,\mathfrak{p}(\Delta(z))^2\}}+\inf_{\lambda\in\triangle_{\mathcal{X}}}\max_{z\in\mathcal{Z}\setminus z_*}\frac{\|z-z_*\|^2_{A(\lambda)^{-1}}}{\max\{\mathfrak{p}(-\Delta_{\text{safe}}(z))^2,\mathfrak{p}(\Delta(z))^2\}}\right)+C_0.$$

While this does not match the lower bound of Theorem 3 exactly, it scales in a similar manner. As in Theorem 3, we pay only for the larger of the optimality gap, $\mathfrak{p}(\Delta(z))$, and safety gap $\mathfrak{p}(-\Delta_{\text{safe}}(z))$ (if the arm is unsafe). The primary difference between Theorem 3 and this complexity are the terms in the numerator—in Theorem 3, the numerator scales as $\|z-z_*\|^2_{A(\lambda)^{-1}}$ only if an arm is easier to eliminate by showing it is suboptimal, while in our complexity it could scale this way in either case.

The primary difficulty in hitting the lower bound exactly is that Theorem 3 is a *verification* lower bound. It assumes knowledge of the best arm, and is told whether every other arm has smaller safety gap (if the arm is unsafe) or optimality gap. It can therefore simply use this knowledge to focus all samples on verifying an arm is either unsafe, or suboptimal.

In practice, we do not have access to such information. Without knowing whether it is easier to eliminate an arm by showing it is unsafe or suboptimal, the best we can hope to do is to seek to estimate both the safety value and reward value of every arm, until we have estimated one well enough to show the arm is suboptimal or unsafe.

We conjecture that the lower bound of Theorem 3 is loose, and that Theorem 1 is nearly optimal. We believe the gap arises because, as noted, lower bound proof techniques, such as those proposed in [22], which is what we rely on to prove Theorem 3, are lower bounding only the complexity of verifying the optimal solution. In problem settings such as ours where the *order* matters—where we will obtain a very different rate if we focus our attention on one arm versus another, to show it is safe or unsafe—such techniques appear insufficient to obtain a tight lower bound. Indeed, we conjecture that a "moderate-confidence" lower bound can be shown using techniques from [34], and that such a lower bound may have a complexity nearly matching that of Theorem 1. We leave proving this for future work.

*Proof of Theorem 3.* Following the proof of Theorem 1 of [13] and applying the Transportation Lemma of [22], we have that any $\delta$-PAC algorithm must satisfy

$$\sum_{x\in\mathcal{X}}\mathbb{E}[T_x] \geq \log\frac{1}{2.4\delta} \cdot \inf_{\lambda\in\triangle_{\mathcal{X}}} \frac{1}{\min_{(\theta,\mu)\in\mathcal{C}_{\mathrm{alt}}}\sum_{x\in\mathcal{X}}\lambda_x\mathrm{KL}(\nu_{(\theta_*,\mu_*),i}||\nu_{(\theta,\mu),i})}$$

there $T_x$ denotes the number of pulls to arm $x$, and $\mathcal{C}_{\mathrm{alt}}$ is the set of alternate instances defined in Lemma 4. As we assume that the noise is $\mathcal{N}(0,1)$, and since the noise is independent for the safety observations and reward observations, we have

$$\mathrm{KL}(\nu_{(\theta_*,\mu_*),i}||\nu_{(\theta,\mu),i}) = \frac{1}{2}(x_i^\top(\theta_*-\theta))^2 + \frac{1}{2}(x_i^\top(\mu_*-\mu))^2.$$

Some algebra shows that

$$\sum_{x\in\mathcal{X}}\lambda_x\mathrm{KL}(\nu_{(\theta_*,\mu_*),i}||\nu_{(\theta,\mu),i}) = \frac{1}{2}\|\theta_*-\theta\|_{A(\lambda)}^2 + \frac{1}{2}\|\mu_*-\mu\|_{A(\lambda)}^2.$$

The result then follows by applying Lemma 4 to compute

$$\min_{(\theta,\mu)\in\mathcal{C}_{\mathrm{alt}}} \frac{1}{2}\|\theta_*-\theta\|_{A(\lambda)}^2 + \frac{1}{2}\|\mu_*-\mu\|_{A(\lambda)}^2.$$

$\square$

**Lemma 4.** *Define the alternate set:*

$$\mathcal{C}_{\mathrm{alt}} = \{(\theta,\mu) \text{ s.t. } \mu^\top z^* > \alpha\} \cup \{(\theta,\mu) \text{ s.t. } \exists z' \neq z^*, \mu^\top z' \leq \alpha, \theta^\top(z^*-z') \leq 0\},$$

*Then the projection to the alternate is*

$$\min_{(\theta,\mu)\in\mathcal{C}_{\mathrm{alt}}} \|\theta-\theta_*\|_{A(\lambda)}^2 + \|\mu-\mu_*\|_{A(\lambda)}^2 = \min\left\{\min_{z\neq z^*} \frac{\mathfrak{p}(z^\top\mu_*-\alpha)^2}{\|z\|_{A(\lambda)^{-1}}^2} + \frac{\mathfrak{p}((z_*-z)^\top\theta_*)^2}{\|z-z_*\|_{A(\lambda)^{-1}}^2}, \frac{(z_*^\top\mu_*-\alpha)^2}{\|z_*\|_{A(\lambda)^{-1}}^2}\right\}.$$

*Proof.* For each arm $x$ the associated and we want to solve

$$\min_{(\theta,\mu)\in\mathcal{C}_{\mathrm{alt}}} \|\theta-\theta_*\|_{\sum_{x\in\mathcal{X}}\lambda_x xx^\top}^2 + \|\mu-\mu_*\|_{\sum_{x\in\mathcal{X}}\lambda_x xx^\top}^2.$$

To do so, we use that $\min_{x\in A\cup B} f(x) = \min_{S\in\{A,B\}}\min_{x\in S} f(x)$ on the quadratic objective by defining the sets

$$A := \{(\theta,\mu) \text{ s.t. } \mu^\top z^* > \alpha\}, \quad B = \{(\theta,\mu) \text{ s.t. } \exists z' \neq z^*, \mu^\top z' \leq \alpha, \theta^\top(z^*-z') \leq 0\},$$

such that their union is $A \cup B = \mathcal{C}_{\mathrm{alt}}$.

Note that we know from [29] that

$$\min_{(\theta,\mu)\in A} \|\theta-\theta_*\|_{\sum_{x\in\mathcal{X}}\lambda_x xx^\top}^2 + \|\mu-\mu_*\|_{\sum_{x\in\mathcal{X}}\lambda_x xx^\top}^2 = \frac{(z_*^\top\mu_*-\alpha)^2}{\|z_*\|_{A(\lambda)^{-1}}^2}.$$

We now lift $B$ to a set $\mathrm{lift}(B)$ that is defined as

$$\mathrm{lift}(B) = \{[\theta,\mu] \text{ s.t. } \exists z' \neq z^*, [\theta,\mu]^\top[(z^*-z'),0;0,z'] \leq [0,\alpha]\}.$$

Thus we can focus on $D_z = \{\kappa \in \mathbb{R}^{2n} \text{ s.t. } A_z\kappa \leq b\}$ where $A_z = [(z^*-z'),0;0,z'] \in R^{2\times 2n}$. Now we want to solve

$$\min_{z\in\mathcal{Z}\setminus\{z_*\}} \min_{\kappa\in\mathbb{R}^{2n} \text{ s.t. } A_z\kappa\leq b} \|\kappa-\kappa_*\|_\Gamma,$$

where $\Gamma = I_2 \otimes \left(\sum_{x\in\mathcal{X}}\lambda_x xx^\top\right)$.

**Lemma 5.** *The optimal solution of*

$$\min_{\kappa \in \mathbb{R}^{2n} \text{ s.t. } A\kappa \leq b} \frac{\|\kappa - \kappa_*\|_\Gamma}{2}$$

*is $\kappa_0 = \kappa_* - \Gamma^{-1}A^\top(A\Gamma^{-1}A^\top)^{-1}\{A\kappa_* - b\}_+$ and the optimal value is*

$$\frac{1}{2}\mathfrak{p}(A\kappa_* - b)^\top(A\Gamma^{-1}A^\top)^{-1}\mathfrak{p}(A\kappa_* - b),$$

*where $\mathfrak{p}(\cdot)$ is applied element-wise to $A\kappa_* - b$.*

This translate to

$$\min_{(\theta,\mu)\in B} \|\theta - \theta_*\|^2_{\sum_{x\in\mathcal{X}} \lambda_x xx^\top} + \|\mu - \mu_*\|^2_{\sum_{x\in\mathcal{X}} \lambda_x xx^\top} = \min_{z\neq z^*} \frac{\mathfrak{p}(z^\top\mu_* - \alpha)^2}{\|z\|^2_{A(\lambda)^{-1}}} + \frac{\mathfrak{p}((z_* - z)^\top\theta_*)^2}{\|z - z_*\|^2_{A(\lambda)^{-1}}},$$

and we get the desired result.

$\square$

*Proof of Lemma 5.* Consider the Lagrangian

$$\mathcal{L}(\kappa,\mu) = \frac{1}{2}(\kappa - \kappa_*)^\top\Gamma(\kappa - \kappa_*) + \mu^\top(A\kappa - b)$$

$$\mathcal{L}(\kappa',\mu) = \frac{1}{2}\kappa'^\top\Gamma\kappa' + \mu^\top(A\kappa' - b + A\kappa_*)$$

minimized at $\kappa'_0 = -\Gamma^{-1}A^\top\mu$. We have

$$\max_{\mu\geq 0}\min_{\kappa'}\mathcal{L}(\kappa',\mu) = \max_{\mu\geq 0}\frac{1}{2}(\Gamma^{-1}A^\top\mu)^\top\Gamma(\Gamma^{-1}A^\top\mu) + \mu^\top(-A\Gamma^{-1}A^\top\mu - b + A\kappa_*)$$

$$= \max_{\mu\geq 0} -\frac{1}{2}\mu^\top A\Gamma^{-1}A^\top\mu + \mu^\top(A\kappa_* - b)$$

maximized at $\mu_0 = (A\Gamma^{-1}A\top)^{-1}\{A\kappa_* - b\}_+$ where $\{[b_1, b_2]\}_+ = [\max\{b_1, 0\}, \max\{b_2, 0\}]$. Plugging $\mu_0$ back in the solution $\kappa'_0$, we get the solution $\kappa_0$

$$\kappa_0 = \kappa_* - \Gamma^{-1}A^\top(A\Gamma^{-1}A^\top)^{-1}\{A\kappa_* - b\}_+$$

and the optimal value follows.

$\square$

## A.2   Proof of Proposition 2

*Proof of Proposition 2.*

**Proof for $\mathcal{I}^1$.**   Fix $\alpha \in (0, 0.1)$ and consider the following instance with $m = 1$:

$$\mathcal{X} = \{e_1, e_2\}, \quad \mathcal{Z} = \{z_1, z_2\}, \quad z_1 = [1/4, 1/2], \quad z_2 = [3/4, 1/2 + \alpha]$$
$$\theta_* = e_1, \quad \mu_* = [0, 1], \quad \gamma = 1/2 + \alpha/2.$$

On this example, $z_1$ is safe and $z_2$ is unsafe with $\Delta_{\text{safe}}(z_2) = -\alpha/2$.

Let $A(\lambda) = \lambda_1 e_1 e_1^\top + \lambda_2 e_2 e_2^\top$ denote the design matrix. Then the allocation that minimizes $\mathcal{X}\mathcal{Y}_{\text{diff}}$:

$$\max_{z,z'\in\mathcal{Z}} \|z - z'\|^2_{A(\lambda)^{-1}} = \frac{1}{4\lambda_1} + \frac{\alpha^2}{\lambda_2}$$

is

$$\lambda_1 = \frac{1}{1 + 2\alpha}, \quad \lambda_2 = \frac{2\alpha}{1 + 2\alpha}.$$

Denote this allocation as $\widetilde{\lambda}$.

Applying the Transportation Lemma of [22], this implies that any $\delta$-PAC strategy must have

$$\mathbb{E}[T_1]\text{KL}(\nu_{(\theta_*,\mu_*),1}\|\nu_{(\theta,\mu),1}) + \mathbb{E}[T_2]\text{KL}(\nu_{(\theta_*,\mu_*),2}\|\nu_{(\theta,\mu),2}) \geq \log\frac{1}{2.4\delta}$$

for all $(\theta, \mu) \in \mathcal{C}_{\text{alt}}$, where $\mathcal{C}_{\text{alt}}$ is defined as in Lemma 4. If a learner plays $\widetilde{\lambda}$ for $T$ steps, they will have $\mathbb{E}[T_1] = \lambda_1 \mathbb{E}[T], \mathbb{E}[T_2] = \lambda_2 T$. In this case, the above can be rewritten as

$$\mathbb{E}[T] \geq \log \frac{1}{2.4\delta} \cdot \frac{1}{\lambda_1 \text{KL}(\nu_{(\theta_*,\mu_*),1}||\nu_{(\theta,\mu),1}) + \lambda_2 \text{KL}(\nu_{(\theta_*,\mu_*),2}||\nu_{(\theta,\mu),2})}$$

$$= \log \frac{1}{2.4\delta} \cdot \frac{2}{||\theta_* - \theta||^2_{A(\widetilde{\lambda})} + ||\mu_* - \mu||^2_{A(\widetilde{\lambda})}}.$$

where the equality follows by the same calculation as in the proof of Theorem 3. Take $(\theta, \mu)$ to be $\theta = \theta_*, \mu = [0, 1 - \frac{\alpha}{1+2\alpha}]$ and note that $(\theta, \mu) \in \mathcal{C}_{\text{alt}}$ since with this choice of $\mu$, arm $z_2$ is now safe. Now,

$$||\mu_* - \mu||^2_{A(\widetilde{\lambda})} = (\frac{\alpha}{1+2\alpha})^2 \cdot \frac{2\alpha}{1+2\alpha} = \frac{2\alpha^3}{(1+2\alpha)^3}.$$

This gives a lower bound of

$$\mathbb{E}[T] \geq \log \frac{1}{2.4\delta} \cdot \frac{2(1+2\alpha)^3}{2\alpha^3} \geq \log \frac{1}{2.4\delta} \cdot \frac{1}{\alpha^4}.$$

This lower bound is for best-arm identification ($\epsilon = 0$), but setting $\alpha \leftarrow 2\epsilon - a$ for $a$ arbitrarily small, identifying an $\epsilon$-optimal, $\epsilon$-safe arm is equivalent to identifying the best arm, so this therefore holds as a lower bound on $(\epsilon, \delta)$-PAC algorithms.

The upper bound on the performance of BESIDE follows trivially since by setting $\lambda_1 = \lambda_2$, we can make the numerator in both terms of the complexity $\mathcal{O}(1)$, and the denominator of each term will be at least $\epsilon^2$.

**Proof for $\mathcal{I}^2$.** Fix $\alpha \in (0, 0.1)$ and consider the following instance with $m = 1$:

$$\mathcal{X} = \{e_1, e_2\}, \quad \mathcal{Z} = \{z_1, z_2\}, \quad z_1 = [1/2 + \alpha^2/2, 0], \quad z_2 = [1/2, \alpha/2]$$
$$\theta_* = [1/2, 0], \quad \mu_* = [0, 0], \quad \gamma = 1.$$

On this instance, both $z_1$ and $z_2$ are safe, and $z_1$ is optimal.

The $\mathcal{X}\mathcal{Y}_{\text{safe}}$ design minimizes:

$$\max_{z \in \mathcal{Z}} ||z||^2_{A(\lambda)^{-1}} = \max \left\{ \frac{1 + 2\alpha + \alpha^2}{4\lambda_1}, \frac{1}{4\lambda_1} + \frac{\alpha^2}{4\lambda_2} \right\}.$$

Some computation shows that, for $\alpha$ small, the optimal settings are $\lambda_1 = \mathcal{O}(1)$ and $\lambda_2 = \mathcal{O}(\alpha)$ (where here $\mathcal{O}(\cdot)$ hides terms that are $o(\alpha)$). Denote this allocation as $\widetilde{\lambda}$. Following the same argument as above, we have

$$\mathbb{E}[T] \geq \log \frac{1}{2.4\delta} \cdot \frac{2}{||\theta_* - \theta||^2_{A(\widetilde{\lambda})} + ||\mu_* - \mu||^2_{A(\widetilde{\lambda})}}$$

for any $(\mu, \theta) \in \mathcal{C}_{\text{alt}}$. Let $\theta = [1/2, 2\alpha]$ and note that $(\mu_*, \theta) \in \mathcal{C}_{\text{alt}}$ since $z_2$ is now the optimal arm with this $\theta$. We then have

$$||\theta_* - \theta||^2_{A(\widetilde{\lambda})} + ||\mu_* - \mu||^2_{A(\widetilde{\lambda})} = \mathcal{O}(\alpha^3)$$

which gives a lower bound of

$$\mathbb{E}[T] \geq \Omega \left( \frac{1}{\alpha^3} \cdot \log \frac{1}{\delta} \right).$$

This lower bound holds for the best-arm identification problem, but setting $\alpha \leftarrow \sqrt{2\epsilon} - a$ for $a$ arbitrarily small, finding an $\epsilon$-optimal arm is equivalent to finding the best arm, so the lower bound applies in that setting as well.

To compute the sample complexity of BESIDE, we note that $\Delta_{\text{safe}}(z_1) = \Delta_{\text{safe}}(z_2) = 1$, so the first term in the complexity is negligible. We also have that $\Delta^{\widetilde{\epsilon}}(z_2) = \alpha^2/2 = \mathcal{O}(\epsilon)$ for $\widetilde{\epsilon} \leq 1$. Thus, the second term in the complexity scales as

$$\widetilde{\mathcal{O}} \left( \inf_{\lambda \in \triangle_{\mathcal{X}}} \max_{z,z' \in \mathcal{Z}} \frac{||z - z'||^2_{A(\lambda)^{-1}} \cdot \log 1/\delta}{\epsilon^2} \right) = \widetilde{\mathcal{O}} \left( \inf_{\lambda \in \triangle_{\mathcal{X}}} \frac{(\alpha^4/\lambda_1 + \alpha^2/\lambda_2) \cdot \log 1/\delta}{\epsilon^2} \right)$$

$$= \widetilde{\mathcal{O}} \left( \frac{\alpha^2 \cdot \log 1/\delta}{\epsilon^2} \right)$$

$$= \widetilde{\mathcal{O}} \left( \frac{\log 1/\delta}{\epsilon} \right).$$

$\square$

## B  Robust Mean Estimation

In order to form estimates of $z^\top \theta_*$ and $z^\top \mu_{*,i}$, we will rely on the RIPS procedure proposed in [6], instantiated with the robust Catoni estimator [8].

**Catoni Estimation.**  The robust Catoni mean estimator proposed in [8] is defined as follows.

**Definition 2** (Catoni Estimator). *Consider real values $X_1, \ldots, X_T$. Then the robust Catoni mean estimator, $\mathsf{cat}_\alpha[\{X_t\}_{t=1}^T]$, with parameter $\alpha > 0$ is the unique root $z$ of the function*

$$f_{\mathsf{cat}}(z; \{X_i\}_{i=1}^T, \alpha) := \sum_{t=1}^T \psi_{\mathsf{cat}}(\alpha(X_t - z)) \quad for \quad \psi_{\mathsf{cat}}(y) := \begin{cases} \log(1 + y + y^2) & y \geq 0 \\ \log(1 - y + y^2) & y < 0 \end{cases}.$$

*The Catoni estimator satisfies the following guarantee.*

**Proposition 6.** *Let $X_1, \ldots, X_T$ be independent, identically distributed random variables with mean $\zeta$ and variance $\sigma^2 < \infty$. Fix $\delta \in (0, 1)$ and assume $T \geq 4 \log(1/\delta)$. Then the Catoni estimator $\mathsf{cat}_\alpha[\{X_t\}_{t=1}^T]$ with parameter*

$$\alpha = \sqrt{\frac{2 \log 1/\delta}{T \sigma^2}} \tag{6}$$

*satisfies, with probability at least $1 - 2\delta$,*

$$\left| \mathsf{cat}_\alpha[\{X_t\}_{t=1}^T] - \zeta \right| \leq \sqrt{\frac{8 \sigma^2 \log 1/\delta}{T}}.$$

Notably, the estimation error given by Proposition 6 scales only with the variance of the random variables, and not with their magnitude.

**Robust Inverse Propensity Score (RIPS) Estimator.**  We apply the Catoni estimator with the RIPS estimator of [6]. In particular, consider running the following procedure.

---

**Algorithm 3** Robust Inverse Propensity Score Estimation (RIPS)

---

1: **input:** samples $\{(x_t, r_t)\}_{t=1}^T$ for $x_t \sim \lambda$ and $r_t = \theta^\top x_t + w_t$, active set $\mathcal{Y}$, confidence $\delta$
2: For each $y \in \mathcal{Y}$, set $W^y \leftarrow \mathsf{cat}_\alpha[\{y^\top A(\lambda)^{-1} x_t r_t\}_{t=1}^T]$, for $\alpha$ chosen as in (6) with $\delta \leftarrow \frac{\delta}{2|\mathcal{Y}|}$, and $A(\lambda) = \sum_{x \in \mathcal{X}} \lambda_x x x^\top$.
3: Set

$$\widehat{\theta} = \arg \min_\theta \max_{y \in \mathcal{Y}} \frac{|\theta^\top y - W^y|}{\|y\|_{A(\lambda)^{-1}}}.$$

4: **return** $\widehat{\theta}$

---

We have the following guarantee on this procedure.

**Proposition 7** (Theorem 1 of [6]). *If $T \geq 4 \log \frac{2|\mathcal{Z}|}{\delta}$, then with probability at least $1 - \delta$, for all $z \in \mathcal{Z}$, the RIPS estimator of Algorithm 3 returns an estimate $\widehat{\theta}$ which satisfies:*

$$|y^\top(\widehat{\theta} - \theta_*)| \leq \|y\|_{A(\lambda)^{-1}} \cdot \sqrt{\frac{8 \log(2|\mathcal{Z}|/\delta)}{T}}.$$

The use of the RIPS estimator allows us to avoid sophisticated rounding procedures often found in the linear bandit literature. Note that the RIPS estimator can be computed in time scaling polynomially in $|\mathcal{Y}|$, $d$, and $T$.

# C RAGE$^\epsilon$

**A note on constants.** Throughout our algorithm definitions, in both this section and the following, we use generic constants rather than precise numerical settings, and carry these generic constants through our proofs. At various points in the proofs, we require that these constants satisfy certain constraints. The following result shows that there exist suitable settings for all constants such that these constraints are satisfied.

**Lemma 8.** *There exist settings of* $c_a, c_b, c_d, c_e, c_f, c_e, c_g, c_1, c_2, c_3, c_4, c_\Delta$ *and* $c_0$ *such that Equations* (11), (14), (15), (16), (17), (18), (19), (7), (8), *and* (9) *are satisfied, and*

$$\frac{c_3(1 + c_g)}{1 - c_3} \leq 0.2, \quad c_g \leq 0.2, \quad c_0 \geq 0.0001.$$

*Proof.* First, note that in addition to the conditions listed above, we must also have

$$c_1 \leq c_f, \quad 3(c_d + c_e) \leq c_2.$$

Furthermore, by Lemma 20, it suffices to always take $c_\Delta = 3c_d + 3c_e - c_g$. Direct computation then shows that the following settings suffice, up to machine precision:

$$c_1 = 0.05978841810030329$$
$$c_2 = 0.0600087370242953$$
$$c_3 = 0.1$$
$$c_4 = 0.1$$
$$c_a = 0.0013004532984432395$$
$$c_b = 0.41043329378840077$$
$$c_d = 0.01$$
$$c_e = 0.01$$
$$c_c = 0.0014065949472697806$$
$$c_g = 0.178$$
$$c_f = c_1$$

Given these settings, we can bound

$$\frac{c_3(1 + c_g)}{1 - c_3} \leq 0.5.$$

$\square$

## C.1 Preliminaries

**Assumptions and Definitions.** For all $y \in \mathcal{Y}$, $\widehat{\Delta}_{\text{safe}}(y) \geq -c_\Delta \epsilon$. We will also assume that $\mathcal{Y} \subseteq \mathcal{X}$. We define

$$y_\star = \arg\min_{y \in \mathcal{Y}} y^\top \theta_*$$

and

$$\Delta(z) = \theta_*^\top (z - y_\star).$$

We will take $\gamma = 0$, so we set $A(\lambda) = \sum_{x \in \mathcal{X}} \lambda_x x x^\top$.

## C.2 Algorithm and Main Results

At a high-level, RAGE$^\epsilon$ attempts to estimate the difference between the performance of each $z \in \mathcal{Z}$ and the *best* $y \in \mathcal{Y}$. The safety gap estimate, $\widehat{\Delta}_{\text{safe}}(z)$, acts as a regularizer: if $\widehat{\Delta}_{\text{safe}}(z) < 0$, then we do not seek to estimate the gap of $z$ with as high accuracy, since we can already eliminate it by showing it is unsafe. The proof in this section follow closely the proof given in Section 6.4.4 of [17].

**Algorithm 4** $\text{RAGE}^\epsilon$

---

1: **input:** active set $\mathcal{Z}$, optimal set $\mathcal{Y}$, tolerance $\epsilon$, confidence $\delta$, safety gap estimate $\{\widehat{\Delta}_{\text{safe}}(z)\}_{z \in \mathcal{Z}}$
2: Choose $\widehat{y}_0$ arbitrarily from $\mathcal{Y}$, set $\widehat{\Delta}^0(z) \leftarrow 0$ for all $z \in \mathcal{Z}$
3: **for** $\ell = 1, 2, \ldots, \lceil \log(2/c_f \epsilon) \rceil$ **do**
4:     $\epsilon_\ell \leftarrow \frac{2}{c_f} \cdot 2^{-\ell}$
5:     Let $\tau_\ell$ be the minimal value of $\tau = 2^j \geq 4 \log \frac{4|\mathcal{Z}|^2 \ell^2}{\delta}$ such that the objective to the following is no greater than $c_c \epsilon_\ell$, and $\lambda_\ell$ the corresponding optimal distribution

$$\inf_{\lambda \in \triangle_{\mathcal{X}}} \max_{z \in \mathcal{Z}} -c_a(\mathfrak{p}(-\widehat{\Delta}_{\text{safe}}(z)) + \mathfrak{p}(\widehat{\Delta}^{\ell-1}(z)) + \epsilon_\ell) + \sqrt{\frac{\|z - \widehat{y}_{\ell-1}\|^2_{A(\lambda)^{-1}} \cdot \log(\frac{4|\mathcal{Z}|^2 \ell^2}{\delta})}{\tau}}.$$

6:     Sample $x_t \sim \lambda_\ell$, collect observations $\{(x_t, r_t, s_{t,1}, \ldots, s_{t,m})\}_{t=1}^{\tau_\ell}$
7:     $\mathcal{W} \leftarrow \{z - z' \; : \; z, z' \in \mathcal{Z}\}$
8:     $\widehat{\theta}^\ell \leftarrow \text{RIPS}(\{(x_t, r_t)\}_{t=1}^{\tau_\ell}, \mathcal{W}, \frac{\delta}{2\ell^2})$
9:     Set

$$\widehat{y}_\ell \leftarrow \arg\min_{y \in \mathcal{Y}} y^\top \widehat{\theta}^\ell + 8\sqrt{\frac{\|y - \widehat{y}_{\ell-1}\|^2_{A(\lambda_\ell)^{-1}} \cdot \log(\frac{4|\mathcal{Z}|^2 \ell^2}{\delta})}{\tau_\ell}}$$

$$\widehat{\Delta}^\ell(y) \leftarrow (y - \widehat{y}_\ell)^\top \widehat{\theta}^\ell + \sqrt{\frac{\|y - \widehat{y}_\ell\|^2_{A(\lambda_\ell)^{-1}} \cdot \log(\frac{4|\mathcal{Z}|^2 \ell^2}{\delta})}{\tau_\ell}}$$

10: **return** $\{\widehat{\Delta}^\ell(z)\}_{z \in \mathcal{Z}}$

---

**Theorem 1.** *With probability at least $1 - \delta$, $\text{RAGE}^\epsilon$ will terminate after collecting at most*

$$C \cdot \sum_{\ell=1}^{\lceil \log 2/c_f \epsilon \rceil} \inf_{\lambda \in \triangle_{\mathcal{X}}} \max_{z \in \mathcal{Z}} \frac{\|z - y_\star\|^2_{A(\lambda)^{-1}}}{(\mathfrak{p}(-\widehat{\Delta}_{\text{safe}}(z)) + \mathfrak{p}(\Delta(z)) + \epsilon_\ell)^2} \cdot \log(\frac{4|\mathcal{Z}|^2 \ell^2}{\delta}) + 4\lceil \log \frac{2}{c_f \epsilon} \rceil \log(\frac{4|\mathcal{Z}|^2 \lceil \log \frac{2}{c_f \epsilon} \rceil}{\delta})$$

*samples, for a universal constant $C$, and will output estimates of the gaps $\widehat{\Delta}(z)$ such that, for all $z \in \mathcal{Z}$,*

$$|\widehat{\Delta}(z) - \Delta(z)| \leq c_f \left( \epsilon + \mathfrak{p}(\Delta(z)) + \mathfrak{p}(-\widehat{\Delta}_{\text{safe}}(z)) \right).$$

### C.3 Estimating the Gaps

**Lemma 9.** *Let $\mathcal{E}_{\text{RAGE}^\epsilon}$ denote the event that for all $\ell$ and all $z, z' \in \mathcal{X}$, we have:*

$$|(\widehat{\theta}^\ell - \theta_*)^\top (z - z')| \leq \sqrt{8\|z - z'\|^2_{A(\lambda_\ell)^{-1}} \cdot \frac{\log \frac{4|\mathcal{Z}|^2 \ell^2}{\delta}}{\tau_\ell}}.$$

*Then $\Pr[\mathcal{E}_{\text{RAGE}^\epsilon}] \geq 1 - \delta$.*

*Proof.* Since $\tau_\ell \geq 4 \log \frac{4|\mathcal{Z}|^2 \ell^2}{\delta}$, we can apply Proposition 7 to get that, with probability at least $1 - \delta/2\ell^2$, for all $w \in \mathcal{W}$,

$$|(\widehat{\theta}^\ell - \theta_*)^\top w| \leq \sqrt{8\|w\|^2_{A(\lambda_\ell)^{-1}} \cdot \frac{\log \frac{4|\mathcal{Z}|^2 \ell^2}{\delta}}{\tau_\ell}}.$$

The result then follows by a union bound since

$$\sum_{\ell=1}^{\infty} \frac{\delta}{2\ell^2} = \frac{\pi^2}{12}\delta \leq \delta.$$

$\square$

**Lemma 10.** *On $\mathcal{E}_{\text{RAGE}^\epsilon}$, for all $z \in \mathcal{Z}$ and all $\ell$,*

$$|\widehat{\Delta}^\ell(z) - \theta_*^\top(z - \widehat{y}_\ell)| \leq 8c_a\left(\mathfrak{p}(\widehat{\Delta}^{\ell-1}(z)) + \mathfrak{p}(-\widehat{\Delta}_{\text{safe}}(z)) + \mathfrak{p}(\widehat{\Delta}^{\ell-1}(\widehat{y}_\ell))\right) + 8(c_c + c_a + 2c_a c_\Delta)\epsilon_\ell.$$

*Proof.* By construction, we have that

$$\max_{z \in \mathcal{Z}} -c_a(\mathfrak{p}(-\widehat{\Delta}_{\text{safe}}(z)) + \mathfrak{p}(\widehat{\Delta}^{\ell-1}(z)) + \epsilon_\ell) + \sqrt{\frac{\|z - \widehat{y}_{\ell-1}\|^2_{A(\lambda_\ell)^{-1}} \cdot \log(\frac{4|\mathcal{Z}|^2\ell^2}{\delta})}{\tau_\ell}} \leq c_c\epsilon_\ell.$$

This implies that, for all $z \in \mathcal{Z}$:

$$\sqrt{\frac{\|z - \widehat{y}_{\ell-1}\|^2_{A(\lambda_\ell)^{-1}} \cdot \log(\frac{4|\mathcal{Z}|^2\ell^2}{\delta})}{\tau_\ell}} \leq c_a(\mathfrak{p}(-\widehat{\Delta}_{\text{safe}}(z)) + \mathfrak{p}(\widehat{\Delta}^{\ell-1}(z))) + (c_c + c_a)\epsilon_\ell.$$

On $\mathcal{E}_{\text{RAGE}^\epsilon}$, we have

$$|\widehat{\Delta}_\ell(z) - \theta_*^\top(z - \widehat{y}_\ell)| \leq \sqrt{8\|z - \widehat{y}_\ell\|^2_{A(\lambda_\ell)^{-1}} \cdot \frac{\log(\frac{4|\mathcal{Z}|^2\ell^2}{\delta})}{\tau_\ell}}$$

$$\leq \sqrt{16\|\widehat{y}_{\ell-1} - \widehat{y}_\ell\|^2_{A(\lambda_\ell)^{-1}} \cdot \frac{\log(\frac{4|\mathcal{Z}|^2\ell^2}{\delta})}{\tau_\ell}} + \sqrt{16\|\widehat{y}_{\ell-1} - \widehat{y}_\ell\|^2_{A(\lambda_\ell)^{-1}} \cdot \frac{\log(\frac{4|\mathcal{Z}|^2\ell^2}{\delta})}{\tau_\ell}}$$

$$\leq 8c_a\left(\mathfrak{p}(-\widehat{\Delta}_{\text{safe}}(z)) + \mathfrak{p}(\widehat{\Delta}^{\ell-1}(z)) + \mathfrak{p}(-\widehat{\Delta}_{\text{safe}}(\widehat{y}_\ell)) + \mathfrak{p}(\widehat{\Delta}^{\ell-1}(\widehat{y}_\ell))\right) + 8(c_c + c_a)\epsilon_\ell.$$

By construction we have that $\widehat{\Delta}_{\text{safe}}(\widehat{y}_\ell) \geq -c_\Delta\epsilon \geq -2c_\Delta\epsilon_\ell$, so $\mathfrak{p}(-\widehat{\Delta}_{\text{safe}}(\widehat{y}_\ell)) \leq 2c_\Delta\epsilon_\ell$, which proves the result. $\square$

**Lemma 11.** *On $\mathcal{E}_{\text{RAGE}^\epsilon}$ and the event that $\widehat{\Delta}^{\ell-1}(y_\star) \leq c_b\epsilon_\ell$, we have*

$$\Delta(\widehat{y}_\ell) \leq 6(c_c + c_a(1 + c_b + 2c_\Delta))\epsilon_\ell.$$

*Proof.* By the definition of $\mathcal{E}_{\text{RAGE}^\epsilon}$ and $\widehat{y}_\ell$, we can bound

$$\theta_*^\top(\widehat{y}_\ell - \widehat{y}_{\ell-1}) \leq (\widehat{\theta}^\ell)^\top(\widehat{y}_\ell - \widehat{y}_{\ell-1}) + \sqrt{8\|\widehat{y}_\ell - \widehat{y}_{\ell-1}\|^2_{A(\lambda_\ell)^{-1}} \cdot \frac{\log(\frac{4|\mathcal{Z}|^2\ell^2}{\delta})}{\tau_\ell}}$$

$$= \min_{y \in \mathcal{Y}}(\widehat{\theta}^\ell)^\top(y - \widehat{y}_{\ell-1}) + \sqrt{8\|y - \widehat{y}_{\ell-1}\|^2_{A(\lambda_\ell)^{-1}} \cdot \frac{\log(\frac{4|\mathcal{Z}|^2\ell^2}{\delta})}{\tau_\ell}}$$

$$\leq (\widehat{\theta}^\ell)^\top(y_\star - \widehat{y}_{\ell-1}) + \sqrt{8\|y_\star - \widehat{y}_{\ell-1}\|^2_{A(\lambda_\ell)^{-1}} \cdot \frac{\log(\frac{4|\mathcal{Z}|^2\ell^2}{\delta})}{\tau_\ell}}$$

$$\leq \theta_*^\top(y_\star - \widehat{y}_{\ell-1}) + 2\sqrt{8\|y_\star - \widehat{y}_{\ell-1}\|^2_{A(\lambda_\ell)^{-1}} \cdot \frac{\log(\frac{4|\mathcal{Z}|^2\ell^2}{\delta})}{\tau_\ell}}.$$

By the definition of $\tau_\ell$ and $\lambda_\ell$, we have

$$c_c\epsilon_\ell \geq \max_{z \in \mathcal{Z}} -c_a(\mathfrak{p}(-\widehat{\Delta}_{\text{safe}}(z)) + \mathfrak{p}(\widehat{\Delta}^{\ell-1}(z)) + \epsilon_\ell) + \sqrt{\frac{\|z - \widehat{y}_{\ell-1}\|^2_{A(\lambda_\ell)^{-1}} \cdot \log(\frac{4|\mathcal{Z}|^2\ell^2}{\delta})}{\tau_\ell}}$$

$$\geq -c_a(\mathfrak{p}(-\widehat{\Delta}_{\text{safe}}(y_\star)) + \mathfrak{p}(\widehat{\Delta}^{\ell-1}(y_\star)) + \epsilon_\ell) + \sqrt{\frac{\|y_\star - \widehat{y}_{\ell-1}\|^2_{A(\lambda_\ell)^{-1}} \cdot \log(\frac{4|\mathcal{Z}|^2\ell^2}{\delta})}{\tau_\ell}}$$

$$\overset{(a)}{\geq} -c_a(\mathfrak{p}(\widehat{\Delta}^{\ell-1}(y_\star)) + (1 + 2c_\Delta)\epsilon_\ell) + \sqrt{\frac{\|y_\star - \widehat{y}_{\ell-1}\|^2_{A(\lambda_\ell)^{-1}} \cdot \log(\frac{4|\mathcal{Z}|^2\ell^2}{\delta})}{\tau_\ell}}$$

$$\overset{(b)}{\geq} -c_a(1 + c_b + 2c_\Delta)\epsilon_\ell + \sqrt{\|y_\star - \widehat{y}_{\ell-1}\|^2_{A(\lambda_\ell)^{-1}} \cdot \frac{\log(\frac{4|\mathcal{Z}|^2\ell^2}{\delta})}{\tau_\ell}}$$

where $(a)$ uses that $\widehat{\Delta}_{\text{safe}}(y_\star) \geq -c_\Delta\epsilon \geq -2c_\Delta\epsilon_\ell$, by definition, and $(b)$ follows by our assumption on $\widehat{\Delta}^{\ell-1}(y_\star)$. This implies that

$$\sqrt{\|y_\star - \widehat{y}_{\ell-1}\|^2_{A(\lambda_\ell)^{-1}} \cdot \frac{\log(\frac{4|\mathcal{Z}|^2\ell^2}{\delta})}{\tau_\ell}} \leq (c_c + c_a(1 + c_b + 2c_\Delta))\epsilon_\ell.$$

Combining this with the above we have that

$$\theta_*^\top(\widehat{y}_\ell - \widehat{y}_{\ell-1}) \leq \theta_*^\top(y_\star - \widehat{y}_{\ell-1}) + 6(c_c + c_a(1 + c_b + 2c_\Delta))\epsilon_\ell.$$

Rearranging this proves the result. $\qquad\square$

**Lemma 12.** *For all $z \in \mathcal{Z}$ and all $\ell$, on the event $\mathcal{E}_{\text{RAGE}^\epsilon}$,*
$$|\widehat{\Delta}^\ell(z) - \Delta(z)| \leq c_f\left(\epsilon_\ell + \mathfrak{p}(\Delta(z)) + \mathfrak{p}(-\widehat{\Delta}_{\text{safe}}(z))\right).$$

*Proof.* We prove this by induction. Assume that at $\ell - 1$, for all $z \in \mathcal{Z}$,
$$|\widehat{\Delta}^{\ell-1}(z) - \Delta(z)| \leq c_f\left(\epsilon_{\ell-1} + \mathfrak{p}(\Delta(z)) + \mathfrak{p}(-\widehat{\Delta}_{\text{safe}}(z))\right).$$

On $\mathcal{E}_{\text{RAGE}^\epsilon}$ and by Lemma 10 we can bound
$$\begin{aligned}
|\widehat{\Delta}^\ell(z) - \Delta(z)| &= |\widehat{\Delta}^\ell(z) - (R(z) - R(\widehat{y}_\ell) + R(\widehat{y}_\ell) - R(y_\star))| \\
&\leq |\widehat{\Delta}^\ell(z) - (R(z) - R(\widehat{y}_\ell))| + \Delta(\widehat{y}_\ell) \\
&\leq 8c_a\left(\mathfrak{p}(\widehat{\Delta}^{\ell-1}(z)) + \mathfrak{p}(-\widehat{\Delta}_{\text{safe}}(z)) + \mathfrak{p}(\widehat{\Delta}^{\ell-1}(\widehat{y}_\ell))\right) + 8(c_c + c_a + 2c_ac_\Delta)\epsilon_\ell + \Delta(\widehat{y}_\ell).
\end{aligned}$$

By the inductive hypothesis, we can bound
$$\begin{aligned}
\mathfrak{p}(\widehat{\Delta}^{\ell-1}(z)) &\leq (1 + c_f)\mathfrak{p}(\Delta(z)) + c_f\mathfrak{p}(-\widehat{\Delta}_{\text{safe}}(z)) + c_f\epsilon_{\ell-1} \\
\mathfrak{p}(\widehat{\Delta}^{\ell-1}(\widehat{y}_\ell)) &\leq (1 + c_f)\mathfrak{p}(\Delta(\widehat{y}_\ell)) + c_f\mathfrak{p}(-\widehat{\Delta}_{\text{safe}}(\widehat{y}_\ell)) + c_f\epsilon_{\ell-1}.
\end{aligned}$$

By construction $\mathfrak{p}(-\widehat{\Delta}_{\text{safe}}(\widehat{y}_\ell)) \geq -c_\Delta\epsilon \geq -2c_\Delta\epsilon_\ell$, so
$$\begin{aligned}
|\widehat{\Delta}^\ell(z) - \Delta(z)| \leq{}& 8c_a(1 + c_f)\mathfrak{p}(\Delta(z)) + 8c_a(1 + c_f)\mathfrak{p}(\widehat{\Delta}_{\text{safe}}(z)) + (8c_a(1 + c_f) + 1)\Delta(\widehat{y}_\ell) \\
&+ 8(c_ac_f(1 + c_\Delta) + c_c + c_a + 2c_ac_\Delta)\epsilon_\ell.
\end{aligned}$$

It remains to bound $\Delta(\widehat{y}_\ell) = R(\widehat{y}_\ell) - R(y_\star)$. On the inductive hypothesis, we have that
$$|\widehat{\Delta}^{\ell-1}(y_\star) - \Delta(y_\star)| \leq c_f\left(\epsilon_{\ell-1} + \mathfrak{p}(\Delta(y_\star)) + \mathfrak{p}(-\widehat{\Delta}_{\text{safe}}(y_\star))\right).$$

By definition, $\Delta(y_\star) = 0$ and $\widehat{\Delta}_{\text{safe}}(y_\star) \geq -c_\Delta\epsilon \geq -2c_\Delta\epsilon_\ell$, which implies that $\widehat{\Delta}^{\ell-1}(y_\star) \leq 2c_f(1 + c_\Delta)\epsilon_\ell$. It follows that the conditions of Lemma 11 are met as long as
$$2c_f(1 + c_\Delta) \leq c_b, \tag{7}$$
so we can bound $\Delta(\widehat{y}_\ell) \leq 6(c_c + c_a(1 + c_b + 2c_\Delta))\epsilon_\ell$. Thus,
$$\begin{aligned}
|\widehat{\Delta}^\ell(z) - \Delta(z)| \leq{}& 8c_a(1 + c_f)\mathfrak{p}(\Delta(z)) + 8c_a(1 + c_f)\mathfrak{p}(\widehat{\Delta}_{\text{safe}}(z)) + 8(c_ac_f(1 + c_\Delta) + c_c + c_a + 2c_ac_\Delta)\epsilon_\ell \\
&+ (8c_a(1 + c_f) + 1)(6(c_c + c_a(1 + c_b + 2c_\Delta)))\epsilon_\ell.
\end{aligned}$$

which proves the inductive hypothesis as long as
$$\begin{aligned}
8(c_ac_f(1 + c_\Delta) + c_c + c_a + 2c_ac_\Delta) + (8c_a(1 + c_f) + 1)(6(c_c + c_a(1 + c_b + 2c_\Delta))) &\leq c_f \\
8c_a(1 + c_f) &\leq c_f \tag{8}
\end{aligned}$$

For the base case, we need to show that
$$|\widehat{\Delta}^0(z) - \Delta(z)| \leq c_f\left(\epsilon_0 + \mathfrak{p}(\Delta(z)) + \mathfrak{p}(-\widehat{\Delta}_{\text{safe}}(z))\right).$$

By construction $\widehat{\Delta}^0(z) = 0$ for all $z$, and $\mathfrak{p}(\Delta(z)) \geq 0$, $\mathfrak{p}(-\widehat{\Delta}_{\text{safe}}(z)) \geq 0$. Thus, it suffices to show $|\Delta(z)| \leq c_f\epsilon_0$. However, by construction $|\Delta(z)| \leq 1$, and $c_f\epsilon_0 = 1$, which proves the base case. $\qquad\square$

## C.4 Bounding the Sample Complexity

**Lemma 13.** *On the event $\mathcal{E}_{\text{RAGE}^\epsilon}$, $\text{RAGE}^\epsilon$ will terminate after collecting at most*

$$C \cdot \sum_{\ell=1}^{\lceil \log(2/c_f\epsilon) \rceil} \inf_{\lambda \in \triangle_{\mathcal{X}}} \max_{z \in \mathcal{Z}} \frac{\|z - y_\star\|_{A(\lambda)^{-1}}^2}{(\mathfrak{p}(-\widehat{\Delta}_{\text{safe}}(z)) + \mathfrak{p}(\Delta(z)) + \epsilon_\ell)^2} \cdot \log(\tfrac{4|\mathcal{Z}|^2\ell^2}{\delta}) + 8\lceil \log \tfrac{2}{c_f\epsilon} \rceil \log(\tfrac{4|\mathcal{Z}|^2\lceil \log \frac{2}{c_f\epsilon} \rceil^2}{\delta})$$

*samples, for a universal constant $C$.*

*Proof.* If, for all $z \in \mathcal{Z}$,

$$\tau \geq \frac{\|z - \widehat{y}_{\ell-1}\|_{A(\lambda)^{-1}}^2}{(c_a(\mathfrak{p}(-\widehat{\Delta}_{\text{safe}}(z)) + \mathfrak{p}(\widehat{\Delta}^{\ell-1}(z)) + \epsilon_\ell) + c_c\epsilon_\ell)^2} \cdot \log(\tfrac{4|\mathcal{Z}|^2\ell^2}{\delta})$$

we will have that the objective on Algorithm 4 of $\text{RAGE}^\epsilon$ is less than $c_c\epsilon_\ell$. Since we can take the best-case $\lambda \in \triangle_{\mathcal{X}}$, and since we have that $\tau_\ell$ will be at most a factor of 2 from the optimal $\tau$, it follows that

$$\tau_\ell \leq \inf_{\lambda \in \triangle_{\mathcal{X}}} \max_{z \in \mathcal{Z}} \frac{2\|z - \widehat{y}_{\ell-1}\|_{A(\lambda)^{-1}}^2}{(c_a(\mathfrak{p}(-\widehat{\Delta}_{\text{safe}}(z)) + \mathfrak{p}(\widehat{\Delta}^{\ell-1}(z)) + \epsilon_\ell) + c_c\epsilon_\ell)^2} \cdot \log(\tfrac{4|\mathcal{Z}|^2\ell^2}{\delta}) \vee 8\log\tfrac{4|\mathcal{Z}|^2\ell^2}{\delta}$$

$$\leq \inf_{\lambda \in \triangle_{\mathcal{X}}} \max_{z \in \mathcal{Z}} \frac{2\|z - \widehat{y}_{\ell-1}\|_{A(\lambda)^{-1}}^2}{(c_a(\mathfrak{p}(-\widehat{\Delta}_{\text{safe}}(z)) + \mathfrak{p}(\widehat{\Delta}^{\ell-1}(z)) + \epsilon_\ell) + c_c\epsilon_\ell)^2} \cdot \log(\tfrac{4|\mathcal{Z}|^2\ell^2}{\delta}) + 8\log\tfrac{4|\mathcal{Z}|^2\ell^2}{\delta}$$

where the additional $8\log\tfrac{4|\mathcal{Z}|^2\ell^2}{\delta}$ factor arises since we always require $\tau_\ell \geq 4\log\tfrac{4|\mathcal{Z}|^2\ell^2}{\delta}$.

We can upper bound

$$\|z - \widehat{y}_{\ell-1}\|_{A(\lambda)^{-1}}^2 \leq 2\|z - y_\star\|_{A(\lambda)^{-1}}^2 + 2\|y_\star - \widehat{y}_{\ell-1}\|_{A(\lambda)^{-1}}^2.$$

By construction, $\mathfrak{p}(-\widehat{\Delta}_{\text{safe}}(\widehat{y}_{\ell-1})) \leq 2c_\Delta\epsilon_\ell$, so for any $z$, $\mathfrak{p}(-\widehat{\Delta}_{\text{safe}}(\widehat{y}_{\ell-1})) - 2c_\Delta\epsilon_\ell \leq \mathfrak{p}(-\widehat{\Delta}_{\text{safe}}(z))$. Furthermore, by definition,

$$\widehat{\Delta}^{\ell-1}(\widehat{y}_{\ell-1}) = 0$$

so $\mathfrak{p}(\widehat{\Delta}^{\ell-1}(z)) \geq \mathfrak{p}(\widehat{\Delta}^{\ell-1}(\widehat{y}_{\ell-1}))$. Thus,

$$\inf_{\lambda \in \triangle_{\mathcal{X}}} \max_{z \in \mathcal{Z}} \frac{\|z - \widehat{y}_{\ell-1}\|_{A(\lambda)^{-1}}^2}{(c_a\mathfrak{p}(-\widehat{\Delta}_{\text{safe}}(z)) + c_a\mathfrak{p}(\widehat{\Delta}^{\ell-1}(z)) + (c_a + c_c)\epsilon_\ell)^2}$$

$$\leq \inf_{\lambda \in \triangle_{\mathcal{X}}} \max_{z \in \mathcal{Z}} \frac{2\|z - y_\star\|_{A(\lambda)^{-1}}^2}{(c_a\mathfrak{p}(-\widehat{\Delta}_{\text{safe}}(z)) + c_a\mathfrak{p}(\widehat{\Delta}^{\ell-1}(z)) + (c_a + c_c)\epsilon_\ell)^2}$$

$$+ \frac{2\|\widehat{y}_{\ell-1} - y_\star\|_{A(\lambda)^{-1}}^2}{(c_a\mathfrak{p}(-\widehat{\Delta}_{\text{safe}}(\widehat{y}_{\ell-1})) + c_a\mathfrak{p}(\widehat{\Delta}^{\ell-1}(\widehat{y}_{\ell-1})) + (c_a + c_c - 2c_ac_\Delta)\epsilon_\ell)^2}$$

$$\leq \inf_{\lambda \in \triangle_{\mathcal{X}}} \max_{z \in \mathcal{Z}} \frac{4\|z - y_\star\|_{A(\lambda)^{-1}}^2}{(c_a\mathfrak{p}(-\widehat{\Delta}_{\text{safe}}(z)) + c_a\mathfrak{p}(\widehat{\Delta}^{\ell-1}(z)) + (c_a + c_c - 2c_ac_\Delta)\epsilon_\ell)^2}.$$

By Lemma 12, we can lower bound

$$\widehat{\Delta}^{\ell-1}(z) \geq \Delta(z) - c_f(\epsilon_\ell + \mathfrak{p}(\Delta(z)) + \mathfrak{p}(-\widehat{\Delta}_{\text{safe}}(z)))$$

so

$$c_a\mathfrak{p}(-\widehat{\Delta}_{\text{safe}}(z)) + c_a\mathfrak{p}(\widehat{\Delta}^{\ell-1}(z)) + (c_a + c_c - 2c_ac_\Delta)\epsilon_\ell$$

$$\geq c_a(1 - c_f)\mathfrak{p}(-\widehat{\Delta}_{\text{safe}}(z)) + c_a(1 - c_f)\mathfrak{p}(\Delta(z)) + (c_a + c_c - 2c_ac_\Delta - c_ac_f)\epsilon_\ell.$$

The result follows by combining these inequalities and as long as

$$c_a(1 - c_f) \geq c_0, \qquad c_a + c_c - 2c_ac_\Delta - c_ac_f \geq c_0. \tag{9}$$

$\square$

*Proof of Theorem 1.* Theorem 1 follows directly from Lemma 13 and Lemma 12 since, by Lemma 9, $\mathcal{E}_{\text{RAGE}^\epsilon}$ holds with probability at least $1 - \delta$. $\square$

**Algorithm 5 Be**st **S**afe Arm **Ide**ntification (BESIDE, defined with generic constants)

1: **input:** tolerance $\epsilon$, confidence $\delta$

2: $\iota_\epsilon \leftarrow \lceil \log(\frac{2}{\min\{c_3,c_4\}\cdot\epsilon}) \rceil, \widehat{\Delta}^0_{\text{safe}}(z) \leftarrow 0, \widehat{\Delta}^0(z) \leftarrow 0$ for all $z \in \mathcal{Z}$

3: **for** $\ell = 1, 2, \ldots, \iota_\epsilon$ **do**

4:      $\epsilon_\ell \leftarrow \frac{2}{\min\{c_3,c_4\}} \cdot 2^{-\ell}$

     `// Solve experiment to reduce uncertainty on safety constraints`

5:      Let $\tau_\ell$ be the minimal value of $\tau = 2^j \geq 4\log\frac{4m|\mathcal{Z}|\ell^2}{\delta}$ such that the objective to the following is no greater than $c_e\epsilon_\ell$, and $\lambda_\ell$ the corresponding optimal distribution

$$\inf_{\lambda \in \triangle_{\mathcal{X}}} \max_{z \in \mathcal{Z}} -c_d \left( \min_j |\widehat{\Delta}^{j,\ell-1}_{\text{safe}}(z)| + \max_j \mathfrak{p}(-\widehat{\Delta}^{j,\ell-1}_{\text{safe}}(z)) + \mathfrak{p}(\widehat{\Delta}^{\ell-1}(z)) + \epsilon_\ell \right) + \sqrt{\frac{\|z\|^2_{A(\lambda)^{-1}} \cdot \log(\frac{4m|\mathcal{Z}|\ell^2}{\delta})}{\tau}}$$

6:      Sample $x_t \sim \lambda_\ell$, collect $\tau_\ell$ observations $\{(x_t, r_t, s_{t,1}, \ldots, s_{t,m})\}_{t=1}^{\tau_\ell}$

7:      $\{\widehat{\mu}^{i,\ell}\}_{i=1}^m \leftarrow \mathsf{RIPS}(\{(x_t, s_{t,i})\}_{t=1}^{\tau_\ell}, \mathcal{Z}, \frac{\delta}{2m\ell^2})$      `// Estimate safety constraints`

8:      $\widehat{\Delta}^{i,\ell}_{\text{safe}}(z) \leftarrow \gamma - z^\top \widehat{\mu}^{i,\ell} + \|z\|_{A(\lambda_\ell)^{-1}}\sqrt{\tau_\ell^{-1}\log(\frac{4m|\mathcal{Z}|\ell^2}{\delta})}$ `// Safety gap estimates`

     `// Form set of arms guaranteed to be safe`

9:

$$\mathcal{Y}_\ell \leftarrow \left\{ z \in \mathcal{Z} : 8c_d \left( \min_j |\widehat{\Delta}^{j,\ell-1}_{\text{safe}}(z)| + \max_j \mathfrak{p}(-\widehat{\Delta}^{j,\ell-1}_{\text{safe}}(z)) + \mathfrak{p}(\widehat{\Delta}^{\ell-1}(z)) \right) \right.$$
$$\left. + 8(c_d + c_e)\epsilon_\ell \leq \widehat{\Delta}^{i,\ell}_{\text{safe}}(z), \forall i \in [n] \right\} \cup \mathcal{Y}_{\ell-1}$$

     `// Refine estimates of optimality gaps`

10:      $\{\widehat{\Delta}^\ell(z)\}_{z \in \mathcal{Z}} \leftarrow \mathsf{RAGE}^\epsilon\left(\mathcal{Z}, \mathcal{Y}_\ell, \epsilon_\ell, \frac{\delta}{4\ell^2}, \{\widehat{\Delta}_{\text{safe}}(z) \leftarrow \max_j \mathfrak{p}(-\widehat{\Delta}^{j,\ell}_{\text{safe}}(z))\}_{z \in \mathcal{Z}}\right)$

     `// Form set of arms guaranteed to be at most ε-unsafe`

11:

$$\mathcal{Y}_{\text{end}} \leftarrow \left\{ z \in \mathcal{Z} : 8c_d \left( \min_j |\widehat{\Delta}^{j,\iota_\epsilon}_{\text{safe}}(z)| + \max_j \mathfrak{p}(-\widehat{\Delta}^{j,\iota_\epsilon}_{\text{safe}}(z)) + \mathfrak{p}(\widehat{\Delta}^{\iota_\epsilon}(z)) \right) \right.$$
$$\left. + 8(c_d + c_e)\epsilon - c_g\epsilon \leq \widehat{\Delta}^{i,\iota_\epsilon}_{\text{safe}}(z), \forall i \in [n] \right\}$$

     `// Find ε-good arm out of ε-safe arms`

12: $\{\widehat{\Delta}^{\text{end}}(z)\}_{z \in \mathcal{Y}_{\text{end}}} \leftarrow \mathsf{RAGE}^\epsilon(\mathcal{Y}_{\text{end}}, \mathcal{Y}_{\text{end}}, \epsilon, \delta)$

13: **return** $\widehat{z} = \arg\min_{z \in \mathcal{Y}_{\text{end}}} \widehat{\Delta}^{\text{end}}(z)$

# D    Safe Best-Arm Identification

## D.1    Preliminaries

In general we want to consider multiple safety constraints, and let $m$ denote the number of constraints. In such settings, we will denote $\Delta^i_{\text{safe}}(z)$ the safety gap for safety constraint $i$.

Define

$$\widetilde{\Delta}^\ell(z) := \theta_*^\top z - \min_{y \in \mathcal{Y}_\ell} \theta_*^\top y.$$

## D.2    Algorithm and Main Result

**Theorem 14** (Full version of Theorem 1)**.** *With probability at least $1 - 2\delta$, Algorithm 1 returns an arm $\widehat{z}$ such that*

$$\widehat{z}^\top \theta_* \geq (z_*)^\top \theta_* - \epsilon, \quad \Delta_{\text{safe}}(\widehat{z}) \geq -\epsilon \tag{10}$$

*and terminates after collecting at most*

$$C \cdot \sum_{\ell=1}^{\iota_\epsilon} \inf_{\lambda \in \triangle_{\mathcal{X}}} \max_{z \in \mathcal{Z}} \frac{\|z\|^2_{A(\lambda)^{-1}} \cdot \log(\frac{4m|\mathcal{Z}|\ell^2}{\delta})}{\left( \min_j |\Delta^j_{\text{safe}}(z)| + \max_j \mathfrak{p}(-\Delta^j_{\text{safe}}(z)) + \mathfrak{p}(\Delta^{\epsilon_{\ell-1}}(z)) + \epsilon_\ell \right)^2}$$

$$+ C \log \frac{1}{\epsilon} \cdot \sum_{\ell=1}^{\iota_\epsilon} \inf_{\lambda \in \triangle_{\mathcal{X}}} \max_{z \in \mathcal{Z}} \frac{\|z - z_*\|^2_{A(\lambda)^{-1}} \cdot \log(\frac{8|\mathcal{Z}|^2 \log^4(1/\epsilon)}{\delta})}{(\max_j \mathfrak{p}(-\Delta^j_{\mathrm{safe}}(z)) + \mathfrak{p}(\Delta^{\epsilon_\ell}(z)) + \epsilon_\ell)^2} + C_0$$

*samples for a universal constant* $C$, $C_0 = \mathrm{poly} \log(\frac{1}{\epsilon}, |\mathcal{Z}|) \cdot \log \frac{1}{\delta}$.

## D.3 Estimating the Safety Value

**Lemma 15.** *Let* $\mathcal{E}_{\mathrm{safe}}$ *denote the event that, for all* $\ell$, $z \in \mathcal{Z}$, $i \in [m]$:

$$|z^\top(\widehat{\mu}^{i,\ell} - \mu^i_*)| \leq \sqrt{8\|z\|^2_{A(\lambda_\ell)^{-1}} \cdot \frac{\log(\frac{4m|\mathcal{Z}|\ell^2}{\delta})}{\tau_\ell}}.$$

*Then* $\Pr[\mathcal{E}_{\mathrm{safe}}] \geq 1 - \delta$.

*Proof.* This follows directly from Proposition 7 and a union bound, as in Lemma 9. $\qquad\square$

**Lemma 16.** *On* $\mathcal{E}_{\mathrm{safe}}$, *for all* $z \in \mathcal{Z}$, $i \in [m]$, *and all* $\ell$,

$$|\widehat{\Delta}^{i,\ell}_{\mathrm{safe}}(z) - \Delta^i_{\mathrm{safe}}(z)| \leq 3c_d \left( \min_j |\widehat{\Delta}^{j,\ell-1}_{\mathrm{safe}}(z)| + \max_j \mathfrak{p}(-\widehat{\Delta}^{j,\ell-1}_{\mathrm{safe}}(z)) + \mathfrak{p}(\widehat{\Delta}^{\ell-1}(z)) \right) + 3(c_d + c_e)\epsilon_\ell.$$

*Proof.* By construction, we have that

$$\max_{z \in \mathcal{Z}} -c_d \left( \min_j |\widehat{\Delta}^{j,\ell-1}_{\mathrm{safe}}(z)| + \max_j \mathfrak{p}(-\widehat{\Delta}^{j,\ell-1}_{\mathrm{safe}}(z)) + \mathfrak{p}(\widehat{\Delta}^{\ell-1}(z)) + \epsilon_\ell \right) + \sqrt{\frac{\|z\|^2_{A(\lambda_\ell)^{-1}} \cdot \log(\frac{4m|\mathcal{Z}|\ell^2}{\delta})}{\tau_\ell}} \leq c_e \epsilon_\ell.$$

This implies that, for all $z \in \mathcal{Z}$,

$$\sqrt{\frac{\|z\|^2_{A(\lambda_\ell)^{-1}} \cdot \log(\frac{4m|\mathcal{Z}|\ell^2}{\delta})}{\tau_\ell}} \leq \min_j c_d|\widehat{\Delta}^{j,\ell-1}_{\mathrm{safe}}(z)| + \max_j c_d\mathfrak{p}(-\widehat{\Delta}^{j,\ell-1}_{\mathrm{safe}}(z)) + c_d\mathfrak{p}(\widehat{\Delta}^{\ell-1}(z)) + (c_d + c_e)\epsilon_\ell.$$

On $\mathcal{E}_{\mathrm{safe}}$, we have

$$\begin{aligned} |\widehat{\Delta}^{i,\ell}_{\mathrm{safe}}(z) - \Delta^i_{\mathrm{safe}}(z)| &\leq \sqrt{8\frac{\|z\|^2_{A(\lambda_\ell)^{-1}} \cdot \log(\frac{4m|\mathcal{Z}|\ell^2}{\delta})}{\tau_\ell}} \\ &\leq \min_j 3c_d|\widehat{\Delta}^{j,\ell-1}_{\mathrm{safe}}(z)| + \max_j 3c_d\mathfrak{p}(-\widehat{\Delta}^{j,\ell-1}_{\mathrm{safe}}(z)) + 3c_d\mathfrak{p}(\widehat{\Delta}^{\ell-1}(z)) + 3(c_d + c_e)\epsilon_\ell \end{aligned}$$

which proves the result. $\qquad\square$

## D.4 Tying Together Safety Estimation with Optimality Estimation

**Definition D.1** (Optimality Good Event). *Let* $\mathcal{E}^\ell_{\mathrm{RAGE}^\epsilon}$ *denote the success event of* RAGE$^\epsilon$ *when called at the* $\ell$*th epoch, and* $\mathcal{E}_{\mathrm{RAGE}^\epsilon} := \cup_\ell \mathcal{E}^\ell_{\mathrm{RAGE}^\epsilon}$.

**Lemma 17.** *On the event* $\mathcal{E}_{\mathrm{safe}} \cap \mathcal{E}_{\mathrm{RAGE}^\epsilon}$, *we have that:*

1. *For all* $\ell \leq \iota_\epsilon$, $y \in \mathcal{Y}_\ell$, *and* $i \in [m]$, $y^\top \mu_{*,i} \leq \gamma$.
2. *For all* $\ell$ *and* $z \in \mathcal{Z}$, $\widetilde{\Delta}^{\ell-1}(z) \leq \widetilde{\Delta}^\ell(z)$.

*Proof.* By Lemma 16, we have that

$$\widehat{\Delta}^{i,\ell}_{\mathrm{safe}}(z) - 3c_d \left( \min_j |\widehat{\Delta}^{j,\ell-1}_{\mathrm{safe}}(z)| + \max_j \mathfrak{p}(-\widehat{\Delta}^{j,\ell-1}_{\mathrm{safe}}(z)) + \mathfrak{p}(\widehat{\Delta}^{\ell-1}(z)) \right) - 3(c_d + c_e)\epsilon_\ell \leq \Delta^i_{\mathrm{safe}}(z).$$

Thus, if the inclusion condition of $\mathcal{Y}_\ell$ is met, it must be the case that $\Delta^i_{\mathrm{safe}}(z) \geq 0$ for all $i$.

The second conclusion follows directly since $\mathcal{Y}_{\ell-1} \subseteq \mathcal{Y}_\ell$. $\qquad\square$

**Lemma 18** (Key Estimation Error Bound). *On the event $\mathcal{E}_{\text{safe}} \cap \mathcal{E}_{\text{RAGE}\epsilon}$, for all $z \in \mathcal{Z}$, $\ell$, and $i$, we have*

$$|\widehat{\Delta}^\ell(z) - \widetilde{\Delta}^\ell(z)| \leq c_3 \left( \epsilon_\ell + \mathfrak{p}(\widetilde{\Delta}^\ell(z)) + \max_j \mathfrak{p}(-\Delta^j_{\text{safe}}(z)) \right)$$

$$|\widehat{\Delta}^{i,\ell}_{\text{safe}}(z) - \Delta^i_{\text{safe}}(z)| \leq c_4 \left( \epsilon_\ell + \mathfrak{p}(\widetilde{\Delta}^\ell(z)) + \min_j |\Delta^j_{\text{safe}}(z)| + \max_j \mathfrak{p}(-\Delta^j_{\text{safe}}(z)) \right).$$

*Proof.* We prove this by induction. Assume that the above inequalities hold at epoch $\ell - 1$. On $\mathcal{E}_{\text{safe}} \cap \mathcal{E}_{\text{RAGE}\epsilon}$, by Lemma 12 and Lemma 16, we have

$$|\widehat{\Delta}^\ell(z) - \widetilde{\Delta}^\ell(z)| \leq c_1(\epsilon_\ell + \mathfrak{p}(\widetilde{\Delta}^\ell(z)) + \max_j \mathfrak{p}(-\widehat{\Delta}^{j,\ell-1}_{\text{safe}}(z)))$$

$$|\widehat{\Delta}^{i,\ell}_{\text{safe}}(z) - \Delta^i_{\text{safe}}(z)| \leq c_2(\epsilon_\ell + \mathfrak{p}(\widehat{\Delta}^{\ell-1}(z)) + \min_j |\widehat{\Delta}^{j,\ell-1}_{\text{safe}}(z)| + \max_j \mathfrak{p}(-\widehat{\Delta}^{j,\ell-1}_{\text{safe}}(z))).$$

By the inductive hypothesis, we can bound

$$\mathfrak{p}(\widehat{\Delta}^{\ell-1}(z)) \leq \mathfrak{p}\left( \widetilde{\Delta}^{\ell-1}(z) + c_3(\epsilon_{\ell-1} + \mathfrak{p}(\widetilde{\Delta}^{\ell-1}(z)) + \max_j \mathfrak{p}(-\Delta^j_{\text{safe}}(z))) \right)$$

$$\leq (1 + c_3)\mathfrak{p}(\widetilde{\Delta}^{\ell-1}(z)) + c_3\epsilon_{\ell-1} + \max_j c_3\mathfrak{p}(-\Delta^j_{\text{safe}}(z))$$

$$\leq (1 + c_3)\mathfrak{p}(\widetilde{\Delta}^\ell(z)) + 2c_3\epsilon_\ell + \max_j c_3\mathfrak{p}(-\Delta^j_{\text{safe}}(z))$$

where the last inequality follows since, by Lemma 17, $\widetilde{\Delta}^{\ell-1}(z) \leq \widetilde{\Delta}^\ell(z)$.

Furthermore, again applying the inductive hypothesis,

$$\widehat{\Delta}^{i,\ell-1}_{\text{safe}}(z) \leq \Delta^i_{\text{safe}}(z) + c_4(\epsilon_{\ell-1} + \mathfrak{p}(\widetilde{\Delta}^{\ell-1}(z)) + \min_j |\Delta^j_{\text{safe}}(z)| + \max_j \mathfrak{p}(-\Delta^j_{\text{safe}}(z)))$$

$$\leq \Delta^i_{\text{safe}}(z) + 2c_4\epsilon_\ell + c_4\mathfrak{p}(\widetilde{\Delta}^\ell(z)) + \min_j c_4|\Delta^j_{\text{safe}}(z)| + \max_j c_4\mathfrak{p}(-\Delta^j_{\text{safe}}(z))$$

$$\leq \Delta^i_{\text{safe}}(z) + 2c_4\epsilon_\ell + c_4\mathfrak{p}(\widetilde{\Delta}^\ell(z)) + c_4|\Delta^i_{\text{safe}}(z)| + \max_j c_4\mathfrak{p}(-\Delta^j_{\text{safe}}(z)).$$

Similarly,

$$\mathfrak{p}(-\widehat{\Delta}^{i,\ell-1}_{\text{safe}}(z)) \leq \mathfrak{p}\left( -\Delta^i_{\text{safe}}(z) + 2c_4\epsilon_\ell + c_4\mathfrak{p}(\widetilde{\Delta}^\ell(z)) + \min_j c_4|\Delta^j_{\text{safe}}(z)| + \max_j c_4\mathfrak{p}(-\Delta^j_{\text{safe}}(z)) \right)$$

$$\leq \mathfrak{p}\left( -\Delta^i_{\text{safe}}(z) + \min_j c_4|\Delta^j_{\text{safe}}(z)| \right) + 2c_4\epsilon_\ell + c_4\mathfrak{p}(\widetilde{\Delta}^\ell(z)) + \max_j c_4\mathfrak{p}(-\Delta^j_{\text{safe}}(z))$$

$$\leq \mathfrak{p}\left( -\Delta^i_{\text{safe}}(z) + c_4|\Delta^i_{\text{safe}}(z)| \right) + 2c_4\epsilon_\ell + c_4\mathfrak{p}(\widetilde{\Delta}^\ell(z)) + \max_j c_4\mathfrak{p}(-\Delta^j_{\text{safe}}(z)).$$

Note that if $\Delta^i_{\text{safe}}(z) \leq 0$, then

$$\mathfrak{p}(-\Delta^i_{\text{safe}}(z) + c_4|\Delta^i_{\text{safe}}(z)|) = \mathfrak{p}(-\Delta^i_{\text{safe}}(z) - c_4\Delta^i_{\text{safe}}(z)) = (1 + c_4)\mathfrak{p}(-\Delta^i_{\text{safe}}(z))$$

and if $\Delta^i_{\text{safe}}(z) > 0$, then for $c_4 < 1$, $-\Delta^i_{\text{safe}}(z) + c_4|\Delta^i_{\text{safe}}(z)| \leq 0$, so

$$\mathfrak{p}(-\Delta^i_{\text{safe}}(z) + c_4|\Delta^i_{\text{safe}}(z)|) = 0 = (1 + c_4)\mathfrak{p}(-\Delta^i_{\text{safe}}(z)).$$

Thus,

$$\mathfrak{p}(-\widehat{\Delta}^{i,\ell-1}_{\text{safe}}(z)) \leq (1 + c_4)\mathfrak{p}(-\Delta^i_{\text{safe}}(z)) + 2c_4\epsilon_\ell + c_4\mathfrak{p}(\widetilde{\Delta}^\ell(z)) + \max_j c_4\mathfrak{p}(-\Delta^j_{\text{safe}}(z)).$$

Combining these inequalities, it follows that

$$|\widehat{\Delta}^{i,\ell}_{\text{safe}}(z) - \Delta^i_{\text{safe}}(z)| \leq c_2 \left( \epsilon_\ell + \mathfrak{p}(\widehat{\Delta}^{\ell-1}(z)) + \min_j |\widehat{\Delta}^{j,\ell-1}_{\text{safe}}(z)| + \max_j \mathfrak{p}(-\widehat{\Delta}^{j,\ell-1}_{\text{safe}}(z)) \right)$$

$$\leq c_2(1 + 2c_3 + 4c_4)\epsilon_\ell + c_2(1 + c_3 + 2c_4)\mathfrak{p}(\widetilde{\Delta}^\ell(z))$$

$$+ c_2(1 + c_3 + 3c_4) \max_j \mathfrak{p}(-\Delta_{\text{safe}}^j(z)) + c_2(1 + c_4) \min_j |\Delta_{\text{safe}}^j(z)|$$

and

$$|\widehat{\Delta}^\ell(z) - \widetilde{\Delta}^\ell(z)| \leq c_1(\epsilon_\ell + \mathfrak{p}(\widetilde{\Delta}^\ell(z)) + \max_j \mathfrak{p}(-\widehat{\Delta}_{\text{safe}}^{j,\ell-1}(z)))$$

$$\leq c_1(1 + 2c_4)\epsilon_\ell + c_1(1 + c_4)\mathfrak{p}(\widetilde{\Delta}^\ell(z)) + c_1(1 + 2c_4) \max_j \mathfrak{p}(-\Delta_{\text{safe}}^j(z)).$$

This proves the inductive hypothesis, as long as

$$c_1(1 + 2c_4) \leq c_3, \quad c_2(1 + 2c_3 + 4c_4) \leq c_4. \tag{11}$$

For the base case, we need to show that

$$|\widehat{\Delta}^0(z) - \widetilde{\Delta}^0(z)| \leq c_3(\epsilon_0 + \mathfrak{p}(\widetilde{\Delta}^0(z)) + \max_j \mathfrak{p}(-\Delta_{\text{safe}}^j(z)))$$

$$|\widehat{\Delta}_{\text{safe}}^0(z) - \Delta_{\text{safe}}(z)| \leq c_4(\epsilon_0 + \mathfrak{p}(\widetilde{\Delta}^0(z)) + \min_j |\Delta_{\text{safe}}^j(z)| + \max_j \mathfrak{p}(-\Delta_{\text{safe}}^j(z))).$$

By construction, $\widehat{\Delta}^0(z) = \widehat{\Delta}_{\text{safe}}^0(z) = 0$. Thus, it suffices to show $|\widetilde{\Delta}^0(z)| \leq c_3\epsilon_0$ and $|\Delta_{\text{safe}}(z)| \leq c_4\epsilon_0$. However, both of these are true by our choice of $\epsilon_0$. $\qquad\square$

**Lemma 19.** *On the event $\mathcal{E}_{\text{safe}} \cap \mathcal{E}_{\text{RAGE}^\epsilon}$, for all $z \in \mathcal{Z}$ and all $\ell$, we will have*

$$\widetilde{\Delta}^\ell(z) \geq \Delta^{\epsilon_\ell}(z) \quad \text{where} \quad \Delta^{\epsilon_\ell}(z) = \max_{y \in \mathcal{Z} \,:\, \epsilon_\ell \leq \min_i \Delta_{\text{safe}}^i(y)} y^\top \theta_* - z^\top \theta_*.$$

*Proof.* By definition, we will have $z \in \mathcal{Y}_\ell$ if

$$8c_d \left( \min_j |\widehat{\Delta}_{\text{safe}}^{j,\ell-1}(z)| + \max_j \mathfrak{p}(-\widehat{\Delta}_{\text{safe}}^{j,\ell-1}(z)) + \mathfrak{p}(\widehat{\Delta}^{\ell-1}(z)) \right) + 8(c_d + c_e)\epsilon_\ell \leq \widehat{\Delta}_{\text{safe}}^{i,\ell}(z).$$

The following claim allows us to obtain a sufficient condition to guarantee $z \in \mathcal{Y}_\ell$.

**Claim D.1.** *On the event $\mathcal{E}_{\text{safe}} \cap \mathcal{E}_{\text{RAGE}^\epsilon}$,*

$$\min_j |\widehat{\Delta}_{\text{safe}}^{j,\ell-1}(z)| + \max_j \mathfrak{p}(-\widehat{\Delta}_{\text{safe}}^{j,\ell-1}(z)) + \mathfrak{p}(\widehat{\Delta}^{\ell-1}(z))$$

$$\leq 2(c_3 + 2c_4)\epsilon_\ell + (1 + c_3 + 2c_4)\mathfrak{p}(\widetilde{\Delta}^\ell(z)) + (1 + 2c_4) \min_j |\Delta_{\text{safe}}^j(z)| + (1 + c_3 + 2c_4) \max_j \mathfrak{p}(-\Delta_{\text{safe}}^j(z)).$$

*Proof of Claim D.1.* By Lemma 17 and Lemma 18, we can bound

$$\min_j |\widehat{\Delta}_{\text{safe}}^{j,\ell-1}(z)| \leq (1 + c_4) \min_j |\Delta_{\text{safe}}^j(z)| + 2c_4\epsilon_\ell + c_4\mathfrak{p}(\widetilde{\Delta}^\ell(z)) + c_4 \max_j \mathfrak{p}(-\Delta_{\text{safe}}^j(z))$$

$$\max_j \mathfrak{p}(-\widehat{\Delta}_{\text{safe}}^{j,\ell-1}(z)) \leq (1 + c_4) \max_j \mathfrak{p}(-\Delta_{\text{safe}}^j(z)) + 2c_4\epsilon_\ell + c_4\mathfrak{p}(\widetilde{\Delta}^\ell(z)) + c_4 \min_j |\Delta_{\text{safe}}^j(z)|$$

$$\mathfrak{p}(\widehat{\Delta}^{\ell-1}(z)) \leq (1 + c_3)\mathfrak{p}(\widetilde{\Delta}^\ell(z)) + 2c_3\epsilon_\ell + c_3 \max_j \mathfrak{p}(-\Delta_{\text{safe}}^j(z)).$$

The claim follows by summing these upper bounds. $\qquad\square$

Thus, by Claim D.1, we can bound

$$3c_d \left( \min_j |\widehat{\Delta}_{\text{safe}}^{j,\ell-1}(z)| + \max_j \mathfrak{p}(-\widehat{\Delta}_{\text{safe}}^{j,\ell-1}(z)) + \mathfrak{p}(\widehat{\Delta}^{\ell-1}(z)) \right) + 3(c_d + c_e)\epsilon_\ell$$

$$\leq 3(c_d + c_e + 2c_dc_3 + 4c_dc_4)\epsilon_\ell + 3c_d(1 + c_3 + 2c_4)\mathfrak{p}(\widetilde{\Delta}^\ell(z))$$

$$+ 3c_d(1 + 2c_4) \min_j |\Delta_{\text{safe}}^j(z)| + 3c_d(1 + c_3 + 2c_4) \max_j \mathfrak{p}(-\Delta_{\text{safe}}^j(z)).$$

Furthermore, by Lemma 18,

$$\Delta_{\text{safe}}^i(z) - c_4 \left( \epsilon_\ell + \mathfrak{p}(\widetilde{\Delta}^\ell(z)) + \min_j |\Delta_{\text{safe}}^j(z)| + \max_j \mathfrak{p}(-\Delta_{\text{safe}}^j(z)) \right) \leq \widehat{\Delta}_{\text{safe}}^{i,\ell}(z)$$

It follows that a sufficient condition for $z \in \mathcal{Y}_\ell$ is

$$
(3c_d + 3c_e + 6c_dc_3 + 12c_dc_4 + c_4)\left(\epsilon_\ell + \mathfrak{p}(\widetilde{\Delta}^\ell(z)) + \min_j |\Delta^j_{\text{safe}}(z)| + \max_j \mathfrak{p}(-\Delta^j_{\text{safe}}(z))\right)
$$
$$
\leq \Delta^i_{\text{safe}}(z), \quad \forall i \in [m].
$$
(12)

If $y_\ell = \arg\max_{y \in \mathcal{Z} \, : \, \epsilon_\ell \leq \min_i \Delta^i_{\text{safe}}(y)} y^\top \theta_*$ is in $\mathcal{Y}_\ell$, then we are done. Assume then that $y_\ell \notin \mathcal{Y}_\ell$. By construction, since $\Delta^i_{\text{safe}}(y_\ell) > 0$ for all $i$, $\max_j \mathfrak{p}(-\Delta^j_{\text{safe}}(z)) = 0$. Using that (12) is a sufficient condition for inclusion in $\mathcal{Y}_\ell$, this implies that

$$
\exists i \in [m] \quad \text{s.t.} \quad (3c_d + 3c_e + 6c_dc_3 + 12c_dc_4 + c_4)\left(\epsilon_\ell + \mathfrak{p}(\widetilde{\Delta}^\ell(y_\ell)) + \min_j |\Delta^j_{\text{safe}}(y_\ell)|\right) > \Delta^i_{\text{safe}}(y_\ell).
$$

which implies

$$
\exists i \in [m] \quad \text{s.t.} \quad (3c_d + 3c_e + 6c_dc_3 + 12c_dc_4 + c_4)\left(\epsilon_\ell + \mathfrak{p}(\widetilde{\Delta}^\ell(y_\ell)) + |\Delta^i_{\text{safe}}(y_\ell)|\right) > \Delta^i_{\text{safe}}(y_\ell).
$$
(13)

By construction, though, $\Delta^i_{\text{safe}}(y_\ell) \geq \epsilon_\ell$. If we assume that

$$
3c_d + 3c_e + 6c_dc_3 + 12c_dc_4 + c_4 \leq 1/4,
$$
(14)

then (13) can only hold if $\mathfrak{p}(\widetilde{\Delta}^\ell(y_\ell)) > 0$. This implies that $\max_{y \in \mathcal{Y}_\ell} y^\top \theta_* > y_\ell^\top \theta_*$. Thus, in this case,

$$
\widetilde{\Delta}^\ell(z) = \max_{y \in \mathcal{Y}_\ell} y^\top \theta_* - z^\top \theta_* > y_\ell^\top \theta_* - z^\top \theta_* = \Delta^{\epsilon_\ell}(z)
$$

which proves the result. $\qquad\square$

**Lemma 20.** *On $\mathcal{E}_{\text{safe}} \cap \mathcal{E}_{\text{RAGE}^\epsilon}$, for all $z \in \mathcal{Y}_{\text{end}}$ we have*

$$
\Delta^i_{\text{safe}}(z) \geq -c_g\epsilon, \quad \forall i \in [m],
$$
$$
\widehat{\Delta}^{i,\iota_\epsilon}_{\text{safe}}(z) \geq (3c_d + 3c_e - c_g)\epsilon, \quad \forall i \in [m].
$$

*Furthermore, $z_* \in \mathcal{Y}_{\text{end}}$.*

*Proof.* Recall that

$$
\mathcal{Y}_{\text{end}} = \{z \in \mathcal{Z} \, : 3c_d \left(\min_j |\widehat{\Delta}^{j,\iota_\epsilon}_{\text{safe}}(z)| + \max_j \mathfrak{p}(-\widehat{\Delta}^{j,\iota_\epsilon}_{\text{safe}}(z)) + \mathfrak{p}(\widehat{\Delta}^{\iota_\epsilon}(z))\right)
$$
$$
+ 3(c_d + c_e)\epsilon - c_g\epsilon \leq \widehat{\Delta}^{i,\iota_\epsilon}_{\text{safe}}(z), \forall i \in [m]\}
$$

On $\mathcal{E}_{\text{safe}}$, we have

$$
\widehat{\Delta}^{i,\iota_\epsilon}_{\text{safe}}(z) \leq \Delta^i_{\text{safe}}(z) + 3c_d \left(\min_j |\widehat{\Delta}^{j,\iota_\epsilon}_{\text{safe}}(z)| + \max_j \mathfrak{p}(-\widehat{\Delta}^{j,\iota_\epsilon}_{\text{safe}}(z)) + \mathfrak{p}(\widehat{\Delta}^{\iota_\epsilon}(z))\right) + 3(c_d + c_e)\epsilon
$$

so it follows that if $z \in \mathcal{Y}_{\text{end}}$, then

$$
-c_g\epsilon \leq \Delta^i_{\text{safe}}(x).
$$

To see that $z_* \in \mathcal{Y}_{\text{end}}$, note that by definition of $\mathcal{Y}_{\text{end}}$, using a calculation analogous to (12), a sufficient condition for $z \in \mathcal{Y}_{\text{end}}$ is

$$
(3c_d + 3c_e + 6c_dc_3 + 12c_dc_4 + c_4 - c_g)\epsilon + (3c_d + 3c_dc_3 + 6c_dc_4 + c_4)\mathfrak{p}(\widetilde{\Delta}^{\iota_\epsilon}(z))
$$
$$
+ (3c_d + 6c_dc_4 + c_4)\min_j |\Delta^j_{\text{safe}}(z)| + (3c_d + 3c_dc_3 + 6c_dc_4 + c_4)\max_j \mathfrak{p}(-\Delta^j_{\text{safe}}(z))
$$
$$
\leq \Delta^i_{\text{safe}}(z), \quad \forall i \in [m].
$$

By definition of $z_*$ and since, by Lemma 17, all $z \in \mathcal{Y}_{\iota_\epsilon}$ are safe, we have $\Delta^{\epsilon_{\iota_\epsilon}}(z_*) \leq 0$. Furthermore, by definition we also have $\Delta_{\text{safe}}^j(z_*) \geq 0$ for all $j$, so $\mathfrak{p}(-\Delta_{\text{safe}}^j(z_*)) = 0$. Thus, assuming that

$$3c_d + 3c_e + 6c_dc_3 + 12c_dc_4 + c_4 - c_g \leq 0 \tag{15}$$

a sufficient condition to guarantee $z_* \in \mathcal{Y}_{\text{end}}$ is that

$$(8c_d + 16c_dc_4 + c_4)\min_j |\Delta_{\text{safe}}^j(z_*)| \leq \Delta_{\text{safe}}^i(z_*), \quad \forall i \in [m].$$

However, as long as

$$3c_d + 6c_dc_4 + c_4 \leq 1, \tag{16}$$

this is true, since by definition $\Delta_{\text{safe}}^i(z_*) \geq 0$. $\qquad\square$

## D.5 Algorithm Correctness and Sample Complexity

**Lemma 21** (Correctness). *On $\mathcal{E}_{\text{safe}} \cap \mathcal{E}_{\text{RAGE}^\epsilon}$, we will have that*

$$\widehat{z}^\top \theta_* \geq (z_*)^\top \theta_* - \frac{c_3(1+c_g)}{1-c_3}\epsilon, \quad \Delta_{\text{safe}}^i(\widehat{z}) \geq -c_g\epsilon, \forall i \in [m].$$

*Proof.* We choose $\widehat{z}$ to be any $z \in \mathcal{Y}_{\text{end}}$ such that $\widehat{\Delta}^{\text{end}}(z) = 0$. By Lemma 20, we have that $\Delta_{\text{safe}}^i(\widehat{z}) \geq -c_g\epsilon$ for all $i \in [m]$. If $\widetilde{\Delta}^{\text{end}}(\widehat{z}) \leq 0$, we are done, since by Lemma 20, $z_* \in \mathcal{Y}_{\text{end}}$, so $\widehat{z}^\top \theta_* \geq (z_*)^\top \theta_*$. Assume that $\widetilde{\Delta}^{\text{end}}(\widehat{z}) > 0$. By Lemma 18, we have that

$$\widetilde{\Delta}^{\text{end}}(\widehat{z}) \leq c_3\epsilon + c_3\mathfrak{p}(\widetilde{\Delta}^{\text{end}}(\widehat{z})) + c_3\max_j \mathfrak{p}(-\Delta_{\text{safe}}^j(\widehat{z})).$$

By Lemma 20, since $\widehat{z} \in \mathcal{Y}_{\text{end}}$, $\mathfrak{p}(-\Delta_{\text{safe}}^j(\widehat{z})) \leq c_g\epsilon$ for all $j$, so we can bound

$$\widetilde{\Delta}^{\text{end}}(\widehat{z}) \leq c_3(1+c_g)\epsilon + c_3\mathfrak{p}(\widetilde{\Delta}^{\text{end}}(\widehat{z})) = c_3(1+c_g)\epsilon + c_3\widetilde{\Delta}^{\text{end}}(\widehat{z}).$$

We can rearrange this as

$$\widetilde{\Delta}^{\text{end}}(\widehat{z}) \leq \frac{c_3(1+c_g)}{1-c_3}\epsilon$$

which proves the result, since, by Lemma 20, $\Delta^{\text{end}}(\widehat{z}) = \max_{y \in \mathcal{Y}_{\text{end}}} y^\top \theta_* - \widehat{z}^\top \theta_* \geq (z_*)^\top \theta_* - \widehat{z}^\top \theta_*$. $\qquad\square$

**Lemma 22.** *On $\mathcal{E}_{\text{RAGE}^\epsilon} \cap \mathcal{E}_{\text{safe}}$, the total complexity of Algorithm 1 is bounded by*

$$C \cdot \sum_{\ell=1}^{\iota_\epsilon} \inf_{\lambda \in \triangle_\mathcal{X}} \max_{z \in \mathcal{Z}} \frac{\|z\|_{A(\lambda)^{-1}}^2 \cdot \log(\frac{4m|\mathcal{Z}|\ell^2}{\delta})}{\left(\min_j |\Delta_{\text{safe}}^j(z)| + \max_j \mathfrak{p}(-\Delta_{\text{safe}}^j(z)) + \mathfrak{p}(\Delta^{\epsilon_{\ell-1}}(z)) + \epsilon_\ell\right)^2} + 4\iota_\epsilon \log(\frac{4m|\mathcal{Z}|\iota_\epsilon^2}{\delta})$$

*for an absolute constant $C$.*

*Proof.* Applying the same argument as in Claim D.1 but in the opposite direction, we have

$$\min_j |\widehat{\Delta}_{\text{safe}}^{j,\ell-1}(z)| + \max_j \mathfrak{p}(-\widehat{\Delta}_{\text{safe}}^{j,\ell-1}(z)) + \mathfrak{p}(\widehat{\Delta}^{\ell-1}(z))$$

$$\geq -2(c_3 + 2c_4)\epsilon_\ell + (1 - c_3 - 2c_4)\mathfrak{p}(\widetilde{\Delta}^\ell(z)) + (1 - 2c_4)\min_j |\Delta_{\text{safe}}^j(z)| + (1 - c_3 - 2c_4)\max_j \mathfrak{p}(-\Delta_{\text{safe}}^j(z)).$$

We assume that $c_3$, $c_4$, and $c_0$ are chosen such that

$$1 - 2c_3 - 4c_4 \geq c_0, \tag{17}$$

which allows us to bound:

$$\inf_{\lambda \in \triangle_\mathcal{X}} \max_{z \in \mathcal{Z}} -c_d \left(\min_j |\widehat{\Delta}_{\text{safe}}^{j,\ell-1}(z)| + \max_j \mathfrak{p}(-\widehat{\Delta}_{\text{safe}}^{j,\ell-1}(z)) + \mathfrak{p}(\widehat{\Delta}^{\ell-1}(z)) + \epsilon_\ell\right) + \sqrt{\frac{\|z\|_{A(\lambda)^{-1}}^2 \cdot \log(\frac{4m|\mathcal{Z}|\ell^2}{\delta})}{\tau}}$$

$$\leq \inf_{\lambda \in \triangle_{\mathcal{X}}} \max_{z \in \mathcal{Z}} -c_d c_0 \left( \min_j |\Delta^j_{\text{safe}}(z)| + \max_j \mathfrak{p}(-\Delta^j_{\text{safe}}(z)) + \mathfrak{p}(\widetilde{\Delta}^{\ell-1}(z)) + \epsilon_\ell \right) + \sqrt{\frac{\|z\|^2_{A(\lambda)^{-1}} \cdot \log(\frac{4m|\mathcal{Z}|\ell^2}{\delta})}{\tau}}.$$

It follows that if, for all $z \in \mathcal{Z}$,

$$\tau \geq \frac{\|z\|^2_{A(\lambda)^{-1}}}{\left( c_d c_0 \min_j |\Delta^j_{\text{safe}}(z)| + c_d c_0 \max_j \mathfrak{p}(-\Delta^j_{\text{safe}}(z)) + c_d c_0 \mathfrak{p}(\widetilde{\Delta}^{\ell-1}(z)) + (c_d c_0 + c_e)\epsilon_\ell \right)^2} \cdot \log \frac{4m|\mathcal{Z}|\ell^2}{\delta}$$

we will have that this is less than $c_e \epsilon_\ell$. Since we can take the best-case $\lambda \in \triangle_{\mathcal{X}}$, and since $\tau_\ell$ is always within a factor of 2 of the optimal, it follows that

$$\tau_\ell \leq \inf_{\lambda \in \triangle_{\mathcal{X}}} \max_{z \in \mathcal{Z}} \frac{2\|z\|^2_{A(\lambda)^{-1}} \cdot \log \frac{4m|\mathcal{Z}|\ell^2}{\delta}}{\left( c_d c_0 \min_j |\Delta^j_{\text{safe}}(z)| + c_d c_0 \max_j \mathfrak{p}(-\Delta^j_{\text{safe}}(z)) + c_d c_0 \mathfrak{p}(\widetilde{\Delta}^{\ell-1}(z)) + (c_d c_0 + c_e)\epsilon_\ell \right)^2}$$
$$+ 4 \log \frac{4m|\mathcal{Z}|\ell^2}{\delta}$$

The result then follows by summing over epochs and lower bounding $\widetilde{\Delta}^{\ell-1}(z)$ by $\Delta^{\epsilon_{\ell-1}}(z)$ using Lemma 19, and assuming that

$$c_d c_0 + c_e \geq c_0. \tag{18}$$

$\square$

*Proof of Theorem 14.* By Lemma 15 we have that $\mathcal{E}_{\text{safe}}$ holds with probability at least $1 - \delta$. By Lemma 9, we have that $\mathcal{E}^\ell_{\text{RAGE}^\epsilon}$ holds with probability at least $1 - \delta/(4\ell^2)$. It follows then that $\mathcal{E}_{\text{safe}} \cup (\cup_\ell \mathcal{E}^\ell_{\text{RAGE}^\epsilon})$ holds with probability at least

$$1 - \delta - \sum_\ell \frac{\delta}{4\ell^2} \geq 1 - 2\delta.$$

Assume henceforth that $\mathcal{E}_{\text{safe}} \cup (\cup_\ell \mathcal{E}^\ell_{\text{RAGE}^\epsilon})$ holds. Equation (10) follows by Lemma 21. The total number of samples collected on Algorithm 1 can be bounded by Lemma 22. It remains to bound the total number of samples used by $\text{RAGE}^\epsilon$.

By Lemma 13, at epoch $\ell$ $\text{RAGE}^\epsilon$ will collect at most

$$C \lceil \log \frac{2}{c_f \epsilon_\ell} \rceil \cdot \inf_{\lambda \in \triangle_{\mathcal{X}}} \max_{z \in \mathcal{Z}} \frac{\|z - y^\ell_\star\|^2_{A(\lambda)^{-1}} \cdot \log(\frac{8|\mathcal{Z}|^2 \log^4(1/\epsilon)}{\delta})}{(\max_j \mathfrak{p}(-\widehat{\Delta}^{j,\ell-1}_{\text{safe}}(z)) + \mathfrak{p}(\widetilde{\Delta}^\ell(z)) + \epsilon_\ell)^2} + 8 \lceil \log \frac{2}{c_f \epsilon_\ell} \rceil \log(\frac{4|\mathcal{Z}|^2 \lceil \log \frac{2}{c_f \epsilon} \rceil^2}{\delta})$$

samples, where $y^\ell_\star = \arg \max_{y \in \mathcal{Y}_\ell} y^\top \theta_\star$. Assume that $\max_j \mathfrak{p}(-\Delta^j_{\text{safe}}(z)) > 0$, then we can upper bound $\min_j |\Delta^j_{\text{safe}}(z)| \leq \max_j \mathfrak{p}(-\Delta^j_{\text{safe}}(z))$, and by Lemma 18 we can lower bound

$$\max_j \mathfrak{p}(-\widehat{\Delta}^{j,\ell-1}_{\text{safe}}(z)) \geq (1 - 2c_4) \max_j \mathfrak{p}(-\Delta^j_{\text{safe}}(z)) - c_4 \mathfrak{p}(\widetilde{\Delta}^{\ell-1}(z)) - c_4 \epsilon_{\ell-1}.$$

Assume instead that $\max_j \mathfrak{p}(-\Delta^j_{\text{safe}}(z)) = 0$. Then again by Lemma 18,

$$\max_j \mathfrak{p}(-\widehat{\Delta}^{j,\ell-1}_{\text{safe}}(z)) \geq 0 = \max_j \mathfrak{p}(-\Delta^j_{\text{safe}}(z)) \geq (1 - 2c_4) \max_j \mathfrak{p}(-\Delta^j_{\text{safe}}(z)) - c_4 \mathfrak{p}(\widetilde{\Delta}^{\ell-1}(z)) - c_4 \epsilon_{\ell-1}.$$

By Lemma 17, it follows that

$$\max_j \mathfrak{p}(-\widehat{\Delta}^{j,\ell-1}_{\text{safe}}(z)) + \mathfrak{p}(\widetilde{\Delta}^\ell(z)) + \epsilon_\ell$$
$$\geq (1 - 2c_4) \max_j \mathfrak{p}(-\Delta^j_{\text{safe}}(z)) + (1 - c_4)\mathfrak{p}(\widetilde{\Delta}^\ell(z)) + (1 - 2c_4)\epsilon_\ell.$$

By definition and Lemma 17 and Lemma 20 for all $\ell$ including $\ell = \text{end}$, we can bound $\mathfrak{p}(-\Delta^j_{\text{safe}}(y^\ell_\star)) \leq c_g \epsilon$. Furthermore, by definition $\mathfrak{p}(\widetilde{\Delta}^\ell(y^\ell_\star)) = 0$. Putting all of this together, we have:

$$\inf_{\lambda \in \triangle_{\mathcal{X}}} \max_{z \in \mathcal{Z}} \frac{\|z - y^\ell_\star\|^2_{A(\lambda)^{-1}} \cdot \log(\frac{8|\mathcal{Z}|^2 \log^4(1/\epsilon)}{\delta})}{(\max_j \mathfrak{p}(-\widehat{\Delta}^{j,\ell-1}_{\text{safe}}(z)) + \mathfrak{p}(\widetilde{\Delta}^\ell(z)) + \epsilon_\ell)^2}$$

$$\leq \inf_{\lambda \in \triangle_{\mathcal{X}}} \max_{z \in \mathcal{Z}} \frac{\|z - y_\star^\ell\|^2_{A(\lambda)^{-1}} \cdot \log(\frac{8|\mathcal{Z}|^2 \log^4(1/\epsilon)}{\delta})}{((1 - 2c_4) \max_j \mathfrak{p}(-\Delta^j_{\text{safe}}(z)) + (1 - c_4)\mathfrak{p}(\widetilde{\Delta}^\ell(z)) + (1 - 2c_4)\epsilon_\ell)^2}$$

$$\leq \inf_{\lambda \in \triangle_{\mathcal{X}}} \max_{z \in \mathcal{Z}} \frac{2\|z - z_*\|^2_{A(\lambda)^{-1}} \cdot \log(\frac{8|\mathcal{Z}|^2 \log^4(1/\epsilon)}{\delta})}{((1 - 2c_4) \max_j \mathfrak{p}(-\Delta^j_{\text{safe}}(z)) + (1 - c_4)\mathfrak{p}(\widetilde{\Delta}^\ell(z)) + (1 - 2c_4)\epsilon_\ell)^2}$$

$$+ \inf_{\lambda \in \triangle_{\mathcal{X}}} \frac{2\|z_* - y_\star^\ell\|^2_{A(\lambda)^{-1}} \cdot \log(\frac{8|\mathcal{Z}|^2 \log^4(1/\epsilon)}{\delta})}{((1 - 2c_4) \max_j \mathfrak{p}(-\Delta^j_{\text{safe}}(y_\star^\ell)) + (1 - c_4)\mathfrak{p}(\widetilde{\Delta}^\ell(y_\star^\ell)) + (1 - 2c_4 - c_g)\epsilon_\ell)^2}$$

$$\leq \inf_{\lambda \in \triangle_{\mathcal{X}}} \max_{z \in \mathcal{Z}} \frac{4\|z - z_*\|^2_{A(\lambda)^{-1}} \cdot \log(\frac{8|\mathcal{Z}|^2 \log^4(1/\epsilon)}{\delta})}{((1 - 2c_4) \max_j \mathfrak{p}(-\Delta^j_{\text{safe}}(z)) + (1 - c_4)\mathfrak{p}(\widetilde{\Delta}^\ell(z)) + (1 - 2c_4 - c_g)\epsilon_\ell)^2}$$

As long as

$$1 - 2c_4 - c_g \geq c_0, \tag{19}$$

summing over the epochs and lower bounding $\widetilde{\Delta}^\ell(z)$ by $\Delta^{\epsilon_\ell}(z)$ via Lemma 19 gives the result. Finally, the settings of the constants follows from Lemma 8. $\qquad\square$

### D.6 Proofs of Corollaries to Theorem 1

*Proof of Corollary 1.* If $m = 1$, $\mu_{*,1} = 0$, and $\gamma = 1$, then we have $\Delta_{\text{safe}}(z) = 1$ for each $z$, and $\Delta^{\widetilde{\epsilon}}(z) = \Delta(z)$ for $\epsilon \leq 1$. The result follows directly from this and some algebra. $\qquad\square$

*Proof of Corollary 2.* We can trivially upper bound the complexity given in Theorem 1 by

$$C \cdot \inf_{\lambda \in \triangle_{\mathcal{X}}} \max_{z \in \mathcal{Z}} \frac{\|z\|^2_{A(\lambda)^{-1}} \cdot \log(\frac{m|\mathcal{Z}|}{\delta})}{\epsilon^2} + C \cdot \inf_{\lambda \in \triangle_{\mathcal{X}}} \max_{z \in \mathcal{Z}} \frac{\|z - z_*\|^2_{A(\lambda)^{-1}} \cdot \log(\frac{|\mathcal{Z}|}{\delta})}{\epsilon^2} + C_0$$

$$\leq C \cdot \inf_{\lambda \in \triangle_{\mathcal{X}}} \max_{z \in \mathcal{Z}} \frac{\|z\|^2_{A(\lambda)^{-1}} \cdot \log(\frac{m|\mathcal{Z}|}{\delta})}{\epsilon^2} + C_0.$$

In the case when $\mathcal{X} = \mathcal{Z}$, we can bound $\inf_{\lambda \in \triangle_{\mathcal{X}}} \max_{z \in \mathcal{Z}} \|z\|^2_{A(\lambda)^{-1}} \leq d$ by Kiefer-Wolfowitz [26], which proves the result. $\qquad\square$

## E Computationally Efficient Optimization

Throughout, we will let $\mathcal{R}(z; \xi_1, \dots, \xi_n)$ denote some generic weighted risk estimate of the form

$$\mathcal{R}(z; \xi_1, \dots, \xi_n) = \sum_{t=1}^{T} f_t(\xi_1, \dots, \xi_n) \mathbb{I}\{z(u_t) \neq v_t\}$$

for some weights $f_t(\xi_1, \dots, \xi_n)$ and observations $(u_t, v_t)$. The exact setting of $\mathcal{R}$ will change from line to line—we simply use it as a stand-in for an objective that a cost-sensitive-classification oracle can efficiently minimize. We will also use $f(\xi_1, \dots, \xi_n)$ to refer to some generic function (the particular form of which is not important).

**Lemma E.1.** *Consider some $z, \widetilde{z} \in \triangle_{\mathcal{H}}$. Denote*

$$\rho_\lambda(h, h') = \mathbb{E}_{U \sim \nu} \left[ \frac{\mathbb{I}\{h(U) \neq h'(U)\}}{\lambda(U)/\nu(U)} \right] = \|h - h'\|^2_{A(\lambda)^{-1}}$$

*and overload notation so that $z = \sum_{h \in \mathcal{H}} z_h h$ denotes the feature vector for the mixed classifier $z$. Then,*

$$\sum_{h, h' \in \mathcal{H}} z_h \widetilde{z}_{h'} \rho_\lambda(h, h') = \mathbb{E}_{U \sim \nu} \left[ \frac{(z(U) - \widetilde{z}(U))^2}{\lambda(U)/\nu(U)} \right] = \|z - \widetilde{z}\|^2_{A(\lambda)^{-1}}.$$

*Proof.* Note that

$$\rho_\lambda(h,h') = \mathbb{E}_{U\sim\nu}\left[\frac{\mathbb{I}\{h(U)\neq h'(U)\}}{\lambda(U)/\nu(U)}\right] = \mathbb{E}_{U\sim\nu}\left[\frac{|h(U)-h'(U)|}{\lambda(U)/\nu(U)}\right] = \mathbb{E}_{U\sim\nu}\left[\frac{(h(U)-h'(U))^2}{\lambda(U)/\nu(U)}\right]$$

where the final equality holds because $|h(U)-h'(U)|$ is always either 0 or 1. Thus,

$$\sum_{h,h'\in\mathcal{H}} z_h\widetilde{z}_{h'}\rho_\lambda(h,h') = \sum_{h,h'\in\mathcal{H}} z_h\widetilde{z}_{h'}\mathbb{E}_{U\sim\nu}\left[\frac{(h(U)-h'(U))^2}{\lambda(U)/\nu(U)}\right]$$

$$= \mathbb{E}_{U\sim\nu}\left[\frac{\sum_{h,h'\in\mathcal{H}} z_h\widetilde{z}_{h'}(h(U)-h'(U))^2}{\lambda(U)/\nu(U)}\right]$$

$$= \mathbb{E}_{U\sim\nu}\left[\frac{\sum_{h,h'\in\mathcal{H}} z_h\widetilde{z}_{h'}(h(U)+h'(U)-2h(U)h'(U))}{\lambda(U)/\nu(U)}\right]. \qquad (20)$$

However,

$$\sum_{h,h'\in\mathcal{H}} z_h\widetilde{z}_{h'}h(U) = \sum_{h\in\mathcal{H}} z_h h(U) = z(U), \qquad \sum_{h,h'\in\mathcal{H}} z_h\widetilde{z}_{h'}h'(U) = \widetilde{z}(U)$$

and

$$\sum_{h,h'\in\mathcal{H}} z_h\widetilde{z}_{h'}h(U)h'(U) = (\sum_{h\in\mathcal{H}} z_h h(U))(\sum_{h'\in\mathcal{H}} \widetilde{z}_{h'}h'(U)) = z(U)\widetilde{z}(U).$$

Thus,

$$(20) = \mathbb{E}_{U\sim\nu}\left[\frac{(z(U)-\widetilde{z}(U))^2}{\lambda(U)/\nu(U)}\right]$$

which proves the first equality. To prove the second, recall that $[h]_u = \nu(u)h(u)$, so $[z]_u = \sum_{h\in\mathcal{H}} z_h[h]_u = \nu(u)z(u)$. It follows that,

$$\|z-\widetilde{z}\|^2_{A(\lambda)^{-1}} = \sum_u \frac{\nu(u)^2}{\lambda(u)}(z(u)-\widetilde{z}(u))^2 = \mathbb{E}_{U\sim\nu}\left[\frac{(z(U)-\widetilde{z}(U))^2}{\lambda(U)/\nu(U)}\right]$$

which proves the second equality. $\qquad\square$

## E.1 Computational Efficiency of RAGE$^\epsilon$

RAGE$^\epsilon$ requires solving the optimization

$$\inf_{\lambda\in\triangle_\mathcal{X}}\max_{z\in\triangle_\mathcal{H}}\min_{\alpha\in\mathcal{A}} -c_a(\mathfrak{p}(-\widehat{\Delta}_{\text{safe}}(z))+\mathfrak{p}(\widehat{\Delta}^{\ell-1}(z))+\epsilon_\ell) + \alpha\|z-\widehat{y}_{\ell-1}\|^2_{A(\lambda)^{-1}} + \frac{\log(2|\mathcal{Z}|^2|\mathcal{A}|\ell^2/\delta)}{\alpha\tau}. \qquad (21)$$

Here we take $\tau$ to be fixed, and recall that

$$\widehat{y}_\ell \leftarrow \arg\min_{y\in\mathcal{Y}}\min_{\alpha\in\mathcal{A}} \widetilde{R}^\alpha_\ell(y) - \widetilde{R}^\alpha_\ell(\widehat{y}_{\ell-1}) + 2\alpha\|y-\widehat{y}_{\ell-1}\|^2_{A(\lambda_\ell)^{-1}} + \frac{2\log(2|\mathcal{Z}|^2|\mathcal{A}|\ell^2/\delta)}{\alpha\tau_\ell}$$

$$\widehat{\Delta}^\ell(y) \leftarrow \min_{\alpha\in\mathcal{A}} \widetilde{R}^\alpha_\ell(y) - \widetilde{R}^\alpha_\ell(\widehat{y}_\ell) + \alpha\|y-\widehat{y}_\ell\|^2_{A(\lambda_\ell)^{-1}} + \frac{\log(2|\mathcal{Z}|^2|\mathcal{A}|\ell^2/\delta)}{\alpha\tau_\ell}.$$

Furthermore, $\mathcal{Y}$ will be a set of the form

$$\bigcup_{k=1}^{\ell'}\mathcal{Y}_k = \bigcup_{k=1}^{\ell'}\left\{z\in\mathcal{Z} : c(\epsilon_k+\mathfrak{p}(\widehat{\Delta}^{k-1}(z))+\max_{j\in[n]}\mathfrak{p}(-\widehat{\Delta}^{j,k-1}_{\text{safe}}(z))+\min_{j\in[n]}|\widehat{\Delta}^{j,k-1}_{\text{safe}}(h)|) \leq \widehat{\Delta}^{i,k}_{\text{safe}}(h), \forall i\in[n]\right\}$$

Recall also that

$$\|h-h'\|^2_{A(\lambda)^{-1}} = \mathbb{E}_{U\sim\nu}\left[\frac{\mathbb{I}\{h(U)\neq h'(U)\}}{(9\lambda(U)/10+1/10d)/\nu(U)}\right] = \sum_{U\in\mathcal{X}}\frac{\nu(U)^2}{9\lambda(U)/10+1/10d}\mathbb{I}\{h(U)\neq h'(U)\}$$

and

$$\widetilde{R}^\alpha_\ell(h) = \frac{1}{\tau_\ell}\sum_{t=1}^{\tau_\ell}\frac{1}{w_t+\alpha}\mathbb{I}\{h(u_t)\neq v_t\}.$$

For $z\in\triangle_\mathcal{H}$, we denote $\widetilde{R}^\alpha_\ell(z) = \sum_{h\in\mathcal{H}} z_h\widetilde{R}^\alpha_\ell(h)$ and $\mathcal{R}(z;\alpha) = \sum_{h\in\mathcal{H}} z_h\mathcal{R}(h;\alpha)$. Finally, we assume that $\widehat{\Delta}_{\text{safe}}(z) = \min_{\alpha\in\mathcal{A}}\mathcal{R}(z;\alpha)+f(\alpha)$.

### E.1.1 Solving for $\widehat{y}_\ell$

Using Lemma E.1, we can write the optimization for $\widehat{y}_\ell$ as

$$\min_{k \in [\ell']} \min_{y \in \mathcal{Y}_k} \min_{\alpha \in \mathcal{A}} \frac{1}{\tau_\ell} \sum_{t=1}^{\tau_\ell} \frac{1}{w_t + \alpha} \sum_{h \in \mathcal{H}} y_h \mathbb{I}\{h(u_t) \neq v_t\} + \alpha \sum_{h,h' \in \mathcal{H}} y_h \widehat{y}_{\ell-1,h'} \sum_{U \in \mathcal{X}} \frac{\nu(U)^2}{9\lambda_\ell(U)/10 + 1/10d} \mathbb{I}\{h(U) \neq h'(U)\}$$

$$- \widetilde{R}_\ell^\alpha(\widehat{y}_{\ell-1}) + \frac{\log(2|\mathcal{Z}|^2|\mathcal{A}|\ell^2/\delta)}{\alpha \tau_\ell}$$

We can rewrite

$$\sum_{h,h' \in \mathcal{H}} y_h \widehat{y}_{\ell-1,h'} \sum_{U \in \mathcal{X}} \frac{\nu(U)^2}{9\lambda_\ell(U)/10 + 1/10d} \mathbb{I}\{h(U) \neq h'(U)\} = \sum_{h \in \mathcal{H}} y_h \sum_{i=1}^{\|\widehat{y}_{\ell-1}\|_0 |\mathcal{X}|} w_i \mathbb{I}\{h(u_i) \neq v_i\}$$

for some weights $w_i$. It follows that if $\|\widehat{y}_{\ell-1}\|_0$ is polynomial in problem parameters then the optimization for $\widehat{y}_\ell$ can be written as

$$\min_{k \in [\ell']} \min_{y \in \mathcal{Y}_k} \min_{\alpha \in \mathcal{A}} \mathcal{R}(y; \alpha) + f(\alpha)$$

for $\mathcal{R}(y; \alpha)$ a CSC loss over only polynomially many points (as well as linear in $y$ and convex in $\alpha$), and $f(\alpha)$ convex in $\alpha$. Note also that, for any $y$, we can upper bound $\mathcal{R}(y; \alpha) \leq \mathcal{O}(\frac{1}{\alpha} + d\alpha)$. Here $\mathcal{Y}_k$ a set of the form

$$\left\{ z \in \triangle_{\mathcal{H}} : \sum_{h \in \mathcal{H}} z_h c\left(\epsilon_k + \mathfrak{p}(\widehat{\Delta}^{k-1}(h)) + \max_{j \in [n]} \mathfrak{p}(-\widehat{\Delta}_{\text{safe}}^{j,k-1}(h)) + \min_{j \in [n]} |\widehat{\Delta}_{\text{safe}}^{j,k-1}(h)|\right) \leq \sum_{h \in \mathcal{H}} z_h \widehat{\Delta}_{\text{safe}}^{i,k}(h), \forall i \in [n] \right\}$$

$\widehat{y}_\ell$ will be the element in $\mathcal{Y}_k$ minimizing the, for the $k$ achieving the minimum. The dual of this problem has the form

$$\min_{k \in [\ell']} \min_{z \in \triangle_{\mathcal{H}}} \min_{\alpha \in \mathcal{A}} \max_{\mu_i \geq 0, i \in [n]} \mathcal{R}(z; \alpha) + f(\alpha)$$

$$+ \sum_{i=1}^{n} \mu_i \left( \sum_{h \in \mathcal{H}} z_h c\left(\epsilon_k + \mathfrak{p}(\widehat{\Delta}^{k-1}(h)) + \max_{j \in [n]} \mathfrak{p}(-\widehat{\Delta}_{\text{safe}}^{j,k-1}(h)) + \min_{j \in [n]} |\widehat{\Delta}_{\text{safe}}^{j,k-1}(h)|\right) - \sum_{h \in \mathcal{H}} z_h \widehat{\Delta}_{\text{safe}}^{i,k}(h)\right).$$

Note that we can swap the min over $\alpha$ and $z$ without issue. Furthermore, for a fixed $\mu$, the objective is linear in $z$, and for a fixed $z$, the objective is linear in $\mu$. By the minimax theorem, we can then swap the min and max to obtain the equivalent optimization:

$$\min_{k \in [\ell']} \min_{\alpha \in \mathcal{A}} \max_{\mu_i \geq 0, i \in [n]} \min_{z \in \triangle_{\mathcal{H}}} \mathcal{R}(z; \alpha) + f(\alpha)$$

$$+ \sum_{i=1}^{n} \mu_i \left( \sum_{h \in \mathcal{H}} z_h c\left(\epsilon_k + \mathfrak{p}(\widehat{\Delta}^{k-1}(h)) + \max_{j \in [n]} \mathfrak{p}(-\widehat{\Delta}_{\text{safe}}^{j,k-1}(h)) + \min_{j \in [n]} |\widehat{\Delta}_{\text{safe}}^{j,k-1}(h)|\right) - \sum_{h \in \mathcal{H}} z_h \widehat{\Delta}_{\text{safe}}^{i,k}(h)\right).$$

We can simply enumerate over $k$ and $\alpha$, as there are a finite number of each of these constraints. For a fixed $k$ and $\alpha$, to solve the inner maxmin problem, we can apply the approach proposed in [1]. In particular, we alternate between running the exponential gradient algorithm for the $\mu$ player, and computing the best-response for the $z$ player. The update to the $\mu$ player is trivial, as the problem is simply linear in $\mu$ (in practice, as in [1], we will also upper bound the domain of $\mu_i$ by some value $B$, to ensure this is finite).

Computing the best-response for the $z$ player (with $\mu$ fixed) is slightly trickier. Ignoring all other parameters, which are all currently fixed, the minimization over $z$ can be written as

$$\min_{z \in \triangle_{\mathcal{H}}} \sum_{h \in \mathcal{H}} z_h \sum_t a_t \mathbb{I}\{h(u_t) \neq o_t\}$$

$$+ \sum_{i=1}^{n} \mu_i \left( \sum_{h \in \mathcal{H}} z_h c\left(\epsilon_k + \mathfrak{p}(\widehat{\Delta}^{k-1}(h)) + \max_{j \in [n]} \mathfrak{p}(-\widehat{\Delta}_{\text{safe}}^{j,k-1}(h)) + \min_{j \in [n]} |\widehat{\Delta}_{\text{safe}}^{j,k-1}(h)|\right) - \sum_{h \in \mathcal{H}} z_h \widehat{\Delta}_{\text{safe}}^{i,k}(h)\right).$$

$$= \min_{z \in \triangle_{\mathcal{H}}} \sum_{h \in \mathcal{H}} z_h \left( \sum_t a_t \mathbb{I}\{h(u_t) \neq o_t\}\right.$$

$$+ \sum_{i \in [n]} c_i \left( \epsilon_k + \mathfrak{p}(\widehat{\Delta}^{k-1}(h)) + \max_{j \in [n]} \mathfrak{p}(-\widehat{\Delta}_{\text{safe}}^{j,k-1}(h)) + \min_{j \in [n]} |\widehat{\Delta}_{\text{safe}}^{j,k-1}(h)| - \widehat{\Delta}_{\text{safe}}^{i,k}(h) \right) \right).$$

Now note that $\max_{j \in [n]} \mathfrak{p}(-\widehat{\Delta}_{\text{safe}}^{j,k-1}(z)) = \sup_{\widetilde{\lambda} \in \triangle_n} \sum_{j \in [n]} \widetilde{\lambda}_j \mathfrak{p}(-\widehat{\Delta}_{\text{safe}}^{j,k-1}(z))$, and similarly for $\min_{j \in [n]} |\widehat{\Delta}_{\text{safe}}^{j,k-1}(z)|$. Using this, we can rewrite the above optimization as

$$\min_{z \in \triangle_{\mathcal{H}}} \max_{\widetilde{\lambda}^{1h} \in \triangle_n, h \in \mathcal{H}} \min_{\widetilde{\lambda}^{2h} \in \triangle_n, h \in \mathcal{H}} \sum_{h \in \mathcal{H}} z_h \left( \sum_t a_t \mathbb{I}\{h(u_t) \neq o_t\} \right.$$
$$\left. + \sum_{i \in [n]} c_i \left( \epsilon_k + \mathfrak{p}(\widehat{\Delta}^{k-1}(h)) + \sum_{j \in [n]} \widetilde{\lambda}_j^{1h} \mathfrak{p}(-\widehat{\Delta}_{\text{safe}}^{j,k-1}(h)) + \sum_{j \in [n]} \widetilde{\lambda}_j^{2h} |\widehat{\Delta}_{\text{safe}}^{j,k-1}(h)| - \widehat{\Delta}_{\text{safe}}^{i,k}(h) \right) \right).$$

We also have:

$$\mathfrak{p}(-\widehat{\Delta}_{\text{safe}}^{j,k-1}(z)) = \max_{\beta \in [0,1]} -\beta \widehat{\Delta}_{\text{safe}}^{j,k-1}(z), \quad |\widehat{\Delta}_{\text{safe}}^{j,k-1}(z)| = \max_{\beta \in [-1,1]} \beta \widehat{\Delta}_{\text{safe}}^{j,k-1}(z).$$

So we can further simplify the above to:

$$\min_{z \in \triangle_{\mathcal{H}}} \max_{\widetilde{\lambda}^{1h} \in \triangle_n, h \in \mathcal{H}} \min_{\widetilde{\lambda}^{2h} \in \triangle_n, h \in \mathcal{H}} \max_{\beta_1^h, \beta_2^{hj} \in [0,1], \beta_3^{hj} \in [-1,1], h \in \mathcal{H}} \sum_{h \in \mathcal{H}} z_h \left( \sum_t a_t \mathbb{I}\{h(u_t) \neq o_t\} \right.$$
$$\left. + \sum_{i \in [n]} c_i \left( \epsilon_k + \beta_1^h \widehat{\Delta}^{k-1}(h) - \sum_{j \in [n]} \widetilde{\lambda}_j^{1h} \beta_2^{hj} \widehat{\Delta}_{\text{safe}}^{j,k-1}(h) + \sum_{j \in [n]} \widetilde{\lambda}_j^{2h} \beta_3^{hj} \widehat{\Delta}_{\text{safe}}^{j,k-1}(h) - \widehat{\Delta}_{\text{safe}}^{i,k}(h) \right) \right).$$

Note that the objective is linear in $\beta$ and $\widetilde{\lambda}^2$, and both have continuous, compact, convex constraint sets, so we can swap the min and max to get that the above is equivalent to

$$\min_{z \in \triangle_{\mathcal{H}}} \max_{\widetilde{\lambda}^{1h} \in \triangle_n, h \in \mathcal{H}} \max_{\beta_1^h, \beta_2^{hj} \in [0,1], \beta_3^{hj} \in [-1,1], h \in \mathcal{H}} \min_{\widetilde{\lambda}^{2h} \in \triangle_n, h \in \mathcal{H}} \sum_{h \in \mathcal{H}} z_h \left( \sum_t a_t \mathbb{I}\{h(u_t) \neq o_t\} \right.$$
$$\left. + \sum_{i \in [n]} c_i \left( \epsilon_k + \beta_1^h \widehat{\Delta}^{k-1}(h) - \sum_{j \in [n]} \widetilde{\lambda}_j^{1h} \beta_2^{hj} \widehat{\Delta}_{\text{safe}}^{j,k-1}(h) + \sum_{j \in [n]} \widetilde{\lambda}_j^{2h} \beta_3^{hj} \widehat{\Delta}_{\text{safe}}^{j,k-1}(h) - \widehat{\Delta}_{\text{safe}}^{i,k}(h) \right) \right).$$

We can write this in the form

$$\min_{z \in \triangle_{\mathcal{H}}} \max_{\widetilde{\lambda}^1, \beta} g(z; \widetilde{\lambda}^1, \beta) \tag{22}$$

for

$$g(z; \widetilde{\lambda}^1, \beta) := \min_{\widetilde{\lambda}^{2h} \in \triangle_n, h \in \mathcal{H}} \sum_{h \in \mathcal{H}} z_h \left( \sum_t a_t \mathbb{I}\{h(u_t) \neq o_t\} \right.$$
$$\left. + \sum_{i \in [n]} c_i \left( \epsilon_k + \beta_1^h \widehat{\Delta}^{k-1}(h) - \sum_{j \in [n]} \widetilde{\lambda}_j^{1h} \beta_2^{hj} \widehat{\Delta}_{\text{safe}}^{j,k-1}(h) + \sum_{j \in [n]} \widetilde{\lambda}_j^{2h} \beta_3^{hj} \widehat{\Delta}_{\text{safe}}^{j,k-1}(h) - \widehat{\Delta}_{\text{safe}}^{i,k}(h) \right) \right).$$

To solve this, we will apply a version of Frank-Wolfe that handles adversarial losses to the outer player (see Section 4.2 of [15]), and will play best response for the inner player.

From the perspective of the outer player, at iteration $t$ of the algorithm given in [15], they must optimize the function

$$f_t(z) = g(z; \widetilde{\lambda}_t^1, \beta_t) = \sum_{h \in \mathcal{H}} z_h c_h(\widetilde{\lambda}_t^1, \beta_t)$$

for some $c_h(\widetilde{\lambda}_t^1, \beta_t)$. Note that this is $L = \max_h |c_h(\widetilde{\lambda}_t^1, \beta_t)|$ Lipschitz in the $\ell_1$-norm, and that we can bound this $L$ for all $t$ by something like $\mathcal{O}(\frac{1}{\alpha} + d\alpha + n)$. The algorithm introduced in Section 4.2 of [15] computes the standard FW update

$$\widetilde{z}_t = \arg\min_{z \in \triangle_{\mathcal{H}}} \nabla F_t(z_t)^\top z, \quad z_{t+1} = (1 - t^{-1/4})z_t + t^{-1/4}\widetilde{z}_t$$

for

$$F_t(z) = \frac{1}{t} \sum_{\tau=1}^{t} \nabla f_\tau(z_\tau)^\top z + \sigma_t \|z - z_1\|_2^2$$

for $\sigma_t = (L/D)t^{-1/4}$ for $D = \max_{z_1,z_2 \in \triangle_\mathcal{H}} \|z_1 - z_2\|_1$ (note that in that work, the function seems to be Lipschitz in the $\ell_2$ norm while here we use $\ell_1$—this does not seem to change their result at all). It is shown in [15] that running this procedure we obtain the bound, for any $z \in \triangle_\mathcal{H}$,

$$\sum_{t=1}^{T} (f_t(z_t) - f_t(z)) \leq 57LDT^{3/4}.$$

It follows that if we are able to compute $\widetilde{z}_t$ efficiently, and if the max player plays best response (and the best response can be computed efficiently), using analysis similar to that in [1], we can show that an approximate solution to (22) will be found in a polynomial number of iterations.

**Computing the Best Response for $\widetilde{\lambda}^1, \beta$.** For the inner player, they must solve

$$\max_{\widetilde{\lambda}^1, \beta} g(z_t; \widetilde{\lambda}^1, \beta).$$

Assume that $\|\widetilde{z}_t\|_0 \leq m$ for each $t$, and that $\|z_1\|_0 = 1$. Then $z_t$ will be $(mt + 1)$-sparse, so the sum in $g(z_t; \widetilde{\lambda}^1, \beta)$ will contain at most $(mt + 1)$ values. Note that the optimization over $\beta^h$ and $\widetilde{\lambda}^{1h}$ is completely independent, so to compute the best-response, we need to solve the following problem at most $(mt + 1)$ times:

$$\max_{\widetilde{\lambda}^{1h} \in \triangle_n} \max_{\beta_1^h, \beta_2^{hj} \in [0,1], \beta_3^{hj} \in [-1,1]} \min_{\widetilde{\lambda}^{2h} \in \triangle_n} \sum_{i \in [n]} c_i \left( \beta_1^h \widehat{\Delta}^{k-1}(h) - \sum_{j \in [n]} \widetilde{\lambda}_j^{1h} \beta_2^{hj} \widehat{\Delta}_{\text{safe}}^{j,k-1}(h) + \sum_{j \in [n]} \widetilde{\lambda}_j^{2h} \beta_3^{hj} \widehat{\Delta}_{\text{safe}}^{j,k-1}(h) \right).$$

The optimization over the first two terms is trivial and can be solved by enumerating. The third term now is a maxmin problem, however, this can also be solved trivially as it is equivalent to $\min_{j \in [n]} |\widehat{\Delta}_{\text{safe}}^{j,k-1}(h)|$. Note that each of these gap terms is themself the solution to an optimization over $\alpha \in \mathcal{A}$, but that can be solved easily for each (since there are at most polynomial of them), so they can be regarded as constants.

Thus, we conclude that the best response for $\widetilde{\lambda}^1, \beta$ can be computed efficiently, assuming that $m$ is polynomial in problem parameters. Note that the values of $\beta^h$ and $\widetilde{\lambda}^{1h}$ do not matter for $h \notin \text{support}(z_t)$ do not matter to compute the best response, so we can set them to the same value for all $h \notin \text{support}(z_t)$.

**Computing $\widetilde{z}_t$.** It remains to show that we can efficiently find a near-optimal $\widetilde{z}_t$ such that $\|\widetilde{z}_t\|_0 \leq m$. The optimization for $\widetilde{z}_t$ will have the form

$$\widetilde{z}_t = \arg\min_{z \in \triangle_\mathcal{H}} \sum_{\tau=1}^{t} \nabla f_\tau(z_\tau)^\top z + 2\sigma_t(z_t - z_1)^\top z$$

for

$$[\nabla f_\tau(z_\tau)]_h = c_h(\widetilde{\lambda}^1, \beta_\tau)$$
$$= \min_{\widetilde{\lambda}^{2h} \in \triangle_n} \sum_j a_j \mathbb{I}\{h(u_j) \neq o_j\} + \sum_{i \in [n]} c_i \left( \epsilon_k + \beta_{1\tau}^h \widehat{\Delta}^{k-1}(h) - \sum_{j \in [n]} \widetilde{\lambda}_{j\tau}^{1h} \beta_{2\tau}^{hj} \widehat{\Delta}_{\text{safe}}^{j,k-1}(h) \right.$$
$$\left. + \sum_{j \in [n]} \widetilde{\lambda}_j^{2h} \beta_{3\tau}^{hj} \widehat{\Delta}_{\text{safe}}^{j,k-1}(h) - \widehat{\Delta}_{\text{safe}}^{i,k}(h) \right).$$

Let $C_t \subseteq \mathcal{H}$ denote the classifiers supported on $z_t$ and assume that $z_1$ is only supported on a single classifier $h_0$. Note from our discussion on computing the best-response for the $\widetilde{\lambda}^1$ and $\beta$ player, we have that $\beta^h$ and $\widetilde{\lambda}^{1h}$ are identical for all $h \notin C_t$. We can therefore rewrite the above objective as (dropping the $\tau$ subscript and denoting, e.g. $\beta_1^h = \sum_{\tau=1}^{t} \beta_{1\tau}^h$):

$$\min_{\widetilde{\lambda}^{2h} \in \triangle_n, h \in \mathcal{H}} \sum_{h \in \mathcal{H} \setminus C_t} z_h \left( \sum_j a_j \mathbb{I}\{h(u_j) \neq o_j\} + \sum_{i \in [n]} c_i \left( \epsilon_k + \beta_1 \widehat{\Delta}^{k-1}(h) - \sum_{j \in [n]} \widetilde{\lambda}_j^1 \beta_2^j \widehat{\Delta}_{\text{safe}}^{j,k-1}(h) \right. \right.$$

$$+ \sum_{j \in [n]} \widetilde{\lambda}_j^{2h} \beta_3^j \widehat{\Delta}_{\text{safe}}^{j,k-1}(h) - \widehat{\Delta}_{\text{safe}}^{i,k}(h) \Big) \Big)$$

$$+ \sum_{h \in C_t} \Big( \sum_j a_j \mathbb{I}\{h(u_j) \neq o_j\} + \sum_{i \in [n]} c_i \Big( \epsilon_k + \beta_{1\tau}^h \widehat{\Delta}^{k-1}(h) - \sum_{j \in [n]} \widetilde{\lambda}_{j\tau}^{1h} \beta_{2\tau}^{hj} \widehat{\Delta}_{\text{safe}}^{j,k-1}(h)$$

$$+ \sum_{j \in [n]} \widetilde{\lambda}_j^{2h} \beta_{3\tau}^{hj} \widehat{\Delta}_{\text{safe}}^{j,k-1}(h) - \widehat{\Delta}_{\text{safe}}^{i,k}(h) \Big) + 2\sigma_t z_t \Big) - 2\sigma_t z_{h_0}.$$

We will focus first on the sum over $\mathcal{H} \backslash C_t$. Note that $\widehat{\Delta}^{k-1}(h)$ and $\widehat{\Delta}_{\text{safe}}^{j,k}(h)$ are both of the form

$$\min_{\alpha \in \mathcal{A}} \sum_t \frac{1}{w_t + \alpha} \mathbb{I}\{h(u_t) \neq o_t\} + \alpha \sum_t \widetilde{w}_t \mathbb{I}\{h(u_t) \neq o_t\} + \frac{c}{\alpha}.$$

Given this, we can rewrite the minimization over the first term as (where the $\widetilde{\alpha}$ correspond to the gaps that have negative coefficients, which is where the max comes from):

$$\min_{z \in \triangle_{\mathcal{H}}} \min_{\widetilde{\lambda}^{2h} \in \triangle_n, h \in \mathcal{H}} \min_{\alpha^h \in \mathcal{A}^k, h \in \mathcal{H}} \max_{\widetilde{\alpha}^h \in \mathcal{A}^k, h \in \mathcal{H}} \sum_{h \in \mathcal{H} \backslash C_t} z_h \Big( \mathcal{R}(h; \alpha^h, \widetilde{\alpha}^h, \widetilde{\lambda}^{2h}) + f(\alpha^h, \widetilde{\lambda}^{2h}) + g(\widetilde{\alpha}^h, \widetilde{\lambda}^{2h}) \Big)$$

for $\mathcal{R}$ convex in $\alpha$, and concave in $\widetilde{\alpha}$, $f$ convex in $\alpha$, and $g$ concave in $\widetilde{\alpha}$, and all functions are linear in $\widetilde{\lambda}^2$. Normally $\mathcal{A}$ is a discrete set, but if we let $\widetilde{\mathcal{A}}$ be a continuous relaxation of it, we can rewrite the above as

$$\min_{z \in \triangle_{\mathcal{H}}} \max_{\widetilde{\alpha}^h \in \mathcal{A}^k, h \in \mathcal{H}} \min_{\widetilde{\lambda}^{2h} \in \triangle_n, h \in \mathcal{H}} \min_{\alpha^h \in \mathcal{A}^k, h \in \mathcal{H}} \sum_{h \in \mathcal{H} \backslash C_t} z_h \Big( \mathcal{R}(h; \alpha^h, \widetilde{\alpha}^h, \widetilde{\lambda}^{2h}) + f(\alpha^h, \widetilde{\lambda}^{2h}) + g(\widetilde{\alpha}^h, \widetilde{\lambda}^{2h}) \Big).$$

To solve this we can again apply the FW algorithm of [15] with the max player playing best-response. As before, as long as $\mathfrak{z}_t$ (where $\mathfrak{z}_t$ denotes the update for this inner optimization) is sparse, we can efficiently compute the best-response for the $\widetilde{\alpha}$ player, since we only need to compute it for $h \in \mathfrak{z}_t$. The FW-style update will then have the form

$$\min_{z \in \triangle_{\mathcal{H}}} \min_{\widetilde{\lambda}^{2h} \in \triangle_n, h \in \mathcal{H}} \min_{\alpha^h \in \mathcal{A}^k, h \in \mathcal{H}} \sum_{h \in \mathcal{H} \backslash C_t} z_h \Big( \mathcal{R}(h; \alpha^h, \widetilde{\alpha}_t^h, \widetilde{\lambda}^{2h}) + f(\alpha^h, \widetilde{\lambda}^{2h}) + g(\widetilde{\alpha}_t^h, \widetilde{\lambda}^{2h}) \Big)$$

$$= \min_{\widetilde{\lambda}^{2h} \in \triangle_n, h \in \mathcal{H}} \min_{\alpha^h \in \mathcal{A}^k, h \in \mathcal{H}} \min_{h \in \mathcal{H} \backslash C_t} \mathcal{R}(h; \alpha^h, \widetilde{\alpha}_t^h, \widetilde{\lambda}^{2h}) + f(\alpha^h, \widetilde{\lambda}^{2h}) + g(\widetilde{\alpha}_t^h, \widetilde{\lambda}^{2h})$$

where the equality follows since we can always swap min, and since there will always be an optimal solution supported on a single $h$. We can solve the inner min using a CSC oracle that is able to optimize over a set $\mathcal{H} \backslash C_t$, and by enumerating $\widetilde{\lambda}^2$ and $\alpha$ (since we can always find an optimal solution supported on a single $h$, we can set $\widetilde{\lambda}^{2h}, \alpha^h$ identical for all $h$ and will arrive at the same minimum).

This will converge in polynomially many steps, and will produce some $\mathfrak{z}_{t'}$ which is $m$-sparse (for $m$ polynomial in parameters). It follows that $\mathfrak{z}_{t'}$ is the near-optimal value for $\widetilde{z}_t$ supported on $\mathcal{H} \backslash C_t$. To pick a final value for $\widetilde{z}_t$, we can simply enumerate over the (polynomially many) $h \in C_t$, compute their loss values, and then pick the minimum out of those and the value achieved by $\mathfrak{z}_{t'}$. This procedure will always return some $\widetilde{z}_t$ supported on at most polynomially many $h$, so $m$ can be chosen suitably to make the best-response of the max player efficient.

Putting all of this together, we can efficiently solve for $\widehat{y}_\ell$.

### E.1.2 Solving for $\lambda_\ell$

We turn now to solving the optimization (21). Using arguments similar to what we have already shown, we have that

$$(21) = \inf_{\lambda \in \triangle_{\mathcal{X}}} \max_{z \in \mathcal{Z}} \min_{\alpha \in \mathcal{A}, \alpha_2, \ldots, \alpha_p \in \mathcal{A}} \max_{\beta_1, \ldots, \beta_m \in \mathcal{B}} \mathcal{R}(z; \alpha, \alpha_2, \ldots, \alpha_p, \beta_1, \ldots, \beta_m)$$

$$+ 2\alpha \sum_{U \in \mathcal{X}} \frac{\nu(U)^2}{9\lambda(U)/10 + 1/10d} \mathbb{I}\{z(U) \neq \widehat{z}_{\ell-1}(U)\} + f(\alpha, \alpha_2, \ldots, \alpha_p, \beta_1, \ldots, \beta_m).$$

As before, we can simply enumerate over all possible choices of $\alpha$ and $\beta$. For a fixed setting of $\alpha$ and $\beta$, to solving the $\inf$ over $\lambda$, we can apply Mirror Descent. In this case we choose the mirror map to be the negative entropy, which is strongly convex with respect to the $\ell_1$ norm.

Given this, to solve this in a computationally efficient manner, all we need is that the objective is convex (which it is) and Lipschitz with respect to the $\ell_1$ norm. Let $g(\lambda)$ denote the objective of the above optimization. By the Mean Value Theorem,

$$|g(\lambda) - g(\widetilde{\lambda})| = \nabla g((1-c)\lambda + c\widetilde{\lambda})^\top (\lambda - \widetilde{\lambda})$$

for some $c \in [0, 1]$. So, for any $\lambda, \widetilde{\lambda} \in \triangle_{\mathcal{X}}$, we can bound

$$|g(\lambda) - g(\widetilde{\lambda})| \leq \left( \sup_{\lambda' \in \triangle_{\mathcal{X}}} \|\nabla g(\lambda')\|_\infty \right) \cdot \|\lambda - \widetilde{\lambda}\|_1.$$

We have,

$$\frac{d}{dt} \sum_{U \in \mathcal{X}} \frac{\nu(U)^2}{9\lambda(U)/10 + 1/10d + 9t\lambda_0(U)/10} \mathbb{I}\{z(U) \neq \widehat{z}_{\ell-1}(U)\}|_{t=0}$$
$$= \sum_{U \in \mathcal{X}} \frac{-\lambda_0(U)\nu(U)^2}{(9\lambda(U)/10 + 1/10d)^2} \mathbb{I}\{z(U) \neq \widehat{z}_{\ell-1}(U)\}.$$

It follows that

$$\sup_{\lambda' \in \triangle_{\mathcal{X}}} \|\nabla g(\lambda')\|_\infty \leq 100d^2$$

so we can apply Mirror Descent to optimize the above with computational complexity scaling only polynomially in problem parameters.

### E.2    Computational Efficiency of BESIDE

The primary computational cost of BESIDE is incurred by calling RAGE$^\epsilon$, and solving the optimization on Line 6 of Algorithm 1. We have already shown that RAGE$^\epsilon$ can be run in a computationally efficient manner. The optimization on Line 6 has a form very similar to the optimization we solve in RAGE$^\epsilon$, so the same argument and solution approach (applying Mirror Descent) allows us to compute the optimal distribution, $\lambda_\ell$, here as well.

## F    Experimental details and additional results

### F.1    Experimental details

All code was written in Python and run on a Intel Xeon 6226R CPU with 64 cores.

Algorithm 6 is the precise implementation of BESIDE using elimination. It largely resemble to Algorithm 1, with the difference that it explicitly eliminates arms.

### F.2    Additional results

We evaluate Algorithm 2 and the passive baseline on two other datasets. Recall that the passive baseline selects points uniformly at randoms from the pool of examples $\mathcal{X}$ and then retrains the model using the same Constrained Empirical Risk Minimization oracle (CERM).

**Half circle dataset.**    We consider a two-dimensional half circle dataset, visualized on Figure 10. We report in Figures 11 and 12 the precision and (respectively) the recall obtained when varying the number of labels given to each method. The confidence intervals are obtained over 25 repetitions. We observe that Algorithm 2 allows us to provide a classifier satisfying a given recall or precision in far fewer queries. This is in line with the results of [16] on One Dimensional Thresholds, where the sample complexity of the active strategy is $O(log(n))$ while the sample complexity of the passive strategy is at least of order $n$.

---
**Algorithm 6** Best Safe Arm Identification with Elimination
---
1: **input:** tolerance $\epsilon$, confidence $\delta$
2: $\iota_\epsilon \leftarrow \lceil \log(\frac{1}{\epsilon}) \rceil$, $\mathcal{Z}_{\text{active}}^0 \leftarrow \mathcal{Z}$, $\mathcal{Z}_{\text{safe}}^0 \leftarrow \emptyset$
3: **for** $\ell = 1, 2, \ldots, \iota_\epsilon$ **do**
4:     $\epsilon_\ell \leftarrow 2^{-\ell}$
5:     Compute allocation $\mathcal{X}\mathcal{Y}_{\text{safe}}$ on $\mathcal{Z}_{\text{active}}^{\ell-1}$ and sample from it $\tau_\ell = \mathcal{O}(\mathcal{X}\mathcal{Y}_{\text{safe}}(\mathcal{Z}_{\text{active}}^{\ell-1})/\epsilon_\ell^2)$ times
6:     $\widehat{\mu}^\ell \leftarrow \mathsf{RIPS}(\{(x_t, s_{t,i})\}_{t=1}^{\tau_\ell}, \mathcal{Z}, \frac{\delta}{2\ell^2})$
7:     Set $\widehat{\Delta}_{\text{safe}}^\ell(z) \leftarrow \gamma - z^\top \widehat{\mu}^\ell$ for all $z \in \mathcal{Z}_{\text{active}}^{\ell-1}$ and

$$\widetilde{Z}_{\text{active}}^\ell = \{z \in \widetilde{Z}_{\text{active}}^{\ell-1} \; : \; \widehat{\Delta}_{\text{safe}}^\ell(z) \in [-\epsilon_\ell, 2\epsilon_\ell]\} \quad \widetilde{Z}_{\text{safe}}^\ell = \{z \in \widetilde{Z}_{\text{active}}^{\ell-1} \; : \; \widehat{\Delta}_{\text{safe}}^\ell(z) \geq 2\epsilon_\ell]\}$$

8:     $\mathcal{Z}_{\text{active}}^\ell, \mathcal{Z}_{\text{safe}}^\ell \leftarrow \textsc{Rage-elim}^\epsilon\Big(\widetilde{Z}_{\text{active}}^\ell \cup \widetilde{Z}_{\text{safe}}^\ell \cup \mathcal{Z}_{\text{safe}}^{\ell-1}, \widetilde{Z}_{\text{safe}}^\ell \cup \mathcal{Z}_{\text{safe}}^{\ell-1}, \epsilon_\ell\Big)$

9: $\mathcal{Z}_{\text{final}}, \emptyset \leftarrow \textsc{Rage-elim}^\epsilon\Big(\mathcal{Z}_{\text{active}}^\ell \cup \mathcal{Z}_{\text{safe}}^\ell, \mathcal{Z}_{\text{active}}^\ell \cup \mathcal{Z}_{\text{safe}}^\ell, \epsilon_\ell\Big)$
10: **return** Any arm in $\mathcal{Z}_{\text{final}}$.
---

---
**Algorithm 7** $\textsc{Rage-elim}^\epsilon$
---
1: **input:** active set $\mathcal{Z}$, optimal set $\mathcal{Y}$, tolerance $\epsilon$
2: $\iota_\epsilon \leftarrow \lceil \log(\frac{1}{\epsilon}) \rceil$, $\mathcal{Z}^0 \leftarrow \mathcal{Z}$, $\mathcal{Y}^0 \leftarrow \mathcal{Y}$
3: **for** $\ell = 1, 2, \ldots, \iota_\epsilon$ **do**
4:     $\epsilon_\ell \leftarrow 2^{-\ell}$
5:     Compute allocation $\mathcal{X}\mathcal{Y}_{\text{diff}}$ on $(\mathcal{Z}^{\ell-1} \cup \mathcal{Y}^{\ell-1}, \mathcal{Y}^{\ell-1})$ and sample from it $\tau_\ell = \mathcal{O}(\mathcal{Z}^{\ell-1} \cup \mathcal{Y}^{\ell-1}, \mathcal{Y}^{\ell-1})/\epsilon_\ell^2)$ times
6:     $\widehat{\theta}^\ell \leftarrow \mathsf{RIPS}(\{(x_t, s_{t,i})\}_{t=1}^{\tau_\ell}, \mathcal{Z}, \frac{\delta}{2\ell^2})$
7:     Set $\widehat{\Delta}^\ell(z) \leftarrow \max_{y \in \mathcal{Y}^{\ell-1}} y^\top \widehat{\theta}^\ell - z^\top \widehat{\theta}^\ell$ for all $z \in \mathcal{Z} \cup \mathcal{Y}$ and

$$\mathcal{Z}^\ell = \{z \in \mathcal{Z}^{\ell-1} \; : \; \widehat{\Delta}^\ell(z) \leq \epsilon_\ell\} \quad \mathcal{Y}^\ell = \{y \in \mathcal{Y}^{\ell-1} \; : \; \widehat{\Delta}^\ell(y) \leq \epsilon_\ell]\}$$

8: **return** $\mathcal{Z}^\ell, \mathcal{Y}^\ell$
---

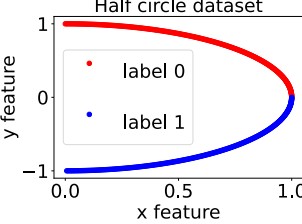

Figure 10: Half circle dataset.

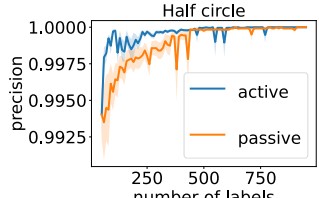

Figure 11: Precision

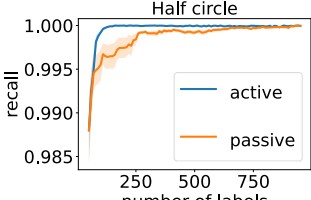

Figure 12: Recall