# OpenReview forum: "Active Learning with Safety Constraints"
_NeurIPS.cc/2022/Conference — NeurIPS 2022 Accept_

### Official Review · Reviewer_4PUH · 2022-07-08

**Rating:** 5
**Confidence:** 2
**Soundness:** 3 good
**Presentation:** 3 good
**Contribution:** 2 fair

**Summary:**

In this paper, the authors attempt to learn the (near) optimal decision in the linear bandit setting with the additional imposition of safety (linear) constraints. They supplement their results with experiments on real-world data sets.


First they define the STBAI problem which aims to find an arm whose utility is maximum but also satisfies a safety requirement i.e its expected risk is below a threshold \gamma.

Then they define an $\epsilon$ approximation to the STBAI problem the goal is to have the expected utility of our chosen arm within $\epsilon$ of that of the best arm and we violate the safety constraint inequality by at most $\epsilon$.


Then they give an algorithm to identify such an arm and explore the sample complexity (and computational complexity) of such a routine. They show that when safety constraints are moot, this complexity is essentially matched by the lower bounds for best-arm identification. Similar results can be shown for the standard linear bandits setting.


**Questions:**

Line 66: should be -\epsilon in eq(1)

Due to paucity of time, i did not verify the correctness of the technical results. But I was wondering if the techniques used in the paper are completely new or have been extended/motivated from techniques used in prior work. As such in the main paper, i did not see any references to related techniques.

What is a practical example of a safety constraint ? Apologies if this is already given in the paper but I might have missed it.


**Limitations:**

None.

**Strengths And Weaknesses:**

Originality: I am not familiar with this area, but I find this work quite original to the best of my limited knowledge.

Quality: The quality of writing is above average. I like how the setting is explained; how different scenarios correspond to previously known settings and the comparison. The algorithm is complicated but they do try to give an intuitive explanation.

Clarity: The non-technical part of the paper is clear to understand. As for the technical part, my domain is not in this area, but the theorem statements and the corresponding corollaries make sense to me.

Significance: Since this is not my field of expertise, i can not make a solid judgment on the significance, but to me, the significance as compared to prior work seems not path breaking but not too minor either.

---

> ### Author Response · Authors · 2022-08-02
> **Answer to Reviewer 4PUH**
>
> - We cover in the related work section the works we use to develop our techniques and build our algorithms. Notably, we inspire ourself from the experiment-design based algorithm Rage from the work of (Fiez et al., 2019).
> - Practical applications of the safe active learning problem cover any setting where the proportion of false-alarms (FDR) among those predicted is relatively small, as well as many other practical settings. Please refer to the responses to all reviewers for more details.

---

### Official Review · Reviewer_rbwS · 2022-07-09

**Rating:** 3
**Confidence:** 3
**Soundness:** 2 fair
**Presentation:** 3 good
**Contribution:** 2 fair

**Summary:**

This paper explores a constrained linear bandit problem, i.e., a learner that can maximize a linear objective subject to linear constraints when given access only to noisy observations of the objective and of constraint satisfaction for an input action. The proposed algorithm is proved to be $(\delta, \epsilon)$-PAC, in that it generates a solution that is at most $\epsilon$-suboptimal/infeasible with probability $1-\delta$; in addition, has a lower bounded number of samples. This methodology can be applied to active learning with custom constraints on model performance and the paper demonstrates its performance over naive random data collection policies.


**Questions:**

- In the active learning experiments, what is the FDR constraint right-hand-side? What is the level of constraint satisfaction?
- What is the batch size? And initial number of rounds? It is understood that active learning algorithms are often heavily sensitive to initial settings and rounds (Munjal et al.)
- My interpretation was that the passive sampler baseline always satisfies the constraints, due to the CERM. Is this interpretation correct? If so, any active learning sampler + CERM should be able to ensure the constraints are satisfied, then, how would other well-known active learning strategies perform, e.g., coreset?

**References**

Munjal et al., Towards Robust and Reproducible Active Learning using Neural Networks, CVPR’22



**Limitations:**

The paper clearly describes societal impact and potential limitations of their framework.

**Strengths And Weaknesses:**

**Strengths**

- The overall problem is interesting and very general, suggesting many potential applications. I only briefly checked the main theoretical result, but it seems sound.

**Weaknesses**

- There is very limited experimental analysis, especially compared to baselines. Pool-based active learning is a well-studied problem and although the constrained active learning problem is new, the paper only considers a passive random sampler baseline. It would be useful to see more meaningful baselines to justify the methodology. In related works, the authors mention several other papers in this space (Sui et al., Sui et al., Wang et al.). Can these be considered as baselines?

**References**

Sui et al., Safe exploration for optimization with gaussian processes.

Sui et al., Stagewise safe bayesian optimization with gaussian processes.

Wang et al., Best arm identification with safety constraints

---

> ### Author Response · Authors · 2022-08-02
> **Answer to Reviewer rbwS**
>
> - About the baselines. Note that we conduct experiments for two different problems: STBAI bandits and active learning with FDR constraints.
> - The focus of (Sui et al., 2015), (Sui et al., 2018) and (Wang et al., 2022) is the best-arm identification with safety constraints problem where the safety constraint must be met for every query point. The feedback structure of (Wang et al., 2022) does not apply in our setting as the action in their framework is to choose both a coordinate and a real value, but the actions are otherwise independent (in contrast to our setting where the linear structure allows information-sharing between actions). While in principle (Sui et al., 2015; Sui et al., 2018) could be applied in our setting, as shown in (Wang et al., 2022), this algorithm can be very suboptimal, and we do not expect it to perform competitively. Furthermore, it lacks a stopping criteria, which is a critical piece in our algorithm, and therefore lacks the type of theoretical guarantee we are aiming for. Finally, since both (Sui et al., 2015; Sui et al., 2018) and (Wang et al., 2022) consider the setting where the constraints must be met at every query point, we would expect them to be extremely conservative in our setting (where we do not require the constraints be met at query points), and therefore would expect them to in general perform quite poorly as compared with our approach.
> - As noted, the constrained active learning problem is new and---to the best of our knowledge---there does not exist any algorithm applicable to this problem beyond simple random sampling which we compared to. Constraining the safety for every query point does not match the constrained active learning problem where the agent may query the label of any data point of the dataset.
>
> - Active learning experimental details: the batch size was set to 25, the initial number of queried labels is 50 and the number of rounds is set so that the number of queried labels at the last round is half of the size of the dataset (that is max_round is of the order size_dataset / (2 batch_size) ). Although active learning algorithms are often heavily sensitive to initial settings and rounds, we did not notice any interesting changes of performance when varying these parameters in our experiments.
> - The constraint is set to be FDR < 0.15 and the level of constraint satisfaction - that is the value of FDR - is reported as the y axis in figure 5. We observed through the experiments that the active method needed half the number of samples as the passive sampling to achieve a given FDR.
> - As we observe in figure 5, the passive sampler baseline and our proposed algorithm do not always satisfy the constraints on the whole dataset. This is due to the fact that the CERM oracle gives a classifier that only empirically satisfies the constraints, on the given set of queried points. With that in mind, active learning samplers (e.g. coreset) using a CERM oracle will not ensure that the constraints are satisfied.

---

> > ### Comment · Reviewer_rbwS · 2022-08-08
> > **Thank you for the review, but I am still a little confused**
> >
> > Apologies for my delayed response.
> >
> > > The constraint is set to be FDR < 0.15 and the level of constraint satisfaction - that is the value of FDR - is reported as the y axis in figure 5. We observed through the experiments that the active method needed half the number of samples as the passive sampling to achieve a given FDR.
> >
> > Perhaps the constraint set was in the paper and I missed it, but if not, I strongly recommend stating it in the main paper since this allows a clearer exploration of this experiment. If I understand correctly, only the final setting ($n > 2000$) in Fig 5 actually satisfies the constraint. Further, at the point of constraint satisfaction, the difference in FDR seems at most 0.02-0.03, which is a very small margin of failure. The major improvement in terms of FDR of the proposed algorithm versus random is in low $n < 750$, but in those settings, neither method is actually beating the constraint. Perhaps you meant to encourage a soft constraint satisfaction with respect to AL? However, that is not clear in the presentation of the paper. I hope you can clarify if there is a misunderstanding on my part.
> >
> >
> > > As we observe in figure 5, the passive sampler baseline and our proposed algorithm do not always satisfy the constraints on the whole dataset. This is due to the fact that the CERM oracle gives a classifier that only empirically satisfies the constraints, on the given set of queried points. With that in mind, active learning samplers (e.g. coreset) using a CERM oracle will not ensure that the constraints are satisfied.
> >
> > If the proposed algorithm and the passive sampler both do not always satisfy the constraints, then doesn't this warrant further analysis with different active learning samplers? I agree standard AL samplers will not ensure that the constraints are satisfied, but is there a clear intuition why other AL samplers will perform better or worse? For example, maybe other AL strategies provide a considerable increase in FDR over random too---which would further justify the proposed method.

---

> > > ### Author Response · Authors · 2022-08-09
> > > **Thank you for your response**
> > >
> > > In the sense that any active learning samplers could be used with the CERM oracle, we could indeed provide a more complete benchmark. Note though that the main intention of this experiments was to illustrate that mixing between using a G-design and an XY-design enables to get an active sampler that needs much less samples as the passive sampler to achieve a given FDR. The experiments show a gain (half the number of samples are needed to achieve a given FDR), and also show how we can use insights from our provably correct algorithm for the STBAI problem to design a safe active learning algorithm.

---

### Official Review · Reviewer_QfJJ · 2022-07-10

**Rating:** 4
**Confidence:** 1
**Soundness:** 3 good
**Presentation:** 1 poor
**Contribution:** 3 good

**Summary:**

The paper proposes a study for the best arm identification problem and an algorithm that constructs an adaptive experimental design to efficiently balance exploration between the refinement of estimates and optimality gaps. The paper studies the problem from the perspective of a linear bandits setup. The algorithm is evaluated on a synthetic and a publicly available real-world dataset.

**Questions:**

- I wonder how realistic the problem setting actually is. The paper studies the problem of estimating both the value and safety of actions. However, in practice, in my opinion the safety of actions can oftentimes be defined a priori.
- A second question I have is about safety violations during learning. In practical applications – when learning on the job – safety violations are not permitted and can also cause harm. This essentially renders the approach useless in such settings, right? In such cases [32, 33] seem to be better suited…


**Limitations:**

Yes.

**Strengths And Weaknesses:**

Unfortunately, for me the paper was not easily accessible as I am not familiar with studies of active learning from a linear bandits perspective.

Strengths of the paper:
- The paper is very well-written
- The paper to me seems technically sound and theoretical assumptions are formally proved
- The algorithm is evaluated on synthetic and real data

Weaknesses of the paper:
- To me the paper was not easily accessible and could at points additionally resort to intuitive explanations (it is not within my field of expertise though)

To me the additional contribution over [35] could be outlines more clearly. In lines 316 to 321 the paper describes one big difference: safety constraints must be satisfied during training – which to me seems more reasonable. The rest of the paragraph only considers [32, 33] but does not further differentiate from [35].

Minor things:
- I was a bit confused by some terminology. In lines 19-21 the paper uses s_t for the price and strategies s \in S for bidding strategies. But those are not compatible right?
- In line 45, when introduction \mu, please immediately explain that these are the set of constraints (although mathematically clear, this is only mentioned at some remote point)
- Line 335: missing “.” after ‘constraints‘

---

> ### Author Response · Authors · 2022-08-02
> **Answer to Reviewer QfJJ**
>
> - In this illustrative example, $s_t$ and $s$ are indeed of different natures: the prices bid $s_t$ are real values and bidding strategies $s$ are mappings, with output prices bid $s_t$. We will update the notations to avoid this confusion between outputs of bidding strategies and prices bid.
> - One difference between our work and (Wang et al., 2022) is that we focus on the problem where constraint satisfaction will be satisfied for the output item while (Wang et al., 2022) focuses on the problem where the safety constraints must be satisfied during training. Another notable difference is that (Wang et al., 2022) covers the unstructured bandit setting---Multi Arm Bandits---while we tackle the (harder) structured bandit setting---Linear Bandits. Extending from multi-armed bandits to linear bandits is non-trivial. This leads to a novel and different trade-off between learning reward functions and constraint satisfaction.
> - There is a wide range of problems where the safety of actions must be learnt and cannot be defined a priori. For example, algorithms seeking for some form of personalization may learn the preferences of the users as safety constraints which cannot be defined a priori. Also the problem of satisfying some FDR constraints is a data-dependent problem where knowing whether some classifier violates FDR constraints is not possible ahead of the learning phase. In both of these examples, the safety constraint must be learnt. See also the examples given in the comment to all reviewers.
> - In the active learning with FDR constraint problem the agent may query the label of any data point and the learning phase is not restricted. Please refer to the responses to all reviewers for more details.

---

> > ### Comment · Reviewer_QfJJ · 2022-08-09
> > **Reply to Author Response**
> >
> > I would like to thank the authors for their response to my review.
> >
> > Unfortunately, I cannot make an educated guess about the depth of the contribution of the paper and how far it is going beyond state of the art mentioned in the paper. But I also see that this is a shared issue among the other reviews. That is why I intend to keep my score on this paper.

---

### Author Response · Authors · 2022-08-02
**Thank you for your reviews.**

Thank you for your careful read and review of our work. We wish to begin by clarifying both the relevance and the contributions of this work to all reviewers to contextualize our responses.

We first provide several examples where we believe our problem setting is compelling and realistic:
- The STBAI framework encompasses the following example: Consider the task of developing a medicinal drug by testing different doses of chemicals in petri dishes on a model organism (e.g., bacteria). In this problem, the ideal strategy would enable us to satisfy two desiderata: find a set of doses of chemicals that can cure as effectively as possible and---at the same time---make sure that this set of doses is safe to deploy to humans. As both the objective (effectiveness of the chemicals) and the constraint (safety on humans) require performing a petri dish experiment, the goal of the scientists designing this safe drug testing is to find a suitable set of chemicals with as few trials that involve human feedback as possible. It is acceptable to evaluate an unsafe controller during training, as this phase is performed on petri dishes; however, the constraints have to be satisfied during deployment.
- The active learning with FDR constraint covers the following example: consider the genomics task of detecting genome structures responsible for a disease. In such tasks, recent growth in computing power enables scientists to perform large and rapid data collection, yet due to the high-dimensional nature of such problems, reducing the sample complexity through active learning is still critical. Recall that the False Discovery Rate (FDR) corresponds to the rate of false-alarms which, in this setting, correponds to the rate of outcomes marked as significant (i.e. being responsible for a disease) but that are actually not. Constraining the FDR is crucial in order to identify promising genes for followup studies: in the active learning with FDR constraints framework, researchers may control the proportion of "false leads" they are willing to accept, while still taking advantage of the power of active learning to reduce sample complexity.
- We believe both the STBAI framework and active learning with FDR constraint are ubiquitous. STBAI is a general framework for any sequential decision making problem where both an objective and a constraint should be learnt in order to deploy a policy. The FDR constraint has gained a broad acceptance in many scientific fields (especially in the life sciences, from genetics to biochemistry, oncology and plant sciences)---see [1]. Lastly, high stake decisions for which data-driven algorithms have been deployed---such as credit assignments, facial authentication, court penalty decisions, or police interventions---ought to have a false-alarm constraint, and thus could be framed as active learning with FDR constraint problems.
- The recent, concurrent work "Interactively Learning Preference Constraints in Linear Bandits" (https://arxiv.org/pdf/2206.05255.pdf), published at ICML 2022, solves the easier problem where the reward model is known and only the constraints are to be learnt. It is known that the optimal design to learn constraints is the G-optimal design, while the optimal design to learn a reward model is the XY-optimal design. The key and novel challenge of our framework is to carefully balance between G and XY designs: naively spending enough budget to either learn the reward model (via a XY design) or to learn the safety constraints (via a G design) will fail catastrophically (see Example 2.1 and Proposition 1). Thus, our work builds on the aforementioned work in a critical way, that we believe is of interest to the community.

With that in mind, we believe that the STBAI problem is very relevant. The contributions of our work are twofold:
- A theoretically justified algorithm for provably safe decision making methods: we provide a provably correct algorithm for the STBAI problem, in the sense that the output is guaranteed to satisfy the given safety constraints.
- A practical algorithm for active learning with constraints: We also show how our theoretical findings can be of interest in practice, by designing the first sample efficient algorithm for active learning with FDR constraints. This novel active learning problem is critical for a number of scientific fields, as noted.

[1] Benjamini, Yoav. “Discovering the False Discovery Rate.” Journal of the Royal Statistical Society. Series B (Statistical Methodology), vol. 72, no. 4, 2010, pp. 405–16. JSTOR,

---

### Meta-Review · Area_Chair_q78f · 2022-08-25

**Recommendation:** Accept
**Confidence:** Less certain

**Metareview:**

3 reviewers evaluated the submitted paper, 2 recommending rejection (1x reject, 1x borderline reject), 1 borderline acceptance. There was some interaction between authors and reviewers and the reviewers considered the authors' responses. Unfortunately, the reviewers' expertise was more on the active learning side and less on the bandit side -- in that regard, the paper title is rather uninformative about what to expect from the paper. Therefore, I decided to read the paper closely myself.

--

Based on my own reading, I would argue that the paper makes some valuable contributions to the problem of solving safety-constrained bandits. The considered problem setting in which constraint violations are possible during exploration is relevant as motivated by the authors by several convincing examples.

The theoretical and algorithmic contributions (assuming the correctness of the proofs which I didn't check in large detail) are interesting, providing some first solid insights into a novel problem setting, and providing the first algorithm to solve the considered problem setting with guarantees.

The coverage of related work is mainly fine but, as also noted by the authors, is missing the very related paper by Lindner et al. (ICML'22) which investigates a similar but simpler problem setting.

The empirical evaluation is not very exhaustive but is in line with typical bandit papers. My only real criticism in that regard is that the chosen baseline is unnecessarily weak in that it estimates all safety gaps up to some desired precision while it would be sufficient to estimate those only until one can be sufficiently certain that an arm is not feasible. One can also think of other sensible baselines, e.g., a greedy policy that focuses on the most promising-looking arms until they are shown to be unsafe, if they are (of course one can then construct examples, where such a strategy will fail but for not constructed examples such a strategy might work reasonably well).

The write-up is mainly clear and easy to follow if one is familiar with general bandit literature. Nevertheless, the title of the paper is rather uninformative and might trigger incorrect expectations.

Overall, as already said above, I think the paper would be a valuable addition to the existing literature.

--

Taking into account my own reading of the paper, as well as the 3 submitted reviews, and the author's feedback, I am recommending acceptance of the paper. The main concerns of the critical reviews were regarding the relevance of the problem setting, violation of constraints during exploration, and the evaluation, in particular the considered baselines. While the raised concerns are valid, I think the authors sufficiently justify their problem setting and the empirical evaluation is also sufficient (although it could clearly be extended; see some suggestions in my review and also below). I see the paper's main contributions on the theoretical side - and in that regard, there are some novel valuable insights.

When preparing the camera-ready version of the paper, please consider the following suggestions for improving the paper:

* Adjust the discussion of related work: in particular, include Lindner et al. in the revised paper
* Consider the addition of further baselines to provide a better sense of the proposed algorithm's performance
* There are some minor issues with notation, e.g., line 55/56 should be an argmax. So please carefully revise your paper in that regard.
* Consider adjusting the title of your paper to better match it to the content
* Carefully consider all reviews, in particular also those recommending rejection, when preparing the camera-ready version and adjust your paper accordingly; For instance, I suggest implementing some of the authors' responses that came up in the discussion to emphasize the importance and practical relevance of the studied problem better.

**Award:**

No

---

### Decision · Program_Chairs · 2022-09-14

Accept